# LINEARLY CONSTRAINED BILEVEL OPTIMIZATION: A SMOOTHED IMPLICIT GRADIENT APPROACH

## ABSTRACT

This work develops an analysis and algorithms for solving a class of bilevel optimization problems where the lower-level (LL) problems have linear constraints. Most of the existing approaches for constrained bilevel problems rely on value function based approximate reformulations, which suffer from issues such as non-convex and non-differentiable constraints. In contrast, in this work, we develop an implicit gradient-based approach, which is easy to implement, and is suitable for machine learning applications. We first provide an in-depth understanding of the problem, by showing that the implicit objective for such problems is in general non-differentiable. However, if we add some small (linear) perturbation to the LL objective, the resulting implicit objective becomes differentiable almost surely. This key observation opens the door for developing (deterministic and stochastic) gradient-based algorithms similar to the state-of-the-art ones for unconstrained bi-level problems. We show that when the implicit function is assumed to be strongly-convex, convex and weakly-convex, the resulting algorithms converge with guaranteed rate. Finally, we experimentally corroborate the theoretical findings and evaluate the performance of the proposed framework on numerical and adversarial learning problems. To our knowledge, this is the first time that (implicit) gradient-based methods have been developed and analyzed for the considered class of bilevel problems.

## 1 INTRODUCTION

*Bilevel optimization problems* (Colson et al., 2005; Dempe & Zemkoho, 2020) can be used to model an important class of hierarchical optimization tasks with two levels of hierarchy, the *upper-level (UL)* and the *lower-level (LL)*. The key characteristics of bilevel problems are: 1) the solution of the UL problem requires access to the solution of the LL problem and, 2) the LL problem is parametrized by the UL variable. Bilevel optimization problems arise in a wide range of machine learning applications, such as meta-learning (Rajeswaran et al., 2019; Franceschi et al., 2018), data hypercleaning (Shaban et al., 2019), hyperparameter optimization (Sinha et al., 2020; Franceschi et al., 2018; 2017; Pedregosa, 2016), adversarial learning (Li et al., 2019; Liu et al., 2021a; Zhang et al., 2021), as well as in other application domains such as network optimization (Migdalas, 1995), economics (Cecchini et al., 2013), and transport research (Didi-Biha et al., 2006; Kalashnikov et al., 2010). In this work, we focus on a special class of stochastic bilevel optimization problems, where the LL problem involves the minimization of a *strongly convex* objective over a set of *linear inequality constraints*. More precisely, we consider the following formulation:

$$\min_{\mathbf{x} \in \mathcal{X}} \left\{ G(\mathbf{x}) := f(\mathbf{x}, \overline{\mathbf{y}}^*(\mathbf{x})) := \mathbb{E}_{\xi}[\widetilde{f}(\mathbf{x}, \overline{\mathbf{y}}^*(\mathbf{x}); \xi)] \right\}, \tag{1a}$$

$$\text{s.t. } \overline{\mathbf{y}}^*(\mathbf{x}) \in \operatorname*{arg\,min}_{\mathbf{y} \in \mathbb{R}^{d_\ell}} \left\{ h(\mathbf{x}, \mathbf{y}) \mid A\mathbf{y} \leq \mathbf{b} \right\}, \tag{1b}$$

where $\xi \sim \mathcal{D}$ represents a stochastic sample of the objective $f(\cdot, \cdot)$, $\mathcal{X} \subseteq \mathbb{R}^{d_u}$ is a convex and closed set, $f : \mathcal{X} \times \mathbb{R}^{d_\ell} \to \mathbb{R}$ is the UL objective, $h : \mathcal{X} \times \mathbb{R}^{d_\ell} \to \mathbb{R}$ is the LL objective, and $f, h$ are smooth functions. We focus on the problems where $h(\mathbf{x}, \mathbf{y})$ is strongly convex with respect to $\mathbf{y}$. The matrix $A \in \mathbb{R}^{k \times d_\ell}$, and vector $\mathbf{b} \in \mathbb{R}^k$ define the linear constraints. In the following, we refer to (1a) as the UL problem, and to (1b) as the LL one.

The success of the bilevel formulation and its algorithms in many machine learning applications can be attributed to the use of the efficient (stochastic) gradient-based methods (Liu et al., 2021a). These methods take the following form, in which an (approximate) gradient direction of the UL problem is computed (using chain rule), and then the UL variable is updated using gradient descent (GD):

$$\widehat{\nabla}G(\mathbf{x}) \approx \nabla_x f(\mathbf{x}, \overline{\mathbf{y}}^*(\mathbf{x})) + [\nabla\overline{\mathbf{y}}^*(\mathbf{x})]^T \nabla_y f(\mathbf{x}, \overline{\mathbf{y}}^*(\mathbf{x})) \texttt{ GD Update:} \mathbf{x}^+ = \mathbf{x} - \beta\widehat{\nabla}G(\mathbf{x}). \quad (2)$$

The gradient of $G(\mathbf{x})$ is often referred as the *implicit gradient*. However, computing this implicit gradient not only requires access to the optimal $\overline{\mathbf{y}}^*(\mathbf{x})$, but also assumes differentiability of the mapping $\overline{\mathbf{y}}^*(\mathbf{x}) : \mathcal{X} \to \mathbb{R}^{d_\ell}$. One can potentially solve the LL problem approximately and obtain an approximation $\widehat{\mathbf{y}}(\mathbf{x})$ such that $\widehat{\mathbf{y}}(\mathbf{x}) \approx \overline{\mathbf{y}}^*(\mathbf{x})$, and use it to compute the implicit gradient (Ghadimi & Wang, 2018). Unfortunately, not all solutions $\mathbf{y}^*(\mathbf{x})$ are differentiable, and when they are not the above approach cannot be applied.

It is known that when the LL problem is strongly convex and *unconstrained*, then $\nabla\overline{\mathbf{y}}^*(\mathbf{x})$ can be easily evaluated using the implicit function theorem (Ghadimi & Wang, 2018). This is the reason that the majority of recent works have focused on developing algorithms for the class of unconstrained bilevel problems (Ghadimi & Wang, 2018; Hong et al., 2020; Ji et al., 2021; Khanduri et al., 2021b; Chen et al., 2021a). However, when the LL problem is constrained, $\nabla\overline{\mathbf{y}}^*(\mathbf{x})$ might not even exist. In that case, most works adopt a value function-based approach to solve problems with LL constraints (Liu et al., 2021b; Sow et al., 2022; Liu et al., 2021c). Value-function-based methods typically transform the original problem into a single-level problem with non-convex and non-differentiable constraints. To resolve the latter issue these approaches regularize the problem by adding a strongly-convex penalty term, altering the problem's structure. In contrast, we introduce a perturbation-based smoothing technique, which at any given $\mathbf{x} \in \mathcal{X}$ makes $\overline{\mathbf{y}}^*(\mathbf{x})$ differentiable almost surely, without practically changing the landscape of the original problem (see (Lu et al., 2020, pg. 5)). It is important to note that the value function-based approaches are more suited for deterministic implementations, and therefore it is difficult to use such algorithms for large scale applications and/or when the data sizes are large. On the other hand, the gradient-based algorithms developed in our work can easily handle stochastic problems. Finally, there is a line of work (Amos & Kolter, 2017; Agrawal et al., 2019; Donti et al., 2017; Gould et al., 2021) about implicit differentiation in deep learning literature. However, in these works the setting (e.g. layers of neural network described by optimization tasks) and the focus (e.g., on gradient computation and implementation, rather than on algorithms and analysis) is different. For more details see Appendix A.

**Contributions.** In this work, we study a class of bilevel optimization problems with strongly convex objective and linear constraints in the LL. Major challenges for solving such problems are the following: 1) How to ensure that the implicit function $G(\mathbf{x})$ is differentiable? and 2) Even if the implicit function is differentiable, how to compute its (approximate) gradient in order to develop first-order methods? Our work addresses these challenges and develops first-order methods to tackle such constrained bilevel problems. Specifically, our contributions are the following:

– We provide an in-depth understanding of bilevel problems with strongly convex linearly constrained LL problems. Specifically, we first show with an example that the implicit objective $G(\mathbf{x})$ is in general non-differentiable. To address the non-differentiability, we propose a perturbation-based smoothing technique that makes the implicit objective $G(\mathbf{x})$ differentiable in an almost sure sense, and we provide a closed-form expression for the (approximate) implicit gradient.

– The smoothed problem we obtain is challenging, since its implicit objective does not have Lipschitz continuous gradients. Therefore, conventional gradient based algorithms may no longer work. To address this issue, we propose the Deterministic Smoothed Implicit Gradient ([D]SIGD) method that utilizes an (approximate) line search-based algorithm and establish asymptotic convergence guarantees. We also analyze [S]SIGD for the stochastic version of problem (1) (with fixed/diminishing step-sizes) and establish finite-time convergence guarantees for the cases when the implicit function is weakly-convex, strongly-convex, and convex (but not Lipschitz smooth).

– Finally, we evaluate the performance of the proposed algorithmic framework via experiments on quadratic bilevel and adversarial learning problems.

Bilevel problem 1 captures several important applications. Below we provide two such applications.

**Adversarial Training.** The problem of robustly training a model $\phi(\mathbf{x}; \mathbf{c})$, where $\mathbf{x}$ denotes the model parameters and $\mathbf{c}$ the input to the model; let $\{(\mathbf{c}_i, \mathbf{d}_i)\}_{i=1}^N$ with $\mathbf{c}_i \in \mathbb{R}^{d_{\ell_i}}, \mathbf{d}_i \in \mathbb{R}$ be the training set (Zhang et al., 2021; Goodfellow et al., 2014). It can be formulated as the following bilevel problem:

$$\min_{\mathbf{x} \in \mathbb{R}^{d_u}} \left\{ \sum_{i=1}^{N} f_i(\phi(\mathbf{x}; \mathbf{c}_i + \overline{\mathbf{y}}_i^*(\mathbf{x})), \mathbf{d}_i) \right\} \; s.t. \; \overline{\mathbf{y}}^*(\mathbf{x}) \in \begin{cases} \arg\min_{\mathbf{y}_i \in \mathbb{R}^{d_{\ell_i}}} \sum_{i=1}^{N} h_i(\phi(\mathbf{x}; \mathbf{c}_i + \mathbf{y}_i), \mathbf{d}_i) \\ s.t. \; -\mathbf{b} \leq \mathbf{y} \leq \mathbf{b} \end{cases}, \quad (3)$$

where $\mathbf{y} = [\mathbf{y}_1^T, \ldots, \mathbf{y}_N^T]^T \in \mathbb{R}^{d_\ell}$; with $\mathbf{y}_i \in \mathbb{R}^{d_{\ell_i}}$ denotes the attack on the $i^{\text{th}}$ example and we have $\sum_{i=1}^{N} d_{\ell_i} = d_\ell$. Moreover, $f_i : \mathbb{R} \times \mathbb{R} \to \mathbb{R}$ denotes the loss function for learning the model parameter $\mathbf{x}$, while $h_i : \mathbb{R} \times \mathbb{R} \to \mathbb{R}$ denotes the adversarial objective used to design the optimal attack $\mathbf{y}$. Note that the linear constraints in the LL problem $-\mathbf{b} \leq \mathbf{y} \leq \mathbf{b}$ models the attack budget.

**Distributed Optimization.** In distributed optimization (Chang et al., 2020; Yang et al., 2019), a set of $N$ agents aim to jointly minimize an objective function $G(\mathbf{x})$ over an undirected graph $\mathcal{G} = (V, E)$. We consider the following distributed bilevel problem

$$\min_{\{\mathbf{x}_i \in \mathcal{X} | A\mathbf{x} = 0\}} \left\{ G(\mathbf{x}) := \sum_{i=1}^{N} f_i(\mathbf{x}_i, \overline{\mathbf{y}}_i^*(\mathbf{x}_i)) \right\} \; s.t. \; \overline{\mathbf{y}}^*(\mathbf{x}) \in \arg\min_{\mathbf{y} \in \mathbb{R}^{d_\ell}} \left\{ \sum_{i=1}^{N} h_i(\mathbf{x}_i, \mathbf{y}_i) \; s.t. \; A\mathbf{y} = \mathbf{0} \right\},$$

where $\mathbf{x} = [\mathbf{x}_1, \ldots, \mathbf{x}_N]$ and $\mathbf{y} = [\mathbf{y}_1, \ldots, \mathbf{y}_N]$. Each agent $i \in [N]$ has access to $f_i$ and $h_i$. The constraint $A\mathbf{y} = 0$ (resp. $A\mathbf{x} = 0$) is introduced to ensure the consensus of LL (resp. UL) variables. Such problems arise in signal processing and sensor networks (Yousefian, 2021). This formulation also models a decentralized meta learning problem where the training and validation data is distributed among agents while each agent aims to solve the meta learning problem globally (Ji et al., 2021).

## 2 Properties and Implicit Gradient of Bilevel Problem (1)

### 2.1 Preliminaries

In this section we study the properties of problem 1. First, let us define the necessary notations. Let $\overline{A}(\mathbf{y})$ be the matrix that contains the rows $S(\mathbf{y}) \subseteq \{1, \ldots, k\}$ of $A$ that correspond to the active constraints of inequality $A\mathbf{y} \leq \mathbf{b}$ in the LL problem, that is we have $\overline{A}(\mathbf{y})\mathbf{y} = \overline{\mathbf{b}}(\mathbf{y})$, where $\overline{\mathbf{b}}(\mathbf{y})$ contains the elements of $\mathbf{b}$ with indices in $S(\mathbf{y})$. Also, we denote with $\overline{\boldsymbol{\lambda}}^*(\mathbf{x})$ the Lagrange multipliers vector that corresponds to the active constraints at $\overline{\mathbf{y}}^*(\mathbf{x})$ Next, we introduce some basic assumptions.

**Assumption 1.** *We assume that the following conditions hold for problem* (1)*:*

    *(a) $f(\mathbf{x}, \mathbf{y})$ is continuously differentiable, and $h(\mathbf{x}, \mathbf{y})$ is twice continuously differentiable.*

    *(b) $\mathcal{X}$ is closed and convex; $\mathcal{Y} = \left\{ \mathbf{y} \in \mathbb{R}^{d_\ell} \; \middle| \; A\mathbf{y} \leq \mathbf{b} \right\}$ is a compact set.*

    *(c) $h(\mathbf{x}, \mathbf{y})$ is strongly convex in $\mathbf{y}$, for every $\mathbf{x} \in \mathcal{X}$, with modulus $\mu_h$.*

    *(d) There exists $\mathbf{y} \in \mathbb{R}^{d_\ell}$ such that $A\mathbf{y} < \mathbf{b}$.*

    *(e) $\overline{A}(\overline{\mathbf{y}}^*(\mathbf{x}))$ is full row rank, for every $\mathbf{x} \in \mathcal{X}$[1].*

The Assumptions 1(a), (b) and (c) are standard assumptions in bilevel optimization literature and are required to ensure the continuity of the implicit function (Proposition 1). Assumption 1(c) ensures that the implicit function $G(\mathbf{x})$ is well defined as the LL problem returns a single point. Assumption 1(d) ensures strict feasibility of the LL problem, while Assumption 1(e) implies that the rows of $A$ corresponding to the active constraints are linearly independent. Note that this assumption is necessary to ensure the differentiability of the implicit function (Lemma 1,2). Also note that there are some special cases in which Assumption 1(e) is automatically satisfied. For instance, consider a problem where the LL problem has box constraints, i.e., $\mathbf{a} \leq \mathbf{y} \leq \mathbf{b}$. Then for any $\mathbf{y} \in \mathcal{Y}$ the only possible non-zero values in the matrix $\overline{A}(\mathbf{y})$ are $+1, -1$, and there is only one non-zero value at each column. Therefore, $\overline{A}(\mathbf{y})$ is full row rank. Next, we utilize the above assumptions to analyze the properties of mapping $\overline{\mathbf{y}}^*(\mathbf{x})$.

**Proposition 1** (Appendix D.1.1). *Under Assumption 1, the mapping $\overline{\mathbf{y}}^*(\mathbf{x}) : \mathcal{X} \to \mathbb{R}^{d_\ell}$ and the implicit function $G(\mathbf{x})$ are both continuous.*

Proposition 1 ensures that $\overline{\mathbf{y}}^*(\mathbf{x})$ and $G(\mathbf{x})$ are both continuous. Now if we can ensure differentiability of $\overline{\mathbf{y}}^*(\mathbf{x})$, then we should be able to implement a gradient-based update rule to solve (1). However, as the following example illustrates, $\overline{\mathbf{y}}^*(\mathbf{x})$ and thus $G(\mathbf{x})$ are not differentiable in general.

---

[1]This is the LICQ condition (Bertsekas, 1998) of the LL problem. It is used to ensure that the optimal solutions satisfy the KKT conditions.

**Example.** Consider the following problem

$$\min_{x \in [0,1]} x + \overline{y}^*(x) \text{ s.t. } \overline{y}^*(x) \in \arg\min_{y \in \mathbb{R}} \left\{ (y^2 - x^2)^2 \mid \sqrt{3/5} \leq y \leq 1 \right\}. \tag{4}$$

The mapping $\overline{y}^*(x)$ is $\overline{y}^*(x) = x$, if $x \in [\sqrt{3/5}, 1]$, and $\overline{y}^*(x) = \sqrt{3/5}$, if $x \in [0, \sqrt{3/5})$. In Figure 1, we plot this mapping. Notice that at the point $\mathbf{x} = \sqrt{3/5}$ the mapping (and thus the implicit function) is non-differentiable. ☐

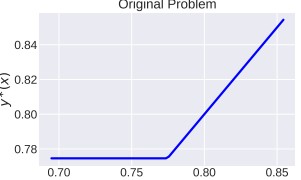

Figure 1: Plot of $\overline{y}^*(x)$.

To address the non-differentiability issue, we introduce a perturbation-based "smoothing" technique. Specifically, for any fixed $\mathbf{x} \in \mathcal{X}$, we modify the LL objective $h(\mathbf{x}, \mathbf{y})$ with the addition of the linear perturbation term $\mathbf{q}^T \mathbf{y}$, where $\mathbf{q}$ is a random vector sampled from some continuous distribution $\mathcal{Q}$. We use the following notation for the "smoothed" LL objective

$$g(\mathbf{x}, \mathbf{y}) := h(\mathbf{x}, \mathbf{y}) + \mathbf{q}^T \mathbf{y} \text{ and } \mathbf{y}^*(\mathbf{x}) := \arg\min_{\mathbf{y} \in \mathbb{R}^{d_\ell}} \{ g(\mathbf{x}, \mathbf{y}) \mid A\mathbf{y} \leq \mathbf{b} \}. \tag{5}$$

Also, we denote $F(\mathbf{x}) := f(\mathbf{x}, \mathbf{y}^*(\mathbf{x}))$ as the respective "smoothed" implicit function. Such a perturbation is used to ensure that at a given $\mathbf{x} \in \mathcal{X}$, the strict complementarity (SC) property holds for the LL problem with probability 1 (w.p. 1); see the lemma below for the formal statement.

**Lemma 1.** *((Lu et al., 2020, Proposition 1)) For a given $\mathbf{x} \in \mathcal{X}$, if $\mathbf{y}^*(\mathbf{x})$ is a KKT point of problem $\min_{\mathbf{y} \in \mathbb{R}^{d_\ell}} g(\mathbf{x}, \mathbf{y})$, $\mathbf{q}$ is generated from a continuous measure, and $\overline{A}(\mathbf{y}^*(\mathbf{x}))$ is full row rank, then the SC condition holds at $\mathbf{x}$, with probability 1 (w.p. 1).*

Combining SC ensured by Lemma 1 with Assumption 1, we can show that the implicit mapping $\mathbf{y}^*(\mathbf{x})$ is (almost surely) differentiable, which further implies that the implicit function $F(\mathbf{x})$ is differentiable at a given $\mathbf{x} \in \mathcal{X}$, and obtain a closed-form expression for the gradient (Lemma 2). We would like to stress that the properties mentioned above (i.e., SC and differentiability) are defined locally, at a given point $\mathbf{x} \in \mathcal{X}$. These properties will be used later to design algorithms that approximately optimize the original problem (1). Finally, it is worth noting that, in the absence of such a perturbation term, we would have to introduce the SC property as an assumption.

## 2.2 IMPLICIT GRADIENT

In this section, we derive a closed-form expression for the gradient of the implicit function $F(\mathbf{x})$.

**Lemma 2** (Implicit Gradient, Appendix D.1.2). *Under Assumption 1, for any given $\mathbf{x} \in \mathcal{X}$, we have*

$$\nabla \mathbf{y}^*(\mathbf{x}) = \left[ \nabla^2_{yy} g(\mathbf{x}, \mathbf{y}^*(\mathbf{x})) \right]^{-1} \left[ -\nabla^2_{xy} g(\mathbf{x}, \mathbf{y}^*(\mathbf{x})) - \overline{A}^T \nabla \overline{\boldsymbol{\lambda}}^*(\mathbf{x}) \right] \tag{6}$$

$$\nabla \overline{\boldsymbol{\lambda}}^*(\mathbf{x}) = - \left[ \overline{A} \left[ \nabla^2_{yy} g(\mathbf{x}, \mathbf{y}^*(\mathbf{x})) \right]^{-1} \overline{A}^T \right]^{-1} \left[ \overline{A} \left[ \nabla^2_{yy} g(\mathbf{x}, \mathbf{y}^*(\mathbf{x})) \right]^{-1} \nabla^2_{xy} g(\mathbf{x}, \mathbf{y}^*(\mathbf{x})) \right], \tag{7}$$

*where we set $\overline{A} := \overline{A}(\mathbf{y}^*(\mathbf{x}))$.*

Note that when LL problem (1b) does not have the LL constraints, the implicit gradient derived in Lemma 2 becomes exactly same as the one in Ghadimi & Wang (2018); Ji et al. (2021). Moreover, if the LL problem has only linear equality constraints, the differentiability of $\mathbf{y}^*(\mathbf{x})$ follows from the implicit function theorem under Assumptions 1(a) and 1(c) along with full row rankness of $A$. In fact, the expression of the implicit gradient stays the same as in Lemma 2 with $\overline{A}$ and $\overline{\boldsymbol{\lambda}}^*(\mathbf{x})$ replaced by $A$ and $\boldsymbol{\lambda}^*(\mathbf{x})$, respectively (i.e., we use the full matrix $A$). Finally, using Lemma 2 above we now have an expression of the implicit gradient as

$$\nabla F(\mathbf{x}) = \nabla_x f(\mathbf{x}, \mathbf{y}^*(\mathbf{x})) + [\nabla \mathbf{y}^*(\mathbf{x})]^T \nabla_y f(\mathbf{x}, \mathbf{y}^*(\mathbf{x})). \tag{8}$$

### 2.2.1 APPROXIMATE IMPLICIT GRADIENT

Note that computing $\nabla F(\mathbf{x})$ requires the precise knowledge of $\mathbf{y}^*(\mathbf{x})$ which is not possible for many problems of interest. Therefore, in practice we define the approximate implicit gradient as

$$\widehat{\nabla} F(\mathbf{x}) = \nabla_x f(\mathbf{x}, \widehat{\mathbf{y}}(\mathbf{x})) + [\widehat{\nabla} \mathbf{y}^*(\mathbf{x})]^T \nabla_y f(\mathbf{x}, \widehat{\mathbf{y}}(\mathbf{x})), \tag{9}$$

where $\widehat{\nabla}\mathbf{y}^*(\mathbf{x})$ is defined by setting the approximate LL solution $\widehat{\mathbf{y}}(\mathbf{x})$ in place of the exact one $\mathbf{y}^*(\mathbf{x})$ in expressions (6) and (7). In order to ensure that (9) returns a useful approximation of the (exact) implicit gradient, we impose a few assumptions on the quality of the estimate $\widehat{\mathbf{y}}(\mathbf{x})$.

**Assumption 2.** *The approximate solution of (perturbed) LL problem* (5) $\widehat{\mathbf{y}}(\mathbf{x})$ *satisfies the following* $\forall \mathbf{x} \in \mathcal{X}$:

   *(a)* $\|\widehat{\mathbf{y}}(\mathbf{x}) - \mathbf{y}^*(\mathbf{x})\| \leq \delta$ *for* $\delta > 0$,

   *(b)* $\widehat{\mathbf{y}}(\mathbf{x})$ *is a feasible point, i.e.,* $A\widehat{\mathbf{y}}(\mathbf{x}) \leq \mathbf{b}$,

   *(c)* *It holds that* $\overline{A}(\mathbf{y}^*(\mathbf{x})) = \overline{A}(\widehat{\mathbf{y}}(\mathbf{x}))$.

The LL problem requires the solution of a strongly convex linearly constrained task. As a result, Assumptions 2(a),(b) can be easily satisfied. Specifically, we can obtain approximate feasible solutions of given accuracy with known methods, such as projected gradient descent, or by using some convex optimization solver; in section B of the Appendix we provide one such method. Moreover, Assumption 2(c) will be satisfied if we find a "sufficiently accurate" solution $\widehat{\mathbf{y}}(\mathbf{x})$. Specifically, from Calamai & Moré (1987, Theorem 4.1) we know that if $\widehat{\mathbf{y}}^k(\mathbf{x}) \in \mathcal{Y}$ is an arbitrary sequence that converges to a non-degenerate (i.e., Assumption 1(e) and SC holds) stationary solution $\mathbf{y}^*(\mathbf{x})$, then there exists an integer $k_0$ such that $\overline{A}(\mathbf{y}^*(\mathbf{x})) = \overline{A}(\widehat{\mathbf{y}}^k(\mathbf{x})), \forall k > k_0$.

**Remark 1.** *There are certain special cases where we can obtain an upper bound for $k_0$. For instance, in the case of non-negative constraints $\mathbf{y} \geq \mathbf{0}$ it can be shown[2] that $\frac{L_h}{\mu_h}\log\left(2L_h\|\mathbf{y}^0 - \mathbf{y}^*(\mathbf{x})\|/\tau\right)$ iterations of the projected gradient descent method suffice to ensure that the active set of the approximate solution $\widehat{\mathbf{y}}(\mathbf{x})$ coincides with the active set of the exact one $\mathbf{y}^*(\mathbf{x})$ (see Nutini et al. (2019, Corollary 1)), where $\tau = \min_{i \in S(\mathbf{y}^*(\mathbf{x}))} \nabla_{y_i} g(\mathbf{x}, \mathbf{y}^*(\mathbf{x}))$ and $\mathbf{y}^0$ is the algorithm's initialization. A similar result can be derived for the case with bound constraints $\mathbf{a} \leq \mathbf{y} \leq \mathbf{b}$.*

Next, we introduce additional assumptions that are required to analyze the properties of (9).

**Assumption 3.** *We assume that the following holds for problem* (1), $\forall \mathbf{x}, \overline{\mathbf{x}} \in \mathcal{X}$ *and* $\mathbf{y}, \overline{\mathbf{y}} \in \mathbb{R}^{d_\ell}$:

   *(a) $f$ has bounded gradients, i.e.,* $\|\nabla f(\mathbf{x}, \mathbf{y})\| \leq \overline{L}_f$.

   *(b) $f$ has Lipschitz continuous gradients, i.e.,* $\|\nabla f(\mathbf{x}, \mathbf{y}) - \nabla f(\overline{\mathbf{x}}, \overline{\mathbf{y}})\| \leq L_f \|[\mathbf{x}; \mathbf{y}] - [\overline{\mathbf{x}}; \overline{\mathbf{y}}]\|$.

   *(c) $h$ has Lipschitz continuous gradient in $\mathbf{y}$, i.e.,* $\|\nabla_y h(\mathbf{x}, \mathbf{y}) - \nabla_y h(\mathbf{x}, \overline{\mathbf{y}})\| \leq L_h \|\mathbf{y} - \overline{\mathbf{y}}\|$.

   *(d) $h$ has Lipschitz continuous Hessian in $\mathbf{y}$, i.e.,* $\|\nabla_{yy}^2 h(\mathbf{x}, \mathbf{y}) - \nabla_{yy}^2 h(\mathbf{x}, \overline{\mathbf{y}})\| \leq L_{h_{yy}} \|\mathbf{y} - \overline{\mathbf{y}}\|$.

   *(e) $h$ has Lipschitz continuous Jacobian, i.e.,* $\|\nabla_{xy}^2 h(\mathbf{x}, \mathbf{y}) - \nabla_{xy}^2 h(\mathbf{x}, \overline{\mathbf{y}})\| \leq L_{h_{xy}} \|\mathbf{y} - \overline{\mathbf{y}}\|$.

   *(f) $h$ has a bounded Jacobian,* $\|\nabla_{xy}^2 h(\mathbf{x}, \mathbf{y})\| \leq \overline{L}_{h_{xy}}$.

Assumption 3 is standard in bilevel optimization literature (Ghadimi & Wang, 2018; Hong et al., 2020; Chen et al., 2021a; Ji et al., 2021) and is used to derive some useful properties of the (approximate) implicit gradient (Lemma 3, Appendix D.1.3). It is easy to see that Assumptions 1(a),(c) and 3 hold directly for the perturbed objective (5) with constants $\mu_g = \mu_h$, $L_h = L_g$, $L_{g_{yy}} = L_{h_{yy}}$, $L_{g_{xy}} = L_{h_{xy}}$, $\overline{L}_{g_{xy}} = \overline{L}_{h_{xy}}$; we also assume that Assumption 1(e) holds for the perturbed LL problem (5).

**Lemma 3** (Appendix D.1.3). *Suppose that Assumptions 1,2,3 hold. Then, for every $\mathbf{x} \in \mathcal{X}$ the following holds*

$$\|\widehat{\nabla}F(\mathbf{x}) - \nabla F(\mathbf{x})\| \leq L_F \cdot \delta, \quad \|\nabla F(\mathbf{x})\| \leq \overline{L}_F \quad and \quad \|\widehat{\nabla}F(\mathbf{x})\| \leq \overline{L}_F,$$

*where $L_F = L_f + L_{\mathbf{y}^*}\overline{L}_f + L_f \overline{L}_{\mathbf{y}^*}$, and $\overline{L}_F = \left(1 + \overline{L}_{\mathbf{y}^*}\right)\overline{L}_f$; the constants $\overline{L}_{\mathbf{y}^*}, L_{\mathbf{y}^*}$ are defined in Lemmas 7,9, respectively, provided in the Appendix.*

### 2.2.2 STOCHASTIC IMPLICIT GRADIENT

In the stochastic setting, the (approximate) stochastic implicit gradient is computed as:

$$\widehat{\nabla}F(\mathbf{x}; \xi) = \nabla_x \widetilde{f}(\mathbf{x}, \widehat{\mathbf{y}}(\mathbf{x}); \xi) + [\widehat{\nabla}\mathbf{y}^*(\mathbf{x})]^T \nabla_y \widetilde{f}(\mathbf{x}, \widehat{\mathbf{y}}(\mathbf{x}); \xi). \quad (10)$$

Also, we make the following assumption on the stochastic gradients of the UL problem.

**Assumption 4.** *We assume that the stochastic gradients are unbiased, i.e.* $\mathbb{E}_\xi[\nabla \widetilde{f}(\mathbf{x}, \mathbf{y}; \xi)] = \nabla f(\mathbf{x}, \mathbf{y})$ *and have bounded variance, i.e.,* $\mathbb{E}_\xi\|\nabla \widetilde{f}(\mathbf{x}, \mathbf{y}; \xi) - \nabla f(\mathbf{x}, \mathbf{y})\|^2 = \sigma_f^2$ *for some* $\sigma_f > 0$.

---

[2]Under the assumption that $\nabla_{y_i} g(\mathbf{x}, \mathbf{y}^*(\mathbf{x})) > 0, \forall i \in S(\mathbf{y}^*(\mathbf{x}))$.

---

**Algorithm 1** [Deterministic] Smoothed Implicit Gradient Descent (**[D]SIGD**)

---

1: **Input:** Initial parameter $\mathbf{x}^0$, # of iteration $T$, LL solution accuracy, $\delta^r$, $\sigma$, measure $\mathcal{Q}$, s
2: Sample $\mathbf{q} \sim \mathcal{Q}$ and perturb LL problem
3: **for** $r = 0, 1, \ldots, T - 1$ **do**
4:     Find an approximate solution $\widehat{\mathbf{y}}(\mathbf{x}^r)$ s.t. Assumption 2 is satisfied.
5:     Compute $\widehat{\nabla}F(\mathbf{x}^r)$ using (9), $\widehat{\mathbf{d}}^r = \widetilde{\mathbf{x}}^r - \mathbf{x}^r$ with $\widetilde{\mathbf{x}}^r = \text{proj}_\mathcal{X}(\mathbf{x}^r - s\widehat{\nabla}F(\mathbf{x}^r))$
6:     Select $a^r$ s.t. the following Armijo-type rule condition is satisfied

$$\widehat{F}(\mathbf{x}^r) - \widehat{F}(\mathbf{x}^r + a^r\widehat{\mathbf{d}}^r) \geq -\sigma \cdot a^r[\widehat{\nabla}F(\mathbf{x}^r)]^T\widehat{\mathbf{d}}^r - \epsilon(\delta; r) \tag{11}$$

    where $\epsilon(\delta; r)$ depends on $\delta^r$, $\alpha^r$ and problem-dependent parameters; $\widehat{F}(\cdot) = f(\cdot, \widehat{\mathbf{y}}(\cdot))$.
7:     Perform one projected gradient step: $\mathbf{x}^{r+1} = \mathbf{x}^r + a^r \cdot \widehat{\mathbf{d}}^r$
8: **end for**

---

**Algorithm 2** [Stochastic] Smoothed Implicit Gradient Descent (**[S]SIGD**)

---

1: **Input:** Initial parameter $\mathbf{x}^0$, # of iterations $T$, step-sizes $\{\beta^r\}_{r=0}^{T-1}$, LL solution accuracy $\delta$
2: Sample $\mathbf{q} \sim \mathcal{Q}$ and perturb LL problem
3: **for** $r = 0, 1, \ldots, T - 1$ **do**
4:     Find an approximate solution $\widehat{\mathbf{y}}(\mathbf{x}^r)$ s.t. Assumption 2 is satisfied.
5:     Compute $\widehat{\nabla}F(\mathbf{x}^r; \xi^r)$ using (10)
6:     Perform one stochastic projected gradient descent step: $\mathbf{x}^{r+1} = \text{proj}_\mathcal{X}(\mathbf{x}^r - \beta^r\widehat{\nabla}F(\mathbf{x}^r; \xi^r))$
7: **end for**

---

Assumption 4 is a typical assumption required to ensure that the approximate implicit stochastic gradient is also unbiased and has finite variance (Ghadimi & Wang, 2018; Hong et al., 2020; Chen et al., 2021a) as shown in Lemma 4 below.

**Lemma 4** (Appendix D.1.4). *Under Assumptions 1,2,3 and 4, the stochastic gradient estimate in (10) is unbiased, i.e., $\mathbb{E}_\xi[\widehat{\nabla}F(\mathbf{x}; \xi)] = \widehat{\nabla}F(\mathbf{x})$ and has bounded variance , i.e., $\mathbb{E}_\xi\|\widehat{\nabla}F(\mathbf{x}; \xi) - \widehat{\nabla}F(\mathbf{x})\| \leq \sigma_F^2$ where $\sigma_F^2 = 2\sigma_f^2 + 2\overline{L}_{\mathbf{y}^*}\sigma_f^2$; where $\overline{L}_{\mathbf{y}^*}$ is defined in Lemma 7 in the Appendix.*

## 3   THE SIGD ALGORITHMS AND CONVERGENCE ANALYSIS

### 3.1   THE PROPOSED ALGORITHMS

In this section, we develop gradient-based methods for solving problem (1) by leveraging the smoothing based technique introduced in the previous section. Recall that for any $\mathbf{x} \in \mathcal{X}$, we can introduce a perturbation to make the optimal solution $\mathbf{y}^*(\mathbf{x})$ of the perturbed LL problem differentiable. Next, to proceed with the algorithm design, there are two options available. First, generate a perturbation for each $\mathbf{x} \in \mathcal{X}$ encountered in the algorithm. Second, generate *a single* perturbation at the beginning of the algorithm, and use it throughout the execution of the algorithm. It is worth mentioning that, both approaches perform equally well in our numerical experiments. However, for the ease of analysis, we adopt the second approach. To justify such an approach, below we show that, just sampling a single $\mathbf{q}$ is suffice to make $F(\mathbf{x})$ differentiable (almost surely) at a sequence of countable points.

**Lemma 5** (Appendix D.2.1). *Let $\{\mathbf{x}^r\}_{r=0}^\infty \in \mathcal{X}$ be an arbitrary sequence. Consider the implicit function $F(\mathbf{x}) := f(\mathbf{x}, \mathbf{y}^*(\mathbf{x}))$, where $\mathbf{y}^*(\mathbf{x})$ is defined in (5), where a single perturbation $\mathbf{q}$ is used. Then $F(\cdot)$ is differentiable at all the points $\{\mathbf{x}^r\}_{r=0}^\infty$ w.p. 1.*

Due to this result, in the following analysis, we assume that the almost sure differentiability will also be satisfied for the iterates generated by our algorithms as suggested by Lemma 5. Further, our algorithm design is guided by the fact that, unlike bilevel programs with unconstrained LL tasks (see Lemma 2.2(c) in Ghadimi & Wang (2018)), the implicit gradient $\nabla F(\mathbf{x})$ in (8) is not Lipschitz smooth in general. This implies that algorithms that provably converge only under the Lipschitz assumption, will not work in our case, particularly when the implicit function is non-convex. Towards this end, we propose the [Deterministic] Smoothed Implicit Gradient Descent ([D]SIGD) method

(Alg. 1), a determinstic line-search-based method, which does not require Lipschitz smoothness or another special structure (e.g., convexity), and show asymptotic convergence (Theorem 1). Moreover, for the cases where the implicit function is weakly-convex, convex or strongly convex (but still not Lipschitz smooth) the [Stochastic] Smoothed Implicit Gradient Descent ([S]SIGD) method (Alg. 2) is developed, a stochastic gradient-based method, for which finite-time convergence guarantees are derived (Theorem 2,3,4).

## 3.2 Convergence Analysis

As discussed above, in the context of algorithm design and analysis we sample a single perturbation $\mathbf{q}$, and keep it fixed during the algorithm execution. As a result, the algorithm is effectively optimizing the following smooth *surrogate* of the original problem (1):

$$\min_{\mathbf{x} \in \mathcal{X}} \left\{ F(\mathbf{x}) = f(\mathbf{x}, \mathbf{y}^*(\mathbf{x})) = \mathbb{E}_\xi[\widetilde{f}(\mathbf{x}, \mathbf{y}^*(\mathbf{x}); \xi)] \right\} \tag{12a}$$

$$\text{s.t. } \mathbf{y}^*(\mathbf{x}) \in \arg\min_{\mathbf{y} \in \mathbb{R}^{d_\ell}} \left\{ g(\mathbf{x}, \mathbf{y}) = h(\mathbf{x}, \mathbf{y}) + \mathbf{q}^T\mathbf{y} \mid A\mathbf{y} \leq \mathbf{b} \right\}, \tag{12b}$$

where $\mathbf{q} \in \mathbb{R}^{d_\ell}$ is generated from a continuous measure only once and thus is considered fixed. Next, we show that the original problem (1) and the smoothed *surrogate* problem (12) are "close". Specifically, we show below that the original implicit function $G(\mathbf{x})$ and the "smoothed" implicit function $F(\mathbf{x})$ differ by a quantity that is controlled by the size of the perturbation vector $\mathbf{q}$.

**Proposition 2** (Appendix D.2.2). *Under Ass. 1 and 3, we have:* $|G(\mathbf{x}) - F(\mathbf{x})| \leq \overline{L}_f \frac{\|\mathbf{q}\|}{\mu_g}, \forall \mathbf{x} \in \mathcal{X}$.

Note that the only requirement on $\mathbf{q}$ is that it is generated from a continuous measure. Therefore we can always choose a distribution such that $\|\mathbf{q}\|$ is arbitrarily small. Next, let us analyze Alg. 1. We have the following asymptotic result.

**Theorem 1** (Appendix D.2.3). *Suppose that Ass. 1, 3 hold. At each iteration $r$ of Alg. 1 we find $0 < a^r < 1$ such that the Armijo-type condition* (11) *is satisfied with $\epsilon(\delta; r) = L_f\delta^r + \overline{L}_F L_F a^r \delta^r + L_f \delta^{r+1} + L_F^2 \sigma a^r (\delta^r)^2 + 2L_F \overline{L}_F \sigma a^r \delta^r$. Further, we select $\delta^r$ such that Ass. 2 is satisfied, $\lim_{r \to \infty} \delta^r = 0$, and it holds that $\delta^r/a^r \sim \mathcal{O}(c^r)$, where $c^r$ is some sequence with $\lim_{r \to \infty} c^r = 0$. In addition, the sequence $\widehat{\mathbf{d}}^r$ is selected such that it is gradient related to $\widehat{\nabla}F(\mathbf{x}^r)$, i.e., "for any subsequence $\{\mathbf{x}^r\}_{r \in \mathcal{R}}$ converging to a non-stationary point, the corresponding subsequence $\{\widehat{\mathbf{d}}^r\}_{r \in \mathcal{R}}$ is bounded and satisfies $\limsup_{r \to \infty, r \in \mathcal{R}} \left[\widehat{\nabla}F(\mathbf{x}^r)\right]^T \widehat{\mathbf{d}}^r < 0$" (Bertsekas, 1998, eq. 1.13). Then w.p. 1 the limit point $\bar{\mathbf{x}}$ of the sequence of iterates generated by the [D]SIGD Alg. 1 is a stationary point.*

Note that in Theorem 1 only asymptotic convergence is guaranteed. However, this is the best we can do since we do not impose any Lipschitz smoothness or convexity assumptions. On the other hand, in the special cases where the implicit function is weakly-convex, strongly convex or convex (but still not Lipschitz smooth), it is possible to derive finite-time convergence guarantees as presented next. Towards this end, we need to impose the additional assumption that the set $\mathcal{X}$ is bounded; combining this property with Assumption 1 implies that $\mathcal{X}$ is compact. So, in the following results we assume that $\mathcal{X}$ is a compact set with diameter $\mathcal{D}_\mathcal{X} := \sup_{\mathbf{x}, \bar{\mathbf{x}} \in \mathcal{X}} \|\mathbf{x} - \bar{\mathbf{x}}\|$.

**Weakly Convex Objective.** We make the following assumption on the implicit function $F(\cdot)$.

**Assumption 5.** *We assume that for some $\rho > 0$ the implicit function $F(\mathbf{x})$ satisfies: $F(\mathbf{z}) \geq F(\mathbf{x}) + \langle \nabla F(\mathbf{x}), \mathbf{z} - \mathbf{x} \rangle - \frac{\rho}{2}\|\mathbf{z} - \mathbf{x}\|^2 \ \forall \mathbf{x}, \mathbf{z} \in \mathbb{R}^{d_u}$.*

Assumption 5 implies that the function $F(\mathbf{x}) + \frac{\hat{\rho}}{2}\|\mathbf{x}\|^2$ for $\hat{\rho} = \rho$ is convex while for $\hat{\rho} > \rho$ is strongly convex with modulus $\hat{\rho} - \rho$. Many problems of practical interest satisfy the weak-convexity, for example, phase retrieval (Davis et al., 2020), covariance matrix estimation (Chen et al., 2015), dictionary learning (Davis & Drusvyatskiy, 2019), Robust PCA (Candès et al., 2011) etc. (please see (Davis & Drusvyatskiy, 2019) and (Drusvyatskiy, 2017) for more details). For providing guarantees for the [S]SIGD algorithm we utilize a Moreau envelope based analysis. For this purpose, we first rephrase the UL problem as an unconstrained one: $\min_{\mathbf{x} \in \mathbb{R}^{d_u}} H(\mathbf{x}) := F(\mathbf{x}) + \mathbf{I}_\mathcal{X}(\mathbf{x})$, where $\mathbf{I}_\mathcal{X}(\mathbf{x})$ is the indicator function of set $\mathcal{X}$ defined as: $\mathbf{I}_\mathcal{X}(\mathbf{x}) := 0$ if $\mathbf{x} \in \mathcal{X}$ and $\mathbf{I}_\mathcal{X}(\mathbf{x}) := \infty$ if $\mathbf{x} \notin \mathcal{X}$. Below we define the Moreau envelope of $H(\mathbf{x})$.

**Definition 1.** *Given $\lambda > 0$, the Moreau envelope of $H(\mathbf{x})$ is defined as*

$$H_\lambda(\mathbf{x}) := \min_{\mathbf{z} \in \mathbb{R}^{d_u}} \left\{ H(\mathbf{z}) + \frac{1}{2\lambda} \|\mathbf{x} - \mathbf{z}\|^2 \right\} = \min_{\mathbf{z} \in \mathcal{X}} \left\{ F(\mathbf{z}) + \frac{1}{2\lambda} \|\mathbf{x} - \mathbf{z}\|^2 \right\},$$

*where the second equality follows from the definition of $H(\mathbf{x})$. Moreover, we denote the proximal map of $H(\mathbf{x})$ as $\hat{\mathbf{x}} := prox_{\lambda H}(x)$ which is defined as*

$$\hat{\mathbf{x}} := \arg\min_{\mathbf{z} \in \mathbb{R}^{d_u}} \left\{ H(\mathbf{z}) + \frac{1}{2\lambda} \|\mathbf{x} - \mathbf{z}\|^2 \right\} = \arg\min_{\mathbf{z} \in \mathcal{X}} \left\{ F(\mathbf{z}) + \frac{1}{2\lambda} \|\mathbf{x} - \mathbf{z}\|^2 \right\}.$$

The norm of the gradient of the Moreau envelope satisfies the following:

$$\|\mathbf{x} - \hat{\mathbf{x}}\| = \lambda \|\nabla H_\lambda(\mathbf{x})\|, \quad H(\hat{\mathbf{x}}) \leq H(\mathbf{x}), \quad \text{and} \quad \text{dist}(0; \partial H(\hat{\mathbf{x}})) \leq \|\nabla H_\lambda(\mathbf{x})\|, \quad (13)$$

where $\text{dist}(0; \partial H(\hat{\mathbf{x}})) = -\inf_{\mathbf{v}:\|\mathbf{v}\| \leq 1} H'(\mathbf{x}; \mathbf{v})$ and $H'(\mathbf{x}; \mathbf{v})$ denotes the directional derivative of $H$ at $\mathbf{x}$ in direction $\mathbf{v}$. Note that a small gradient $\|\nabla H_\lambda(\mathbf{x})\|$ implies that $\mathbf{x}$ is near some point $\hat{\mathbf{x}}$ that is nearly stationary (Davis & Drusvyatskiy, 2019). Then we have the following result.

**Theorem 2** (Appendix D.2.4). *Under Ass. 1, 2, 3, 4 and 5, with step-sizes $\beta^r = \beta$ for all $r \in \{0, \ldots, T-1\}$ and for any constant $\hat{\rho} > \frac{3\rho}{2}$, the iterates generated by Algorithm 2 satisfy (w.p. 1)*

$$\frac{1}{T} \sum_{r=0}^{T-1} \mathbb{E}\|\nabla H_{1/\hat{\rho}}(\mathbf{x}^r)\|^2 \leq \frac{2\hat{\rho}}{2\hat{\rho} - 3\rho} \left[ \frac{H_{1/\hat{\rho}}(\mathbf{x}^0) - H^*}{\beta T} + \beta \hat{\rho} (\sigma_F^2 + \overline{L}_F^2) + \frac{\hat{\rho}}{2\rho} L_F^2 \delta^2 \right].$$

Theorem 2 implies that with the choice of $\beta = \mathcal{O}(1/\sqrt{T})$, the [S]SIGD algorithm converges to a stationary point at a rate of $\mathcal{O}(1/\sqrt{T})$ with an additive error determined by the accuracy of the LL problem's solution $\delta$ (see Assumption 2).

**Strongly Convex and Convex Objective.** Next, we provide the guarantees for the case when the implicit function is strongly convex. We make the following assumption.

**Assumption 6.** *We assume that the objective $F(\mathbf{x})$ is $\mu_F$-strongly convex, i.e., $F(\mathbf{z}) \geq F(\mathbf{x}) + \langle \nabla F(\mathbf{x}), \mathbf{z} - \mathbf{x} \rangle + \frac{\mu_F}{2} \|\mathbf{x} - \mathbf{z}\|^2 \, \forall \mathbf{z}, \mathbf{x} \in \mathcal{X}$. Note that for $\mu_F = 0$, the objective becomes convex.*

**Theorem 3** (Appendix D.2.5). *Under the Assumptions 1, 2, 3, 4 and 6, with $\mu_F > 0$ and the choice of step-sizes $\beta^r = \frac{1}{\mu_F(r+1)}$ the iterates generated by Algorithm 2 satisfy the following (w.p. 1),*

$$\mathbb{E}[F(\underline{\mathbf{x}}) - F^*] \leq \frac{(\sigma_F^2 + \overline{L}_F^2)}{\mu_F} \frac{\log(T)}{T} + D_{\mathcal{X}} L_F \delta.$$

**Theorem 4** (Appendix D.2.6). *Under Assumption 1, 2, 3, 4 and 6, with $\mu_F = 0$, and step-sizes $\beta^r = \beta$ for $r \in \{0, \ldots, T-1\}$, the iterates generated by Algorithm 2 satisfy the following (w.p. 1),*

$$\mathbb{E}[F(\underline{\mathbf{x}}) - F^*] \leq \frac{\|\mathbf{x}^1 - \mathbf{x}^*\|^2}{\beta T} + 2\beta(\sigma_F^2 + \overline{L}_F^2) + D_{\mathcal{X}} L_F \delta, \quad \text{where } \underline{\mathbf{x}} = \frac{1}{T} \sum_{r=0}^{T-1} \mathbf{x}^r.$$

The results of Theorems 3 and 4 imply that the implicit function $F(\mathbf{x})$ converges to the optimal value at a rate of $\mathcal{O}(1/T)$ for strongly-convex objectives with diminishing step-sizes, and at a rate of $\mathcal{O}(1/\sqrt{T})$ for convex objectives with $\beta = \mathcal{O}(1/\sqrt{T})$. Note that convergence is shown to a neighborhood of the optimal solution where its size is determined by the size of the LL error $\delta$.

## 4 EXPERIMENTS

In this section, we evaluate the performance of Algorithms 1 and 2 via numerical experiments. First, we compare the performance of [D]SIGD to the recently proposed PDBO (Sow et al., 2022) for constrained bilevel optimization on a quadratic bilevel problem. Then in the second set of experiments, we evaluate the performance of [S]SIGD against popular adversarial training algorithms.

**Quadratic Bilevel Optimization.** Consider the quadratic bilevel problem of the form (1) with

$$f(\mathbf{x}, \mathbf{y}) = \frac{1}{4} \|\mathbf{x}\|^2 + 10\mathbf{x}^T \mathbf{y} - \frac{1}{4} \|\mathbf{y}\|^2 + \mathbf{1}^T \mathbf{x} + \mathbf{1}^T \mathbf{y} + 1 \text{ and } h(\mathbf{x}, \mathbf{y}) = \mathbf{x}^T \mathbf{y} + \frac{1}{2} \|\mathbf{y}\|^2 + x_1 + y_2, \quad (14)$$

and linear constraints of the form $|y_i| \leq 1, i \in \{1, 2\}$. Here, $\mathbf{x} = [x_1, x_2]^T$, $\mathbf{y} = [y_1, y_2]^T$ with $x_i, y_i \in \mathbb{R}$ for $i \in \{1, 2\}$, and $\mathbf{1} = [1, 1]^T$ . The evaluation criterion is the stationarity gap $\|\nabla F(\mathbf{x})\|$.

On this problem we execute [D]SIGD (Algorithm 1), [S]SIGD (Algorithm 2), and PDBO (Sow et al., 2022). In the first two cases, we solve the inner-level problem using 10 steps of projected gradient descent with stepsize $10^{-1}$. For the stepsize of [S]SIGD, we choose $\beta = 0.1$, while in [D]SIGD we find the proper Armijo step-size by successively adapting (by increasing $m$) the quantity

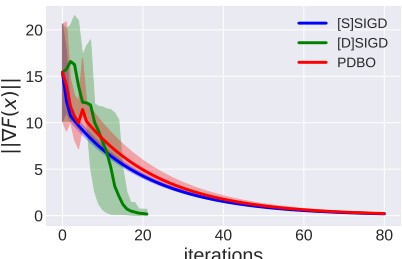

Figure 2: $\|\nabla F(\mathbf{x})\|$ vs # of iterations.

$a_r = (0.9)^m$ until condition (11) is met. In PDBO we select $10^{-1}$ for the stepsizes of both the primal and dual steps, and the number of inner iterations is set to 10. In Figure 2, we plot the convergence curves for the three algorithms with respect to number of iterations; the results are averaged over 10 runs. Note that the line search [D]SIGD method outperforms the fixed step-size [S]SIGD and PDBO while [S]SIGD performs similar to PDBO.

**Adversarial Learning.** We consider an adversarial learning problem of the form given in (3). For the perturbation we focus on the $\epsilon$-tolerant $\ell_\infty$-norm attack constraint, namely $\mathcal{Y} = \{\mathbf{y} \in \mathbb{R}^{d_\ell} \mid \|\mathbf{y}\|_\infty \leq \epsilon\}$, which can easily be expressed as a linear inequality constraint as in the LL problem of (3). We consider two widely accepted adversarial learning method as our baselines, namely AT (Madry et al., 2017) and TRADES (Zhang et al., 2019b). Also, we consider two representative datasets CIFAR-10/100 (Krizhevsky et al., 2009) and adopt the ResNet-18 (He et al., 2016) model; the results for CIFAR-10 are provided in Appendix C. In particular, we studied two widely used,(Madry et al., 2017; Wong et al., 2020) attack budget choices $\epsilon \in \{8/255, 16/255\}$. In the implementation of our [S]SIGD method, we adopt a perturbation generated by a Gaussian random vector $\mathbf{q}$ with variances from the following list $\sigma^2 \in \{2\mathrm{e}{-5}, 4\mathrm{e}{-5}, 6\mathrm{e}{-5}, 8\mathrm{e}{-5}, 1\mathrm{e}{-4}, \}$, in order to study different levels of smoothness. Moreover, for solving the LL problem in each iteration we select a fixed batch of samples. We choose $f_i$ to be cross-entropy loss and $h_i = -f_i + \lambda\|\mathbf{y}_i\|^2$ for hyper-parameter $\lambda > 0$. For [S]SIGD, we follow the implementation of (Zhang et al., 2021) but with perturbations in the LL problem. We evaluate the robustly trained model with two metrics, namely the standard accuracy (SA) and robust accuracy (RA), where we evaluate the accuracy of the robustified model on the clean and attacked test set, respectively; the attacked set is generated using PGD-50-10 (Madry et al., 2017) (i.e., 50-step PGD attack with 10 restarts). Desirably, a well trained model possesses high RA while maintaining simultaneously the SA at a high level. Table 1 shows the performance overview of our experiments. We make the following observations. First, a low level of perturbation variance (*e.g.*, $\sigma^2 \in \{2\mathrm{e}{-5}, 4\mathrm{e}{-5}\}$) in general improves both SA as well as RA, which presents an enhanced RA-SA trade-off. For example, in the setting (CIFAR-100, $\epsilon = 16/255$), our algorithm boosts the RA by over 0.3% and the SA by 2%. Second, a high level of perturbation variance harms the robustness but results in high SA. This is reasonable, since the stochastic gradient becomes too noisy with large variances. Third, our method outperforms AT and closely matches the performance of the stronger baseline TRADES. However, we would like to stress that the intent of our work is not to design a specialized adversarial learning method, and thus robustness gap between our method and the strong baseline does not diminish the value of our method. Additional details are provided in Appendix C.2.

Table 1: Performance overview of different methods on CIFAR-100 Krizhevsky et al. (2009) with ResNet-18 He et al. (2016). The result $a_{\pm b}$ represents the mean value $a$ with a standard deviation of $b$ over 5 random trials.

| | | | | | | | |
|---|---|---|---|---|---|---|---|
| **CIFAR-100, $\epsilon = 8/255$** | | | | | | | |
| Metrics | AT | TRADES | [S]SIGD (Gaussian variance $\sigma^2$) | | | | |
| | | | 2e−5 | 4e−5 | 6e−5 | 8e−5 | 1e−4 |
| SA | $53.83_{\pm 0.19}$ | $53.33_{\pm 0.18}$ | $53.88_{\pm 0.22}$ | $54.01_{\pm 0.24}$ | $53.79_{\pm 0.14}$ | $54.44_{\pm 0.18}$ | $57.74_{\pm 0.22}$ |
| RA | $27.36_{\pm 0.24}$ | $28.44_{\pm 0.17}$ | $27.43_{\pm 0.12}$ | $28.22_{\pm 0.10}$ | $28.12_{\pm 0.14}$ | $27.14_{\pm 0.21}$ | $25.22_{\pm 0.15}$ |
| **$\epsilon = 16/255$** | | | | | | | |
| SA | $42.06_{\pm 0.17}$ | $42.19_{\pm 0.23}$ | $44.06_{\pm 0.19}$ | $45.66_{\pm 0.25}$ | $46.57_{\pm 0.22}$ | $47.11_{\pm 0.32}$ | $47.46_{\pm 0.44}$ |
| RA | $15.10_{\pm 0.28}$ | $16.59_{\pm 0.26}$ | $15.51_{\pm 0.17}$ | $14.18_{\pm 0.22}$ | $13.92_{\pm 0.25}$ | $13.54_{\pm 0.18}$ | $13.42_{\pm 0.26}$ |

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

# A    RELATED LITERATURE

Bilevel optimization problems, initially encountered in the context of Stackelberg (leader-follower) games (Von Stackelberg & Von, 1952), find applications in a multitude of areas, including machine learning (Liu et al., 2021a), economics (Mirrlees, 1999), power systems (Abedi et al., 2021; Arias et al., 2008), chemical industry (Raghunathan & Biegler, 2003), transport research (Didi-Biha et al., 2006; Kalashnikov et al., 2010); see (Colson et al., 2005; Dempe & Zemkoho, 2020; Sinha et al., 2017; Liu et al., 2021a) for a number of survey papers. The "classical" approaches for solving bilevel problems include the use of approximate descent methods (Shaban et al., 2019; Ghadimi & Wang, 2018; Franceschi et al., 2017), penalty methods (Lin et al., 2014), KKT reformulations-based approaches (Allende & Still, 2013), value function-based methods (Ye & Zhu, 1995; Sow et al., 2022), and trust-region algorithms (Marcotte et al., 2001). In addition, bilevel problems are known to be related to mathematical programs with equilibrium constraints (MPEC) (Luo et al., 1996).

Recently, motivated by machine learning applications, gradient-based approaches have gained popularity for solving bilevel optimization problems (Liu et al., 2021a), e.g., in hyperparameter optimization (Shaban et al., 2019; Franceschi et al., 2017; 2018), and meta learning (Rajeswaran et al., 2019; Franceschi et al., 2018). The majority of those works are focused on solving bilevel problems with unconstrained strongly convex LL problem, for both stochastic and deterministic objectives (Ghadimi & Wang, 2018; Hong et al., 2020; Khanduri et al., 2021a;b; Chen et al., 2021a; Ji et al., 2021; Chen et al., 2021b; Yang et al., 2021). An attractive property of such problems is the existence and easy computability of the implicit gradient. Moreover, under mild assumptions, the implicit gradient for these problems can be shown to be Lipschitz smooth (e.g., see (Ghadimi & Wang, 2018, Lemma 2.2) and (Khanduri et al., 2021b, Lemma 3.1)). In contrast, for bilevel problems with linear LL constraints the implicit gradient in general might not exist, and even if it exists computing it in closed-form is a challenging task. As discussed earlier, we develop a perturbation-based smoothing framework for the constrained LL problem that ensures the existence of the implicit gradient in an almost sure sense, and allows us to compute an expression for the implicit gradient.

In Liu et al. (2021c) and Sow et al. (2022) the authors have considered bilevel optimization with (general) constraints in the LL problem. Both papers develop a value function-based framework that leads to a single level problem with non-convex constraints. In Liu et al. (2021c) a sequential minimization approach is followed where the value-function and the LL constraints are incorporated into the objective using penalty or barrier functions. In Sow et al. (2022) a primal-dual-based framework is proposed in which the problem is regularized with the addition of a strongly-convex penalty term, while a constant error term is added to make the constraint set strictly feasible. In contrast, our approach relies only on a small linear perturbation which can be made arbitrarily small without practically changing the landscape of the LL problem.

There is also a line of works (Amos & Kolter, 2017; Agrawal et al., 2019; Donti et al., 2017; Gould et al., 2021) about implicit differentiation in deep learning literature. These works Deep-Learning-type (DL-type) are indeed related to ours, in the sense that at the core of both of them lies the computation of the gradient/Jacobian of the solution of an optimization problem. However, there are some key differences. First, in our work we consider a constrained bilevel optimization problem and we are interested in analyzing this problem from an optimization perspective. On the other hand, in the DL-type works the optimization problems that are studied describe the input-output relationships of neural networks layers and the main focus lies in deriving Jacobians for the backward pass. Secondly, in our work we study a special bilevel problem (the constraints are linear) and derive a closed form expression for the implicit gradient. On the contrary, in the DL-type works the underlying problems have more general constraints and the Jacobian is usually computed using numerical methods (e.g., solving iteratively a system of KKT equations), rather than analytically. Finally, in our work the focus is on studying the properties of the bilevel problem (e.g. differentiability, approximation errors), developing (deterministic and stochastic) algorithms, and performing a convergence analysis. On the other hand, DL-type works focus mainly on the Jacobian computation and its implementation.

Finally, there is a number of works on implicit differentiation on non-smooth problems (Mairal et al., 2011; Bertrand et al., 2021; 2020). However, these works typically deal with special (non-smooth) LL problems, e.g., in Mairal et al. (2011); Bertrand et al. (2020) the non-smooth term in the LL is the $\ell 1$-norm, and in Bertrand et al. (2021) the non-smooth term is separable. On the contrary, in our work we are considering smooth LL problems and general linear inequality constraints.

# B  SOLUTION METHODS FOR THE LL PROBLEM

The LL problem is a strongly convex linearly constrained optimization task. As a result, there exist many efficient ways to find its solutions. In order to discuss about them, we consider two different classes of problems depending on the exact form of the linear constraints and the difficulty of computing the respective projection operator: 1) the projection has a closed-form solution, 2) the projection requires the solution of an optimization problem. Before we proceed, we would like to stress that the problem we are solving, i.e., the bilevel problem with linear constraints in the LL, is a very challenging one, regardless of the specific form and the exact way we approach the solution of the LL problem.

In the first class of problems, where the projection can be computed in closed form, we have problems with special linear constraints. One characteristic example is box constraints, i.e. constraints of the form $\mathbf{a} \leq \mathbf{y} \leq \mathbf{b}$, where the inequalities apply in a component-wise manner. These constraints appear in applications, such as adversarial learning (see the motivating applications in the main text). In this case, we can use some first-order iterative algorithm to solve the LL problem and project each iterate onto the constraint set using the closed-form expression (which only incurs a constant cost per iteration). For instance, we can use the projected gradient descent method which probably converges to the optimal solution with a linear rate.

In the second class of problems, the projection operator does not possess a closed-form expression. In this case we can approach the LL problem as a convex optimization task, and solve it using some convex optimization solver (e.g. employing interior-point methods) to obtain a highly accurate solution with a complexity of $\mathcal{O}\left(p(d_\ell, k) \log(d_\ell/\epsilon)\right)$, where $p(\cdot)$ is some polynomial and $\epsilon$ is solution accuracy. Alternatively, as mentioned in the previous case, we can use a projected gradient descent-type method that enjoys a linear convergence rate guarantee. Differently from the previous case though the projection operator computed at each iteration requires the solution of an optimization problem. Nonetheless, the projection task we are referring to is a (strongly convex) quadratic linearly constrained problem, that is a special quadratic programming task, which is easy to solve in practice. In algorithm 3 we describe the solution of the LL using a projected gradient descent algorithm.

---

**Algorithm 3** Projected Gradient Descent (**PGD**)

---

1: **Input:** Initial parameter $\mathbf{y}^0$, Current iterate $\mathbf{x}$, # iter $T$, step-sizes $\{\gamma^r\}_{r=0}^{T-1}$, Constraints $A, \mathbf{b}$
2: **for** $r = 0, 1, \ldots, T-1$ **do**
3:    $\mathbf{y}^{r+1} = \mathbf{y}^r - \gamma^r \nabla_y g(\mathbf{x}, \mathbf{y}^r)$
4:    Project $\mathbf{y}^{r+1}$ to $\mathcal{Y} = \{\mathbf{y} \in \mathbb{R}^{d_\ell} | A\mathbf{y} \leq \mathbf{b}\}$ by solving the following QP:

$$\min_{\mathbf{y} \in \mathbb{R}^{d_\ell}} \|\mathbf{y} - \mathbf{y}^{r+1}\|^2 \text{ s.t. } A\mathbf{y} \leq \mathbf{b} \tag{15}$$

5: **end for**

---

# C  ADDITIONAL EXPERIMENTS

In this section, we include additional experiments on quadratic bilevel optimization problems and Adversarial training along with the implementation details. First, we evaluate the performance of the [D]SIGD and [S]SIGD on quadratic bilevel optimization problems.

## C.1  NUMERICAL RESULTS

We consider the following linearly constrained quadratic bilevel problems of the form (1) with the UL and the LL objectives defined as:

$$f(\mathbf{x}, \mathbf{y}) = \frac{1}{4}\|\mathbf{x}\|^2 + 5\mathbf{x}^T\mathbf{y} - \frac{1}{4}\|\mathbf{y}\|^2 , \ h(\mathbf{x}, \mathbf{y}) = \frac{1}{4}\|\mathbf{x}\|^2 + \frac{1}{2}\mathbf{x}^T\mathbf{y} + \frac{1}{4}\|\mathbf{y}\|^2 \tag{16}$$

$$f(\mathbf{x}, \mathbf{y}) = \frac{1}{2}\|\mathbf{x}\|^2 + 2\mathbf{x}^T\mathbf{y} - \frac{1}{2}\|\mathbf{y}\|^2 , \ h(\mathbf{x}, \mathbf{y}) = \mathbf{x}^T\mathbf{y} + \frac{1}{2}\|\mathbf{y}\|^2. \tag{17}$$

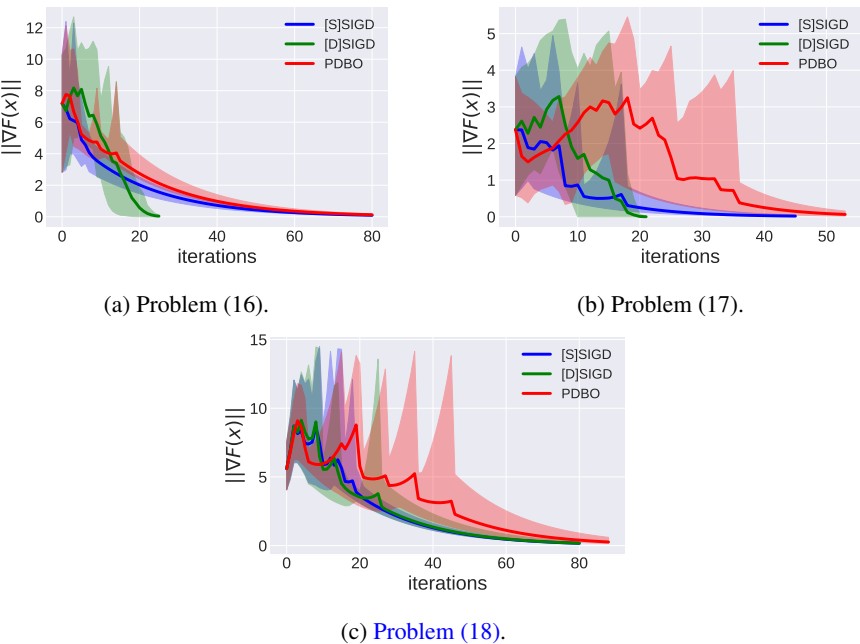

(a) Problem (16).

(b) Problem (17).

(c) Problem (18).

Figure 3: Convergence curves w.r.t. number of iterations.

$$f(\mathbf{x}, \mathbf{y}) = \frac{1}{4}\|\mathbf{x}\|^2 + 2\mathbf{x}^T\mathbf{y} - \frac{1}{4}\|\mathbf{y}\|^2 + 1 \, , \; h(\mathbf{x}, \mathbf{y}) = \mathbf{x}^T\mathbf{y} + \frac{1}{2}\|\mathbf{y}\|^2 + \mathbf{1^T}\mathbf{x} + \mathbf{1^T}\mathbf{y}. \qquad (18)$$

In the first two cases, we have $d_u = d_l = 2$, and the linear constraints in the LL are of the form $-1 \le y_i \le 1, \; i \in \{1, 2\}$. In the third example, we have $d_u = d_l = 2$, and the linear constraints in the LL are of the form $-5 \le y_i \le 5, \; i \in \{1, 2\}, \; -5 \le y_1 + y_2 \le 5$. We compare the performance of SIGD algorithms to recently proposed PDBO (Sow et al., 2022). In Figures 3a, 3b and 3c, we present the evolution of the stationarity gap $\|\nabla F(\mathbf{x})\|$ during the execution of the three algorithms, for the problems (16) (17) and (18), respectively. The results are averaged over 10 random runs, and the variance of the results across these runs is reflected on the shaded region across the convergence curves. In our experiments, we choose the step-size using the backtracking line search for [D]SIGD as stated in Algorithm 1, while for [S]SIGD we choose a constant step-size. Note that since all problems are deterministic [S]SIGD utilizes a gradient estimator with zero variance.

In problem (16), we solve the LL problem using 10 steps of projected gradient descent with stepsize 0.1; in the case of [D]SIGD the stepsize is 1. For the stepsize of [S]SIGD, we choose $\beta = 0.1$, while in [D]SIGD we find the proper Armijo step-size by successively adapting (by increasing $m$) the quantity $a_r = (0.9)^m$ until condition (11) is met. In PDBO we select $0.1$ for the stepsizes of both the primal and dual steps, and the number of inner iterations is set to 10. In problem (17), we solve the LL problem using 20 steps of projected gradient descent with stepsize 0.1; in the case of [D]SIGD the number of steps is 10 and the stepsize is 1. For the stepsize of [S]SIGD, we choose $\beta = 0.1$, while in [D]SIGD we find the proper Armijo step-size by successively adapting (by increasing $m$) the quantity $a_r = (0.95)^m$ until condition (11) is met. In PDBO we select $0.1$ for the stepsizes of both the primal and dual steps, and the number of inner iterations is set to 20. In problem (18), we solve the LL problem (of both [D]SIGD and [S]SIGD) using 10 steps of projected gradient descent with stepsize 0.1. For the stepsize of [S]SIGD, we choose $\beta = 0.1$, while in [D]SIGD we find the proper Armijo step-size by successively adapting (by increasing $m$) the quantity $a_r = (0.9)^m$ until condition (11) is met. In PDBO we select $0.1$ for the stepsizes of both the primal and dual steps, and the number of inner iterations is set to 10.

## C.2    ADVERSARIAL LEARNING

In this section, we present some additional results along with the implementation details for the adversarial learning problem. As noted earlier, we consider the adversarial learning problem of form (3). For learning the perturbation $\mathbf{y}^*(\mathbf{x})$, we focus on the $\epsilon$-tolerant $\ell_\infty$-norm attack constraint, i.e., $\mathcal{Y} = \{\mathbf{y} \in \mathbb{R}^{d_\ell} \,|\, \|\mathbf{y}\|_\infty \leq \epsilon\}$. Note that this constraint can easily be expressed as a linear inequality constraint as in the LL problem in (3). In particular, we evaluate the performance of [S]SIGD on two widely used attack budget choices of $\epsilon \in \{8/255, 16/255\}$ (Madry et al., 2017; Zhang et al., 2019b; Wong et al., 2020; Andriushchenko & Flammarion, 2020; Zhang et al., 2019a). In the implementation of our [S]SIGD method, we adopt a perturbation generated by a Gaussian random vector $\mathbf{q}$ with variances from the following list $\sigma^2 \in \{2e{-}5, 4e{-}5, 6e{-}5, 8e{-}5, 1e{-}4, \}$, in order to study different levels of smoothness. We choose $f_i$ to be cross-entropy loss and $h_i = -f_i + \lambda\|\mathbf{y}_i\|^2$ with $\lambda > 0$ as a hyper-parameter. For solving (3), in each iteration we select a fixed batch of samples for both the UL and LL problems. Also, note that the ReLU-based neural networks commonly lead to a piece-wise linear decision boundary w.r.t. the inputs (Moosavi-Dezfooli et al., 2019). This implies that the implicit gradient in (10) can be further approximated using a Hessian-free implementation, where the Hessian of the LL problem can be approximated by $\lambda I$ (Zhang et al., 2021, Eq. (25)). Note that these approximations are common in practice and do not lead to performance degradation compared to the case when full Hessian is used to compute the implicit-gradient (Zhang et al., 2021, Table 5). Next, we analyze the effect of adding different perturbations $\mathbf{q}$ in the LL problem on the performance of [S]SIGD. Specifically, we choose $\mathbf{q} \sim \mathcal{N}(\mathbf{0}, \sigma^2\mathbf{I})$ and evaluate the performance of [S]SIGD with $\sigma^2$.

In Figure 4, we plot the robust accuracy (RA) and the standard accuracy (SA) with respect to the variance of the Gaussian perturbation vector used in the LL problem. As can be seen, the RA increases as the variance increases within a certain range. However, with stronger noise (i.e., $\sigma^2 > 10^{-4}$), the RA drops sharply, while the SA increases. This is reasonable, since high variance makes the true LL gradient noisy. For easier observation, in Figure 5 we zoom in the part of Figure 4 where $\sigma^2 \in [0, 8 \cdot 10^{-5}]$. It can be clearly seen that adding a small perturbation $\mathbf{q}$ helps in improving the RA.

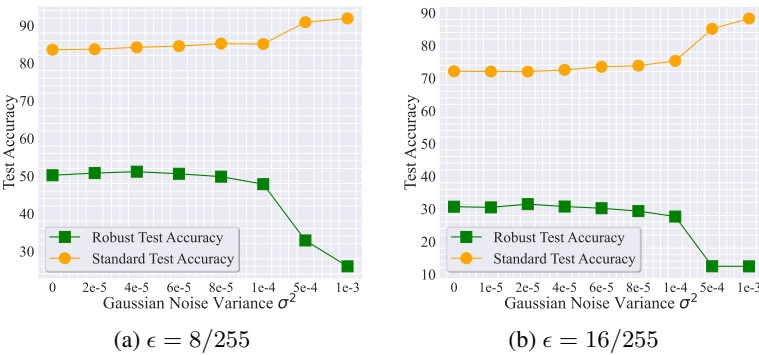

(a) $\epsilon = 8/255$            (b) $\epsilon = 16/255$

Figure 4: The influence of Gaussian variance on the RA and SA. The experiments are based on CIFAR-10 with ResNet-18 model.

Next, we compare the performance of [S]SIGD against two widely accepted adversarial learning methods as baselines, namely AT (Madry et al., 2017) and TRADES (Zhang et al., 2019b). Here, we present the results for CIFAR-10 dataset (Krizhevsky et al., 2009) and adopt the ResNet-18 (He et al., 2016). In Table 2, we compare the performance of [S]SIGD for different perturbation variances with classical AT (Madry et al., 2017) algorithm and TRADES (Zhang et al., 2019b). Note that for appropriate choice of perturbation variance [S]SIGD outperforms the classical AT algorithm while performs is only slightly worse compared to TRADES, especially, for higher attack budget of $\epsilon = 16/255$.

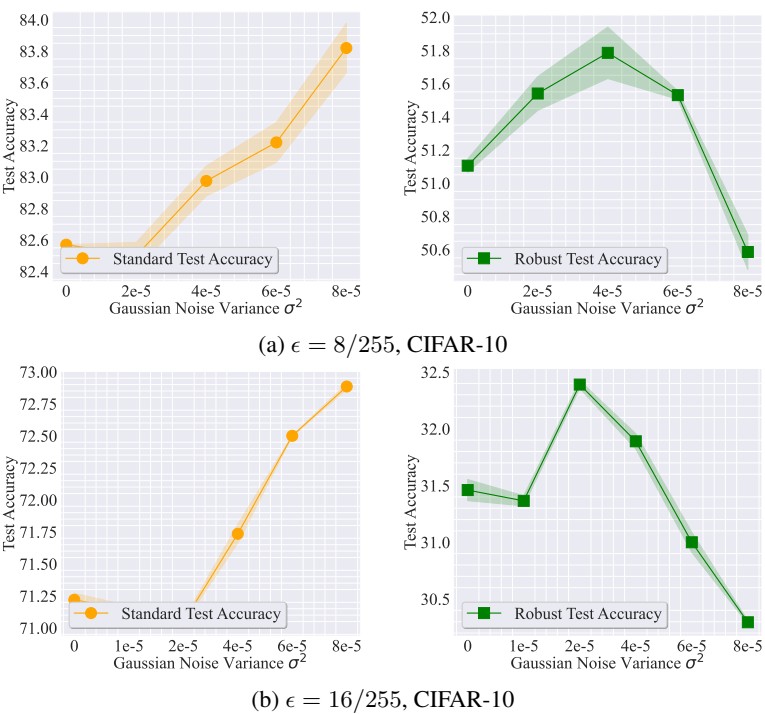

Figure 5: The influence of Gaussian variance on the RA and SA.

Table 2: Performance overview of different methods on CIFAR-10 (Krizhevsky et al., 2009) with ResNet-18 (He et al., 2016). The result $a_{\pm b}$ represents the mean value $a$ with a standard deviation of $b$ over 5 random trials.

| | | | CIFAR-10, $\epsilon = 8/255$ | | | | |
|---|---|---|---|---|---|---|---|
| | | | [S]SIGD (Gaussian variance $\sigma^2$) | | | | |
| Metrics | AT | TRADES | $2e-5$ | $4e-5$ | $6e-5$ | $8e-5$ | $1e-4$ |
| SA | $80.78_{\pm 0.23}$ | $80.23_{\pm 0.23}$ | $80.70_{\pm 0.14}$ | $81.20_{\pm 0.22}$ | $81.52_{\pm 0.21}$ | $83.19_{\pm 0.24}$ | $85.08_{\pm 0.44}$ |
| RA | $50.71_{\pm 0.21}$ | $51.17_{\pm 0.19}$ | $50.78_{\pm 0.21}$ | $51.15_{\pm 0.19}$ | $50.59_{\pm 0.18}$ | $49.83_{\pm 0.23}$ | $47.83_{\pm 0.13}$ |
| | | | $\epsilon = 16/255$ | | | | |
| SA | $70.31_{\pm 0.11}$ | $70.22_{\pm 0.29}$ | $71.43_{\pm 0.14}$ | $72.79_{\pm 0.24}$ | $73.50_{\pm 0.09}$ | $73.98_{\pm 0.35}$ | $75.31_{\pm 0.33}$ |
| RA | $32.12_{\pm 0.18}$ | $33.35_{\pm 0.14}$ | $32.72_{\pm 0.25}$ | $31.73_{\pm 0.10}$ | $29.97_{\pm 0.14}$ | $29.39_{\pm 0.15}$ | $27.67_{\pm 0.07}$ |

# D  PROOFS

## D.1  PROOFS OF SECTION 2

### D.1.1  PROOF OF PROPOSITION 1

Note that the goal of Proposition 1 is to establish the continuity of the mapping $\overline{\mathbf{y}}^*(\mathbf{x})$ and the implicit function $G(\mathbf{x}) := f(\mathbf{x}, \overline{\mathbf{y}}^*(\mathbf{x}))$. In the following, we will show that under Assumption 1, $\overline{\mathbf{y}}^*(\mathbf{x})$ is in fact continuous, which will then utilize to establish the continuity of $G(\mathbf{x})$. Before starting the proof we need a few definitions. Consider the LL problem (1b) and let us denote the set $\mathcal{Y} = \{\mathbf{y} \in \mathbb{R}^{d_\ell} | A\mathbf{y} \le \mathbf{b}\}$. Note that in general the constraint set $\mathcal{Y}$ can depend on the UL variable $\mathbf{x} \in \mathcal{X}$. For such cases, $\mathcal{Y}(\mathbf{x})$ is a set valued map $\mathcal{Y} : \mathcal{X} \to \mathbb{R}^{d_\ell}$ and is referred to as a correspondence. However, for the bilevel problem in (1a) and (1b) the correspondence $\mathcal{Y}$ is independent of $\mathbf{x} \in \mathcal{X}$ and is a fixed set. Also, we define the upper-semi continuity (USC) and the lower-semi continuity (LSC) for the correspondence $\mathcal{Y}(\mathbf{x})$. To define these notions of continuity, we will utilize the notion of an $\epsilon$-ball defined below.

**Definition 2** ($\epsilon$-Ball). *For $\mathcal{Y} \subset \mathbb{R}^{d_\ell}$, and given $\epsilon > 0$, we define the open ball about $\mathcal{Y}$ as*

$$B_\epsilon(\mathcal{Y}) \coloneqq \left\{ \mathbf{y} \in \mathbb{R}^{d_\ell} \mid \|\mathbf{y} - \mathbf{y}'\| < \epsilon, \text{ for some } \mathbf{y}' \in \mathcal{Y} \right\},$$

*where $\|\cdot\|$ is the standard Euclidean norm.*

Using the $\epsilon$-ball we define the Upper Semi-Continuity (USC) of the correspondence $\mathcal{Y}$.

**Definition 3** (Upper Semi-Continuity (USC)). *The correspondence $\mathcal{Y} : \mathcal{X} \to \mathbb{R}^{d_\ell}$ is USC if for every $\mathbf{x} \in \mathcal{X}$ and $\epsilon > 0$, there exists a $\delta > 0$ such that $\mathcal{Y}(\mathbf{x}') \subset B_\epsilon(\mathcal{Y}(\mathbf{x})))$, if $\mathbf{x}' \in \mathcal{X}$ and $\|\mathbf{x} - \mathbf{x}'\| < \delta$.*

Next, we define the notion of Lower Semi-Continuity (LSC).

**Definition 4** (Lower Semi-Continuity (LSC)). *The correspondence $\mathcal{Y} : \mathcal{X} \to \mathbb{R}^{d_\ell}$ is LSC if for any sequence $\mathbf{x}_n$ in $\mathcal{X}$ that converges to a point $\mathbf{x} \in \mathcal{X}$, and $\mathbf{y} \in \mathcal{Y}(\mathbf{x})$, there exists a sequence $\mathbf{y}_n$ such that $\mathbf{y}_n \in \mathcal{Y}(\mathbf{x}_n)$, for all $n \in \mathbb{N}$, and $\lim_{n\to\infty} \mathbf{y}_n = \mathbf{y}$.*

**Theorem 5** (Berge's Theorem of Maximum (Lecture, 2017)). *Let $\mathcal{X} \subset \mathbb{R}^{d_u}$ be a non-empty set. Also, let $\mathcal{Y} : \mathcal{X} \to \mathbb{R}^{d_\ell}$ be a correspondence such that the set $\mathcal{Y}(\mathbf{x})$ is compact and non-empty for all $\mathbf{x} \in \mathcal{X}$, and $\mathcal{Y}$ is USC and LSC. Then, if $g : \mathcal{X} \times \mathbb{R}^{d_\ell} \to \mathbb{R}$ is a continuous function with $\overline{\mathbf{y}}^*(\mathbf{x})$ defined as*

$$\overline{\mathbf{y}}^*(\mathbf{x}) \in \underset{\mathbf{y}\in\mathbb{R}^{d_\ell}}{\arg\min} \left\{ g(\mathbf{x}, \mathbf{y}) \mid \mathbf{y} \in \mathcal{Y}(\mathbf{x}) \right\},$$

*the correspondence $\overline{\mathbf{y}}^*(\mathbf{x})$ is non-empty for all $\mathbf{x} \in \mathcal{X}$, and USC.*

**Remark 2.** *If $\overline{\mathbf{y}}^*(\mathbf{x})$ is singleton, then USC implies the continuity of the map $\overline{\mathbf{y}}^*(\mathbf{x}) : \mathcal{X} \to \mathcal{Y}$.* $\square$

Next, we present the proof of Proposition 1.

*Proof.* The proof of proposition 1 follows from the application of Berge's theorem.

To begin with, note that for our problem the set $\mathcal{Y}$ is a fixed set independent of $\mathbf{x} \in \mathcal{X}$. We are going to verify the conditions of Theorem 5. First, note from Assumption 1(b) that the set $\mathcal{Y}$ is non-empty and compact. Then, it is easy to see that $\mathcal{Y} \subset B_\epsilon(\mathcal{Y})$, for every $\epsilon > 0$, and that implies the LSC of $\mathcal{Y}$. Moreover, since the set $\mathcal{Y}$ is independent of $\mathbf{x} \in \mathcal{X}$ and compact, for every sequence $\mathbf{x}_n \to \mathbf{x}$ in $\mathcal{X}$, we can always find a sequence $\mathbf{y}_n \to \mathbf{y}$, such that $\mathbf{y}_n, \mathbf{y} \in \mathcal{Y}$. Therefore, $\mathcal{Y}$ is LSC. Finally, using Assumption 1(a) we see that the function $g(\mathbf{x}, \mathbf{y})$ is continuous. Then, Theorem 5 implies that the set $\overline{\mathbf{y}}^*(\mathbf{x})$ is non-empty and the correspondence USC.

Using the strong-convexity of $g(\mathbf{x}, \mathbf{y})$ with respect to $\mathbf{y}$ (Assumption 1(c)) we claim that $\overline{\mathbf{y}}^*(\mathbf{x})$ will be a singleton, and thereby a continuous mapping. Then, the continuity of $\overline{\mathbf{y}}^*(\mathbf{x})$ implies the continuity of $G(\mathbf{x}) \coloneqq f(\mathbf{x}, \overline{\mathbf{y}}^*(\mathbf{x}))$, since the composition of two continuous functions is continuous. The proof is now complete. $\square$

### D.1.2 Proof of Lemma 2

*Proof.* In this proof we follow a reasoning similar to (Parise & Ozdaglar, 2017, Thm. 1). However, differently from that work we consider bilevel problems rather than Nash games. To begin with, consider the Lagrangian of problem (5), i.e.,

$$L(\mathbf{x}, \mathbf{y}, \boldsymbol{\lambda}) = g(\mathbf{x}, \mathbf{y}) + \boldsymbol{\lambda}^T (A\mathbf{y} - \mathbf{b}).$$

Then, for some fixed $\mathbf{x} \in \mathcal{X}$, consider a KKT point $(\mathbf{y}^*(\mathbf{x}), \boldsymbol{\lambda}^*(\mathbf{x}))$ of (5), for which it holds that,

- $\nabla_y L(\mathbf{x}, \mathbf{y}^*(\mathbf{x}), \boldsymbol{\lambda}^*(\mathbf{x})) = \nabla_y g(\mathbf{x}, \mathbf{y}^*(\mathbf{x})) + A^T \boldsymbol{\lambda}^*(\mathbf{x}) = 0$

- $[\boldsymbol{\lambda}^*(\mathbf{x})]^T (A\mathbf{y}^*(\mathbf{x}) - \mathbf{b}) = 0$

- $\boldsymbol{\lambda}^*(\mathbf{x}) \geq 0$

- $A\mathbf{y}^*(\mathbf{x}) - \mathbf{b} \leq 0.$

Now, consider the active constraints at $(\mathbf{y}^*(\mathbf{x}), \boldsymbol{\lambda}^*(\mathbf{x}))$, and to simplify notation let us set $\overline{A} := \overline{A}(\mathbf{y}^*(\mathbf{x}))$. Using the notations defined in Section 2 and the SC property, the KKT conditions given above can be equivalently rewritten as

$$\nabla_y g(\mathbf{x}, \mathbf{y}^*(\mathbf{x})) + \overline{A}^T \overline{\boldsymbol{\lambda}}^*(\mathbf{x}) = 0, \quad \overline{A}\mathbf{y}^*(\mathbf{x}) - \overline{\mathbf{b}} = 0, \quad \overline{\boldsymbol{\lambda}}^*(\mathbf{x}) > 0, \tag{19}$$

where $\overline{\boldsymbol{\lambda}}^*(\mathbf{x})$ is the subvector of $\boldsymbol{\lambda}^*(\mathbf{x})$ that contains only the elements whose indices correspond to the active constraints at $\mathbf{y} = \mathbf{y}^*(\mathbf{x})$. Moreover, notice that the point $(\mathbf{y}^*(\mathbf{x}), \boldsymbol{\lambda}^*(\mathbf{x}))$ is unique. The uniqueness of $\mathbf{y}^*(\mathbf{x})$ follows from the strong convexity of $g(\mathbf{x}, \cdot)$; the uniqueness of $\boldsymbol{\lambda}^*(\mathbf{x})$ results from the fact that matrix $\overline{A}$ has full row rank (which guarantees regularity, e.g., see Bertsekas (1998)).

As mentioned in section 2, the SC condition (from Lemma 1) combined with Assumption 1 implies that the mapping $\mathbf{y}^*(\mathbf{x})$ is differentiable almost surely (Friesz & Bernstein, 2015, Theorem 2.22). As a result, at any given point $\mathbf{x}$, we can consider a sufficiently small neighborhood around it, such that the active constraints $\overline{A}$ remain unchanged. Then, we can compute the gradient of (19) using the implicit function theorem as follows

$$\nabla_{xy}^2 g(\mathbf{x}, \mathbf{y}^*(\mathbf{x})) + \nabla_{yy}^2 g(\mathbf{x}, \mathbf{y}^*(\mathbf{x}))\nabla\mathbf{y}^*(\mathbf{x}) + \overline{A}^T \nabla\overline{\boldsymbol{\lambda}}^*(\mathbf{x}) = 0 \tag{20}$$

$$\overline{A}\nabla\mathbf{y}^*(\mathbf{x}) = 0. \tag{21}$$

Solving the (20) for $\nabla\mathbf{y}^*(\mathbf{x})$ yields

$$\nabla\mathbf{y}^*(\mathbf{x}) = \left[\nabla_{yy}^2 g(\mathbf{x}, \mathbf{y}^*(\mathbf{x}))\right]^{-1}\left[-\nabla_{xy}^2 g(\mathbf{x}, \mathbf{y}^*(\mathbf{x})) - \overline{A}^T \nabla\overline{\boldsymbol{\lambda}}^*(\mathbf{x})\right], \tag{22}$$

where we exploited the fact that the Hessian matrix $\nabla_{yy}^2 g(\mathbf{x}, \mathbf{y}^*(\mathbf{x}))$ is positive definite and thus invertible. Substituting (22) into (21) gives

$$\overline{A}\left[\nabla_{yy}^2 g(\mathbf{x}, \mathbf{y}^*(\mathbf{x}))\right]^{-1}\left[-\nabla_{xy}^2 g(\mathbf{x}, \mathbf{y}^*(\mathbf{x})) - \overline{A}^T \nabla\overline{\boldsymbol{\lambda}}^*(\mathbf{x})\right] = 0$$

$$\implies \nabla\overline{\boldsymbol{\lambda}}^*(\mathbf{x}) = -\left[\overline{A}\left[\nabla_{yy}^2 g(\mathbf{x}, \mathbf{y}^*(\mathbf{x}))^{-1}\right]\overline{A}^T\right]^{-1}\left[\overline{A}\left[\nabla_{yy}^2 g(\mathbf{x}, \mathbf{y}^*(\mathbf{x}))\right]^{-1}\nabla_{xy}^2 g(\mathbf{x}, \mathbf{y}^*(\mathbf{x}))\right].$$

Finally, note that the KKT point $\mathbf{y}^*(\mathbf{x})$ corresponds to the unique global minimum of (5), due to the strong convexity of $g(\mathbf{x}, \cdot)$. The proof is now complete. $\qquad\square$

### D.1.3 THE PROOF OF LEMMA 3

The proof of Lemma 3 requires several intermediate results which we provide below. Note that under Assumption 2(c) it holds that $\overline{A}(\mathbf{y}^*(\mathbf{x})) = \overline{A}(\widehat{\mathbf{y}}(\mathbf{x}))$; for simplicity we will denote these matrices as $\overline{A}$ in the derivations of this subsection. Moreover, for any given matrix $A$ we will denote with $L_A$ the maximum value of the quantity $\|\overline{A}(\widehat{\mathbf{y}}(\mathbf{x}))\|$, across all $\mathbf{x} \in \mathcal{X}$.

**Lemma 6.** *Suppose that Assumption 1,2,3 hold. Then for any $\mathbf{x} \in \mathcal{X}$, we have:*

(a) $\left\|\left[\nabla_{yy}^2 g(\mathbf{x}, \mathbf{y})\right]^{-1}\right\| \leq \frac{1}{\mu_g}, \ \forall \mathbf{y} \in \mathbb{R}^{d_\ell}.$

(b) $\left\|\left[\nabla_{yy}^2 g(\mathbf{x}, \mathbf{y}^*(\mathbf{x}))\right]^{-1} - \left[\nabla_{yy}^2 g(\mathbf{x}, \widehat{\mathbf{y}}(\mathbf{x}))\right]^{-1}\right\| \leq \left(\frac{1}{\mu_g}\right)^2 L_{g_{yy}}\delta.$

(c) $\left\|\left[\overline{A}\left[\nabla_{yy}^2 g(\mathbf{x}, \mathbf{y})\right]^{-1}\overline{A}^T\right]^{-1}\right\| \leq \overline{L}_A, \ \forall \mathbf{y} \in \mathbb{R}^{d_\ell}.$

(d) $\left\|\left[\overline{A}\left[\nabla_{yy}^2 g(\mathbf{x}, \mathbf{y}^*(\mathbf{x}))\right]^{-1}\overline{A}^T\right]^{-1} - \left[\overline{A}\left[\nabla_{yy}^2 g(\mathbf{x}, \widehat{\mathbf{y}}(\mathbf{x}))\right]^{-1}\overline{A}^T\right]^{-1}\right\| \leq L_A^2\overline{L}_A^2\frac{1}{\mu_g^2}L_{g_{yy}}\delta.$

*Proof.* a) We know that $g(\mathbf{x}, \mathbf{y})$ is strongly convex in $\mathbf{y}$ with modulus $\mu_g$. Therefore, for any $\mathbf{x} \in \mathcal{X}$ we have

$$\nabla_{yy}^2 g(\mathbf{x}, \mathbf{y}) \succeq \mu_g I \succ 0, \forall \mathbf{y} \in \mathbb{R}^{d_\ell}$$

$$\implies 0 \prec \left[\nabla_{yy}^2 g(\mathbf{x}, \mathbf{y})\right]^{-1} \preceq \frac{1}{\mu_g}I, \forall \mathbf{y} \in \mathbb{R}^{d_\ell}$$

$$\implies \left\|\left[\nabla_{yy}^2 g(\mathbf{x}, \mathbf{y})\right]^{-1}\right\| \leq \frac{1}{\mu_g}, \forall \mathbf{y} \in \mathbb{R}^{d_\ell}. \tag{23}$$

b) To begin with, notice that for arbitrary square invertible matrices $P, Q$ we have

$$\left\|P^{-1} - Q^{-1}\right\| = \left\|P^{-1}(Q - P)Q^{-1}\right\| \le \left\|Q^{-1}(P - Q)\right\| \left\|P^{-1}\right\| \le \left\|Q^{-1}\right\| \left\|P - Q\right\| \left\|P^{-1}\right\|. \tag{24}$$

Then, using the above inequality we get

$$\left\|\left[\nabla_{yy}^2 g(\mathbf{x}, \mathbf{y}^*(\mathbf{x}))\right]^{-1} - \left[\nabla_{yy}^2 g(\mathbf{x}, \widehat{\mathbf{y}}(\mathbf{x}))\right]^{-1}\right\|$$

$$\le \left\|\left[\nabla_{yy}^2 g(\mathbf{x}, \mathbf{y}^*(\mathbf{x}))\right]^{-1}\right\| \left\|\nabla_{yy}^2 g(\mathbf{x}, \mathbf{y}^*(\mathbf{x})) - \nabla_{yy}^2 g(\mathbf{x}, \widehat{\mathbf{y}}(\mathbf{x}))\right\| \left\|\left[\nabla_{yy}^2 g(\mathbf{x}, \widehat{\mathbf{y}}(\mathbf{x}))\right]^{-1}\right\|$$

$$\le \left(\frac{1}{\mu_g}\right)^2 L_{g_{yy}} \|\mathbf{y}^*(\mathbf{x}) - \widehat{\mathbf{y}}(\mathbf{x})\|$$

$$\le \left(\frac{1}{\mu_g}\right)^2 L_{g_{yy}} \delta,$$

where in the second inequality we used the result from Lemma 6(a) and the Lipschitz Hessian property of $g$ in $\mathbf{yy}$ (Assumption 3(d)); in the third inequality we use the Assumption 2(a) for $\mathbf{y}(\mathbf{x})$.

c) In our problem we have that $g$ strongly convex in $\mathbf{y}$ and Lipschitz gradient in $\mathbf{y}$. Thus, for any $\mathbf{x} \in \mathcal{X}$ we have

$$L_y I \succeq \nabla_{yy}^2 g(\mathbf{x}, \mathbf{y}^*) \succeq \mu_g I \succ 0$$

$$\implies 0 \prec \frac{1}{L_y} I \preceq \left[\nabla_{yy}^2 g(\mathbf{x}, \mathbf{y})\right]^{-1} \preceq \frac{1}{\mu_g} I, \forall \mathbf{y} \in \mathbb{R}^{d_\ell}.$$

Also, for every $\mathbf{x} \in \mathcal{X}$, we have that

$$\left\|\overline{A}^T \mathbf{z}\right\|^2 = \mathbf{z}^T \overline{A} \overline{A}^T \mathbf{z} \ge \lambda_{min}(\overline{A} \overline{A}^T) \|\mathbf{z}\|^2, \forall \mathbf{z} \in \mathbb{R}^{d_\ell}.$$

Using the above two lower bound we get

$$\mathbf{z}^T \overline{A} \left[\nabla_{yy}^2 g(\mathbf{x}, \mathbf{y})\right]^{-1} \overline{A}^T \mathbf{z} \ge \frac{1}{L_y} \lambda_{min}(\overline{A} \overline{A}^T) \|\mathbf{z}\|^2 > 0, \forall \mathbf{z} \in \mathbb{R}^{d_\ell} \setminus \{0\},$$

where the last inequality follows from the fact that $\overline{A}$ is full row rank which implies that $\lambda_{min}(\overline{A} \overline{A}^T) > 0, \forall \mathbf{x} \in \mathcal{X}$.

Since the above inequality holds for every $\mathbf{z} \in \mathbb{R}^{d_\ell} \setminus \{0\}$, it $\forall \mathbf{x} \in \mathcal{X}$ we get that

$$\overline{A} \left[\nabla_{yy}^2 g(\mathbf{x}, \mathbf{y})\right]^{-1} \overline{A}^T \succeq \frac{\lambda_{min}(\overline{A} \overline{A}^T)}{L_y} I \succ 0$$

$$\left[\overline{A} \left[\nabla_{yy}^2 g(\mathbf{x}, \mathbf{y})\right]^{-1} \overline{A}^T\right]^{-1} \preceq \frac{L_y}{\lambda_{min}(\overline{A} \overline{A}^T)} I$$

$$\left\|\left[\overline{A} \left[\nabla_{yy}^2 g(\mathbf{x}, \mathbf{y})\right]^{-1} \overline{A}^T\right]^{-1}\right\| \le \frac{L_y}{\lambda_{min}(\overline{A} \overline{A}^T)}.$$

Finally, for the given matrix $A$, consider the submatrix $\overline{A} = \overline{A}(\widehat{\mathbf{y}}(\mathbf{x}))$ generated by considering only the subset of its rows corresponding to the active constraints at $\widehat{\mathbf{y}}(\mathbf{x})$. From Assumption 1(c) we know that $\overline{A}(\widehat{\mathbf{y}}(\mathbf{x}))$ is full row rank for every $\mathbf{x} \in \mathcal{X}$, and so we can ensure that $\lambda_{min}\left(\overline{A}(\widehat{\mathbf{y}}(\mathbf{x})) \overline{A}(\widehat{\mathbf{y}}(\mathbf{x}))^T\right) > 0, \forall \mathbf{x} \in \mathcal{X}$. Then, we denote with $\lambda_{min}$ the minimum value of the quantity $\lambda_{min}\left(\overline{A}(\widehat{\mathbf{y}}(\mathbf{x})) \overline{A}(\widehat{\mathbf{y}}(\mathbf{x}))^T\right)$ across all $\mathbf{x} \in \mathcal{X}$. Therefore, we conclude that

$$\left\|\left[\overline{A} \left[\nabla_{yy}^2 g(\mathbf{x}, \mathbf{y})\right]^{-1} \overline{A}^T\right]^{-1}\right\| \le \frac{L_y}{\lambda_{min}} := \overline{L}_A.$$

d) Applying formula (24) with $P = \overline{A} \left[ \nabla_{yy}^2 g(\mathbf{x}, \mathbf{y}^*(\mathbf{x})) \right]^{-1} \overline{A}^T, Q = \overline{A} \left[ \nabla_{yy}^2 g(\mathbf{x}, \widehat{\mathbf{y}}(\mathbf{x})) \right]^{-1} \overline{A}^T$ we get

$$\left\| \left[ \overline{A} \left[ \nabla_{yy}^2 g(\mathbf{x}, \mathbf{y}^*(\mathbf{x})) \right]^{-1} \overline{A}^T \right]^{-1} - \left[ \overline{A} \left[ \nabla_{yy}^2 g(\mathbf{x}, \widehat{\mathbf{y}}(\mathbf{x})) \right]^{-1} \overline{A}^T \right]^{-1} \right\|$$

$$\leq \left\| \left[ \overline{A} \left[ \nabla_{yy}^2 g(\mathbf{x}, \mathbf{y}^*(\mathbf{x})) \right]^{-1} \overline{A}^T \right]^{-1} \right\| \left\| \overline{A} \left[ \nabla_{yy}^2 g(\mathbf{x}, \mathbf{y}^*(\mathbf{x})) \right]^{-1} \overline{A}^T - \overline{A} \left[ \nabla_{yy}^2 g(\mathbf{x}, \widehat{\mathbf{y}}(\mathbf{x})) \right]^{-1} \overline{A}^T \right\|$$

$$\left\| \left[ \overline{A} \left[ \nabla_{yy}^2 g(\mathbf{x}, \widehat{\mathbf{y}}(\mathbf{x})) \right]^{-1} \overline{A}^T \right]^{-1} \right\|$$

$$\leq \left\| \left[ \overline{A} \left[ \nabla_{yy}^2 g(\mathbf{x}, \mathbf{y}^*(\mathbf{x})) \right]^{-1} \overline{A}^T \right]^{-1} \right\| \left\| \overline{A} \left[ \left[ \nabla_{yy}^2 g(\mathbf{x}, \mathbf{y}^*(\mathbf{x})) \right]^{-1} - \left[ \nabla_{yy}^2 g(\mathbf{x}, \widehat{y}(\mathbf{x})) \right]^{-1} \right] \overline{A}^T \right\|$$

$$\left\| \left[ \overline{A} \left[ \nabla_{yy}^2 g(\mathbf{x}, \widehat{\mathbf{y}}(\mathbf{x})) \right]^{-1} \overline{A}^T \right]^{-1} \right\|$$

$$\leq \left\| \left[ \overline{A} \left[ \nabla_{yy}^2 g(\mathbf{x}, \mathbf{y}^*(\mathbf{x})) \right]^{-1} \overline{A}^T \right]^{-1} \right\| \|\overline{A}\| \left\| \left[ \nabla_{yy}^2 g(\mathbf{x}, \mathbf{y}^*(\mathbf{x})) \right]^{-1} - \left[ \nabla_{yy}^2 g(\mathbf{x}, \widehat{\mathbf{y}}(\mathbf{x})) \right]^{-1} \right\| \left\| \overline{A}^T \right\|$$

$$\left\| \left[ \overline{A} \left[ \nabla_{yy}^2 g(\mathbf{x}, \widehat{\mathbf{y}}(\mathbf{x})) \right]^{-1} \overline{A}^T \right]^{-1} \right\|$$

$$\leq L_A^2 \overline{L}_A^2 \left( \frac{1}{\mu_g} \right)^2 L_{g_{yy}} \delta,$$

where in the final inequality we used the bounds derived in Lemma 6(b), 6(c), and the bound $\|\overline{A}\| \leq L_A$.

The proof is now complete. $\qquad\square$

Now let us bound the norm of the gradients of the mappings $\boldsymbol{\lambda}^*(\mathbf{x})$ and $\mathbf{y}^*(\mathbf{x})$.

**Lemma 7.** *Under Assumptions 1,2,3, the gradients of the mappings $\boldsymbol{\lambda}^*(\mathbf{x})$ and $\mathbf{y}^*(\mathbf{x})$ satisfy the following bounds for every $\mathbf{x} \in \mathcal{X}$,*

$$\|\nabla \overline{\boldsymbol{\lambda}}^*(\mathbf{x})\| \leq \overline{L}_{\boldsymbol{\lambda}^*}, \quad \left\| \widehat{\nabla} \overline{\boldsymbol{\lambda}}^*(\mathbf{x}) \right\| \leq \overline{L}_{\boldsymbol{\lambda}^*}$$

$$\|\nabla \mathbf{y}^*(\mathbf{x})\| \leq \overline{L}_{\mathbf{y}^*}, \quad \|\widehat{\nabla} \mathbf{y}^*(\mathbf{x})\| \leq \overline{L}_{\mathbf{y}^*}$$

*where $\overline{L}_{\boldsymbol{\lambda}^*} = \frac{1}{\mu_g} \overline{L}_A L_A \overline{L}_{g_{xy}}$ and $\overline{L}_{\mathbf{y}^*} = \frac{1}{\mu_y} \left( \overline{L}_{g_{xy}} + L_A \overline{L}_{\boldsymbol{\lambda}^*} \right)$. Note that $\widehat{\nabla} \overline{\boldsymbol{\lambda}}^*(\mathbf{x})$ and $\widehat{\nabla} \mathbf{y}^*(\mathbf{x})$ are obtained by substituting the estimate $\widehat{\mathbf{y}}(\mathbf{x})$ in place of $\mathbf{y}^*(\mathbf{x})$ in the expressions $\nabla \overline{\boldsymbol{\lambda}}^*(\mathbf{x})$ and $\nabla \mathbf{y}^*(\mathbf{x})$, respectively (Please see Lemma 2).*

*Proof.* From Lemma 2 we have

$$\nabla \overline{\boldsymbol{\lambda}}^*(\mathbf{x}) = - \left[ \overline{A} \left[ \nabla_{yy}^2 g(\mathbf{x}, \mathbf{y}^*(\mathbf{x})) \right]^{-1} \overline{A}^T \right]^{-1} \left[ \overline{A} \left[ \nabla_{yy}^2 g(\mathbf{x}, \mathbf{y}^*(\mathbf{x})) \right]^{-1} \nabla_{xy}^2 g(\mathbf{x}, \mathbf{y}^*(\mathbf{x})) \right]$$

Then, taking the norm of this quantity we get,

$$\left\| \nabla \overline{\boldsymbol{\lambda}}^*(\mathbf{x}) \right\| = \left\| \left[ \overline{A} \left[ \nabla_{yy}^2 g(\mathbf{x}, \mathbf{y}^*(\mathbf{x})) \right]^{-1} \overline{A}^T \right]^{-1} \left[ \overline{A} \left[ \nabla_{yy}^2 g(\mathbf{x}, \mathbf{y}^*(\mathbf{x})) \right]^{-1} \nabla_{xy}^2 g(\mathbf{x}, \mathbf{y}^*(\mathbf{x})) \right] \right\|$$

$$\leq \left\| \left[ \overline{A} \left[ \nabla_{yy}^2 g(\mathbf{x}, \mathbf{y}^*(\mathbf{x})) \right]^{-1} \overline{A}^T \right]^{-1} \right\| \|\overline{A}\| \left\| \left[ \nabla_{yy}^2 g(\mathbf{x}, \mathbf{y}^*(\mathbf{x})) \right]^{-1} \right\| \left\| \nabla_{xy}^2 g(\mathbf{x}, \mathbf{y}^*(\mathbf{x})) \right\|$$

$$\leq \overline{L}_A L_A \frac{1}{\mu_g} \overline{L}_{g_{xy}} := \overline{L}_{\boldsymbol{\lambda}^*},$$

where in the last inequality we used Lemma 6(a), 6(c) and Assumption 3(f).

Similarly, for $\|\widehat{\nabla} \overline{\boldsymbol{\lambda}}^*(\mathbf{x})\|$ we have that

$$\left\| \widehat{\nabla} \overline{\boldsymbol{\lambda}}^*(\mathbf{x}) \right\| = \left\| \left[ \overline{A} \left[ \nabla_{yy}^2 g(\mathbf{x}, \widehat{\mathbf{y}}(\mathbf{x})) \right]^{-1} \overline{A}^T \right]^{-1} \left[ \overline{A} \left[ \nabla_{yy}^2 g(\mathbf{x}, \widehat{\mathbf{y}}(\mathbf{x})) \right]^{-1} \nabla_{xy}^2 g(\mathbf{x}, \widehat{\mathbf{y}}(\mathbf{x})) \right] \right\|$$

$$\leq \overline{L}_A L_A \frac{1}{\mu_g} \overline{L}_{g_{xy}} = \overline{L}_{\boldsymbol{\lambda}^*}.$$

Moving to the bound of $\|\nabla \mathbf{y}^*(\mathbf{x})\|$, we know from Lemma 2 that the formula of the gradient of $\mathbf{y}^*(\mathbf{x})$ is

$$\nabla \mathbf{y}^*(\mathbf{x}) = \left[\nabla^2_{yy} g(\mathbf{x}, \mathbf{y}^*(\mathbf{x}))\right]^{-1} \left[-\nabla^2_{xy} g(\mathbf{x}, \mathbf{y}^*(\mathbf{x})) - \overline{A}^T \nabla \overline{\boldsymbol{\lambda}}^*(\mathbf{x})\right]. \tag{25}$$

Then, we have that

$$
\begin{aligned}
\|\nabla \mathbf{y}^*(\mathbf{x})\| &= \left\|\left[\nabla^2_{yy} g(\mathbf{x}, \mathbf{y}^*(\mathbf{x}))\right]^{-1} \left[-\nabla^2_{xy} g(\mathbf{x}, \mathbf{y}^*(\mathbf{x})) - \overline{A}^T \nabla \overline{\boldsymbol{\lambda}}^*(\mathbf{x})\right]\right\| \\
&\leq \left\|\left[\nabla^2_{yy} g(\mathbf{x}, \mathbf{y}^*(\mathbf{x}))\right]^{-1}\right\| \left\|\left[-\nabla^2_{xy} g(\mathbf{x}, \mathbf{y}^*(\mathbf{x})) - \overline{A}^T \nabla \overline{\boldsymbol{\lambda}}^*(\mathbf{x})\right]\right\| \\
&\leq \frac{1}{\mu_g} \left(\left\|\nabla^2_{xy} g(\mathbf{x}, \mathbf{y}^*(\mathbf{x}))\right\| + \|\overline{A}\| \left\|\nabla \overline{\boldsymbol{\lambda}}^*(\mathbf{x})\right\|\right) \\
&\leq \frac{1}{\mu_g} \left(\overline{L}_{g_{xy}} + L_A \overline{L}_{\boldsymbol{\lambda}^*}\right) := \overline{L}_{\mathbf{y}^*},
\end{aligned}
$$

where in the second inequality we used we used Lemma 6(a); the third inequality follows from Assumption 3(f) and the bound for $\left\|\nabla \overline{\boldsymbol{\lambda}}^*(\mathbf{x})\right\|$ we derived above.

Similarly, for $\left\|\widehat{\nabla} \mathbf{y}(\mathbf{x})\right\|$ we can obtain the following bound

$$
\begin{aligned}
\left\|\widehat{\nabla} \mathbf{y}(\mathbf{x})\right\| &= \left\|\left[\nabla^2_{yy} g(\mathbf{x}, \widehat{\mathbf{y}}(\mathbf{x}))\right]^{-1} \left[-\nabla^2_{xy} g(\mathbf{x}, \widehat{\mathbf{y}}(\mathbf{x})) - \overline{A}^T \widehat{\nabla} \overline{\boldsymbol{\lambda}}^*(\mathbf{x})\right]\right\| \\
&\leq \frac{1}{\mu_g} \left(\left\|\nabla^2_{xy} g(\mathbf{x}, \widehat{\mathbf{y}}(\mathbf{x}))\right\| + \|\overline{A}\| \left\|\widehat{\nabla} \overline{\boldsymbol{\lambda}}^*(\mathbf{x})\right\|\right) \\
&\leq \frac{1}{\mu_g} \left(\overline{L}_{g_{xy}} + L_A \overline{L}_{\boldsymbol{\lambda}^*}\right) = \overline{L}_{\mathbf{y}^*}.
\end{aligned}
$$

The proof is now complete. $\qquad\square$

In the next two results we are going to present bounds for the difference of the exact and approximate gradients of the mappings $\overline{\boldsymbol{\lambda}}^*(\mathbf{x})$ and $\nabla \mathbf{y}^*(\mathbf{x})$.

**Lemma 8.** *Suppose that Assumptions 1,2,3 hold. Then, the following bound holds*

$$\|\nabla \overline{\boldsymbol{\lambda}}^*(\mathbf{x}) - \widehat{\nabla} \overline{\boldsymbol{\lambda}}^*(\mathbf{x})\| \leq L_{\boldsymbol{\lambda}^*} \delta,$$

*where* $L_{\boldsymbol{\lambda}^*} = \left(\frac{1}{\mu_g}\right)^3 \overline{L}_A^2 L_A^3 L_{g_{yy}} \overline{L}_{g_{xy}} + \frac{1}{\mu_g} \overline{L}_A L_A L_{g_{xy}} + \left(\frac{1}{\mu_g}\right)^2 \overline{L}_A L_A L_{g_{yy}} \overline{L}_{g_{xy}}.$

*Proof.* Using the derivation of $\nabla \overline{\boldsymbol{\lambda}}^*(\mathbf{x})$ from Lemma 2, and its approximation $\widehat{\nabla} \overline{\boldsymbol{\lambda}}^*(\mathbf{x})$ where we substitute $\mathbf{y}^*(\mathbf{x})$ with $\widehat{\mathbf{y}}(\mathbf{x})$ in the formula of the former, that is,

$$\widehat{\nabla} \overline{\boldsymbol{\lambda}}^*(\mathbf{x}) = - \left[\overline{A} \left[\nabla^2_{yy} g(\mathbf{x}, \widehat{\mathbf{y}}(\mathbf{x}))\right]^{-1} \overline{A}^T\right]^{-1} \left[\overline{A} \left[\nabla^2_{yy} g(\mathbf{x}, \widehat{\mathbf{y}}(\mathbf{x}))\right]^{-1} \nabla^2_{xy} g(\mathbf{x}, \widehat{\mathbf{y}}(\mathbf{x}))\right],$$

we obtain

$$
\begin{aligned}
\left\|\nabla \overline{\boldsymbol{\lambda}}^*(\mathbf{x}) - \widehat{\nabla} \overline{\boldsymbol{\lambda}}^*(\mathbf{x})\right\| = &\left\|\left[\overline{A} \left[\nabla^2_{yy} g(\mathbf{x}, \mathbf{y}^*(\mathbf{x}))\right]^{-1} \overline{A}^T\right]^{-1} \left[\overline{A} \left[\nabla^2_{yy} g(\mathbf{x}, \mathbf{y}^*(\mathbf{x}))\right]^{-1} \nabla^2_{xy} g(\mathbf{x}, \mathbf{y}^*(\mathbf{x}))\right]\right. \\
&\left. - \left[\overline{A} \left[\nabla^2_{yy} g(\mathbf{x}, \widehat{\mathbf{y}}(\mathbf{x}))\right]^{-1} \overline{A}^T\right]^{-1} \left[\overline{A} \left[\nabla^2_{yy} g(\mathbf{x}, \widehat{\mathbf{y}}(\mathbf{x}))\right]^{-1} \nabla^2_{xy} g(\mathbf{x}, \widehat{\mathbf{y}}(\mathbf{x}))\right]\right\|.
\end{aligned}
$$

Below, we use the following notation in order to simplify the derivations.

$$H(\mathbf{x}) = \left[\overline{A} \left[\nabla^2_{yy} g(\mathbf{x}, \mathbf{y}^*(\mathbf{x}))\right]^{-1} \overline{A}^T\right], \; G(\mathbf{x}) = \left[\nabla^2_{yy} g(\mathbf{x}, \mathbf{y}^*(\mathbf{x}))\right]^{-1}, \; M(\mathbf{x}) = \nabla^2_{xy} g(\mathbf{x}, \mathbf{y}^*(\mathbf{x}))$$

$$\widehat{H}(\mathbf{x}) = \left[\overline{A} \left[\nabla^2_{yy} g(\mathbf{x}, \widehat{\mathbf{y}}(\mathbf{x}))\right]^{-1} \overline{A}^T\right], \; \widehat{G}(\mathbf{x}) = \left[\nabla^2_{yy} g(\mathbf{x}, \widehat{\mathbf{y}}(\mathbf{x}))\right]^{-1}, \; \widehat{M}(\mathbf{x}) = \nabla^2_{xy} g(\mathbf{x}, \widehat{\mathbf{y}}(\mathbf{x}))$$

Then, we have that

$$
\begin{aligned}
\|\nabla\overline{\boldsymbol{\lambda}}^*(\mathbf{x}) - \widehat{\nabla}\overline{\boldsymbol{\lambda}}^*(\mathbf{x})\| &= \left\|H^{-1}(\mathbf{x})\overline{A}G(\mathbf{x})M(\mathbf{x}) - \widehat{H}^{-1}(\mathbf{x})\overline{A}\widehat{G}(\mathbf{x})\widehat{M}(\mathbf{x})\right\| \\
&\stackrel{(a)}{\leq} \left\|H^{-1}(\mathbf{x})\overline{A}G(\mathbf{x})M(\mathbf{x}) - \widehat{H}^{-1}(\mathbf{x})\overline{A}G(\mathbf{x})M(\mathbf{x})\right\| \\
&\quad + \left\|\widehat{H}^{-1}(\mathbf{x})\overline{A}G(\mathbf{x})M(\mathbf{x}) - \widehat{H}^{-1}(\mathbf{x})\overline{A}\widehat{G}(\mathbf{x})\widehat{M}(\mathbf{x})\right\| \\
&\leq \left\|H^{-1}(\mathbf{x}) - \widehat{H}^{-1}(\mathbf{x})\right\| \|\overline{A}\| \|G(\mathbf{x})\| \|M(\mathbf{x})\| \\
&\quad + \left\|\widehat{H}^{-1}(\mathbf{x})\right\| \|\overline{A}\| \left\|G(\mathbf{x})M(\mathbf{x}) - \widehat{G}(\mathbf{x})\widehat{M}(\mathbf{x})\right\| \\
&\stackrel{(b)}{\leq} \left\|H^{-1}(\mathbf{x}) - \widehat{H}^{-1}(\mathbf{x})\right\| \|\overline{A}\| \|G(\mathbf{x})\| \|M(\mathbf{x})\| \\
&\quad + \left\|\widehat{H}^{-1}(\mathbf{x})\right\| \|\overline{A}\| \left[\left\|G(\mathbf{x})M(\mathbf{x}) - G(\mathbf{x})\widehat{M}(\mathbf{x})\right\| + \left\|G(\mathbf{x})\widehat{M}(\mathbf{x}) - \widehat{G}(\mathbf{x})\widehat{M}(\mathbf{x})\right\|\right] \\
&\leq \left\|H^{-1}(\mathbf{x}) - \widehat{H}^{-1}(\mathbf{x})\right\| \|\overline{A}\| \|G(\mathbf{x})\| \|M(\mathbf{x})\| \\
&\quad + \left\|\widehat{H}^{-1}(\mathbf{x})\right\| \|\overline{A}\| \|G(\mathbf{x})\| \left\|M(\mathbf{x}) - \widehat{M}(\mathbf{x})\right\| + \left\|\widehat{H}^{-1}(\mathbf{x})\right\| \|\overline{A}\| \left\|G(\mathbf{x}) - \widehat{G}(\mathbf{x})\right\| \left\|\widehat{M}(\mathbf{x})\right\| \\
&\stackrel{(c)}{\leq} \overline{L}_A^2 L_A^2 \left(\frac{1}{\mu_g}\right)^2 L_{g_{yy}} \delta L_A \frac{1}{\mu_g} \overline{L}_{g_{xy}} + \overline{L}_A L_A \frac{1}{\mu_g} L_{g_{xy}} \delta + \overline{L}_A L_A \left(\frac{1}{\mu_g}\right)^2 L_{g_{yy}} \delta \overline{L}_{g_{xy}} \\
&= \left(\left(\frac{1}{\mu_g}\right)^3 \overline{L}_A^2 L_A^3 L_{g_{yy}} \overline{L}_{g_{xy}} + \frac{1}{\mu_g} \overline{L}_A L_A L_{g_{xy}} + \left(\frac{1}{\mu_g}\right)^2 \overline{L}_A L_A L_{g_{yy}} \overline{L}_{g_{xy}}\right) \delta.
\end{aligned}
$$

In (a) we add and subtract the term $\widehat{H}^{-1}(\mathbf{x})\overline{A}G(\mathbf{x})M(\mathbf{x})$ and apply the triangle inequality. In (b) we add and subtract the term $G(\mathbf{x})\widehat{M}(\mathbf{x})$ and apply the triangle inequality. In (c) we use Lemma 6(d) for $\|H^{-1}(\mathbf{x}) - \widehat{H}^{-1}(\mathbf{x})\|$, the bound $\|\overline{A}\| \leq L_A$, Lemma 6(a) for $\|G(\mathbf{x})\|$, Lemma 6(c) for $\|H^{-1}(\mathbf{x})\|$ and $\|\widehat{H}^{-1}(\mathbf{x})\|$, Assumption 3(f) for $\|M(\mathbf{x})\|$ and $\|\widehat{M}(\mathbf{x})\|$, Assumption 3(e) for $\|M(\mathbf{x}) - \widehat{M}(\mathbf{x})\|$, and finally Lemma 6(b) for $\|G(\mathbf{x}) - \widehat{G}(\mathbf{x})\|$.

The proof is now complete. $\qquad\square$

**Lemma 9.** *Suppose that Assumptions 1,2,3 hold. Then, the following bound holds*

$$
\|\nabla\mathbf{y}^*(\mathbf{x}) - \widehat{\nabla}\mathbf{y}(\mathbf{x})\| \leq L_{\mathbf{y}^*}\delta,
$$

*where* $L_{\mathbf{y}^*} = \left(\frac{1}{\mu_g}\right)^2 L_{g_{yy}} \overline{L}_{g_{xy}} + \frac{1}{\mu_g} L_{g_{xy}} + \left(\frac{1}{\mu_g}\right)^2 L_{g_{yy}} L_A \overline{L}_{\boldsymbol{\lambda}^*} + \frac{1}{\mu_g} L_A L_{\boldsymbol{\lambda}^*}.$

*Proof.* From Lemma 2 we have that

$$
\nabla\mathbf{y}^*(\mathbf{x}) = \left[\nabla_{yy}^2 g(\mathbf{x}, \mathbf{y}^*(\mathbf{x}))\right]^{-1}\left[-\nabla_{xy}^2 g(\mathbf{x}, \mathbf{y}^*(\mathbf{x})) - \overline{A}^T \nabla\overline{\boldsymbol{\lambda}}^*(\mathbf{x})\right].
$$

We can also get $\widehat{\nabla}\mathbf{y}(\mathbf{x})$ by substituting $\widehat{\mathbf{y}}(\mathbf{x})$ in place of $\mathbf{y}^*(\mathbf{x})$ in the above formula, i.e.,

$$
\widehat{\nabla}\mathbf{y}(\mathbf{x}) = \left[\nabla_{yy}^2 g(\mathbf{x}, \widehat{\mathbf{y}}(\mathbf{x}))\right]^{-1}\left[-\nabla_{xy}^2 g(\mathbf{x}, \widehat{\mathbf{y}}(\mathbf{x})) - \overline{A}^T \widehat{\nabla}\overline{\boldsymbol{\lambda}}^*(\mathbf{x})\right].
$$

Then, we have that

$$
\begin{aligned}
\left\|\nabla\mathbf{y}^*(\mathbf{x}) - \widehat{\nabla}\mathbf{y}(\mathbf{x})\right\| &= \left\| \left[\nabla_{yy}g(\mathbf{x},\mathbf{y}^*(\mathbf{x}))\right]^{-1}\left[-\nabla_{xy}g(\mathbf{x},\mathbf{y}^*(\mathbf{x})) - \overline{A}^T\nabla\overline{\boldsymbol{\lambda}}^*(\mathbf{x})\right]\right. \\
&\quad \left. - \left[\nabla_{yy}^2 g(\mathbf{x},\widehat{\mathbf{y}}(\mathbf{x}))\right]^{-1}\left[-\nabla_{xy}g(\mathbf{x},\widehat{\mathbf{y}}(\mathbf{x})) - \overline{A}^T\widehat{\nabla}\overline{\boldsymbol{\lambda}}^*(\mathbf{x})\right]\right\| \\[4pt]
&\overset{(a)}{\leq} \left\|\left[\nabla_{yy}^2 g(\mathbf{x},\mathbf{y}^*(\mathbf{x}))\right]^{-1}\nabla_{xy}g(\mathbf{x},\mathbf{y}^*(\mathbf{x})) - \left[\nabla_{yy}^2 g(\mathbf{x},\widehat{\mathbf{y}}(\mathbf{x}))\right]^{-1}\nabla_{xy}g(\mathbf{x},\widehat{\mathbf{y}}(\mathbf{x}))\right\| \\
&\quad + \left\|\left[\nabla_{yy}^2 g(\mathbf{x},\mathbf{y}^*(\mathbf{x}))\right]^{-1}\overline{A}^T\nabla\overline{\boldsymbol{\lambda}}^*(\mathbf{x}) - \left[\nabla_{yy}^2 g(\mathbf{x},\widehat{\mathbf{y}}(\mathbf{x}))\right]^{-1}\overline{A}^T\widehat{\nabla}\overline{\boldsymbol{\lambda}}^*(\mathbf{x})\right\| \\[4pt]
&\overset{(b)}{\leq} \left\|\left[\nabla_{yy}^2 g(\mathbf{x},\mathbf{y}^*(\mathbf{x}))\right]^{-1}\nabla_{xy}g(\mathbf{x},\mathbf{y}^*(\mathbf{x})) - \left[\nabla_{yy}g(\mathbf{x},\widehat{\mathbf{y}}(\mathbf{x}))\right]^{-1}\nabla_{xy}g(\mathbf{x},\mathbf{y}^*(\mathbf{x}))\right\| \\
&\quad + \left\|\left[\nabla_{yy}^2 g(\mathbf{x},\widehat{\mathbf{y}}(\mathbf{x}))\right]^{-1}\nabla_{xy}g(\mathbf{x},\mathbf{y}^*(\mathbf{x})) - \left[\nabla_{yy}^2 g(\mathbf{x},\widehat{\mathbf{y}}(\mathbf{x}))\right]^{-1}\nabla_{xy}g(\mathbf{x},\widehat{\mathbf{y}}(\mathbf{x}))\right\| \\
&\quad + \left\|\left[\nabla_{yy}^2 g(\mathbf{x},\mathbf{y}^*(\mathbf{x}))\right]^{-1}\overline{A}^T\nabla\overline{\boldsymbol{\lambda}}^*(\mathbf{x}) - \left[\nabla_{yy}^2 g(\mathbf{x},\widehat{\mathbf{y}}(\mathbf{x}))\right]^{-1}\overline{A}^T\nabla\overline{\boldsymbol{\lambda}}^*(\mathbf{x})\right\| \\
&\quad + \left\|\left[\nabla_{yy}^2 g(\mathbf{x},\widehat{\mathbf{y}}(\mathbf{x}))\right]^{-1}\overline{A}^T\nabla\overline{\boldsymbol{\lambda}}^*(\mathbf{x}) - \left[\nabla_{yy}^2 g(\mathbf{x},\widehat{\mathbf{y}}(\mathbf{x}))\right]^{-1}\overline{A}^T\widehat{\nabla}\overline{\boldsymbol{\lambda}}^*(\mathbf{x})\right\| \\[4pt]
&\leq \left\|\left[\nabla_{yy}^2 g(\mathbf{x},\mathbf{y}^*(\mathbf{x}))\right]^{-1} - \left[\nabla_{yy}^2 g(\mathbf{x},\widehat{\mathbf{y}}(\mathbf{x}))\right]^{-1}\right\|\|\nabla_{xy}g(\mathbf{x},\mathbf{y}^*(\mathbf{x}))\| \\
&\quad + \left\|\left[\nabla_{yy}^2 g(\mathbf{x},\widehat{\mathbf{y}}(\mathbf{x}))\right]^{-1}\right\|\|\nabla_{xy}g(\mathbf{x},\mathbf{y}^*(\mathbf{x})) - \nabla_{xy}g(\mathbf{x},\widehat{\mathbf{y}}(\mathbf{x}))\| \\
&\quad + \left\|\left[\nabla_{yy}^2 g(\mathbf{x},\mathbf{y}^*(\mathbf{x}))\right]^{-1} - \left[\nabla_{yy}^2 g(\mathbf{x},\widehat{\mathbf{y}}(\mathbf{x}))\right]^{-1}\right\|\left\|\overline{A}^T\right\|\left\|\nabla\overline{\boldsymbol{\lambda}}^*(\mathbf{x})\right\| \\
&\quad + \left\|\left[\nabla_{yy}^2 g(\mathbf{x},\widehat{\mathbf{y}}(\mathbf{x}))\right]^{-1}\right\|\left\|\overline{A}^T\right\|\left\|\nabla\overline{\boldsymbol{\lambda}}^*(\mathbf{x}) - \widehat{\nabla}\overline{\boldsymbol{\lambda}}^*(\mathbf{x})\right\| \\[4pt]
&\overset{(c)}{\leq} \left(\frac{1}{\mu_g}\right)^2 L_{g_{yy}}\delta\overline{L}_{g_{xy}} + \frac{1}{\mu_g}L_{g_{xy}}\delta + \left(\frac{1}{\mu_g}\right)^2 L_{g_{yy}}\delta L_A\overline{L}_{\boldsymbol{\lambda}^*} + \frac{1}{\mu_g}L_A L_{\boldsymbol{\lambda}^*}\delta \\[4pt]
&= \left(\left(\frac{1}{\mu_g}\right)^2 L_{g_{yy}}\overline{L}_{g_{xy}} + \frac{1}{\mu_g}L_{g_{xy}} + \left(\frac{1}{\mu_g}\right)^2 L_{g_{yy}}L_A\overline{L}_{\boldsymbol{\lambda}^*} + \frac{1}{\mu_g}L_A L_{\boldsymbol{\lambda}^*}\right)\delta.
\end{aligned}
$$

In (a) the triangle inequality was used. In (b) we add and subtract the expressions $\left[\nabla_{yy}g(\mathbf{x},\widehat{\mathbf{y}}(\mathbf{x}))\right]^{-1}\nabla_{xy}g(\mathbf{x},\mathbf{y}^*(\mathbf{x}))$ and $\left[\nabla_{yy}^2 g(\mathbf{x},\widehat{\mathbf{y}}(\mathbf{x}))\right]^{-1}\overline{A}^T\nabla\overline{\boldsymbol{\lambda}}^*(\mathbf{x})$ in the first and second terms, respectively. In (c) we apply Lemma 6(a), 6(b), Assumption 3(e), 3(f), the bound $\|\overline{A}\| \leq L_A$, Lemmas 7 and 8.

The proof is now complete. $\qquad\square$

Now we have all the results needed to prove Lemma 3.

*Proof of Lemma 3.* To begin with, the exact and approximate (due to the inexact solution of the LL problem) implicit gradients of the objective $F(\mathbf{x})$, are given below.

$$
\begin{aligned}
\nabla F(\mathbf{x}) &= \nabla_x f(\mathbf{x},\mathbf{y}^*(\mathbf{x})) + [\nabla\mathbf{y}^*(\mathbf{x})]^T \nabla_y f(\mathbf{x},\mathbf{y}^*(\mathbf{x})) \\
\widehat{\nabla}F(\mathbf{x}) &= \nabla_x f(\mathbf{x},\widehat{\mathbf{y}}(\mathbf{x})) + \left[\widehat{\nabla}\mathbf{y}(\mathbf{x})\right]^T \nabla_y f(\mathbf{x},\widehat{\mathbf{y}}(\mathbf{x})).
\end{aligned}
$$

Then, we can compute the norm of their difference.

$$
\begin{aligned}
\|\widehat{\nabla}F(\mathbf{x}) - \nabla F(\mathbf{x})\| &= \|\nabla_x f(\mathbf{x},\widehat{\mathbf{y}}(\mathbf{x})) + [\nabla\widehat{\mathbf{y}}(\mathbf{x})]^T \nabla_y f(\mathbf{x},\widehat{\mathbf{y}}(\mathbf{x})) \\
&\qquad - \nabla_x f(\mathbf{x},\mathbf{y}^*(\mathbf{x})) - [\nabla\mathbf{y}^*(\mathbf{x})]^T \nabla_y f(\mathbf{x},\mathbf{y}^*(\mathbf{x}))\| \\[4pt]
&\overset{(a)}{\leq} \|\nabla_x f(\mathbf{x},\widehat{\mathbf{y}}(\mathbf{x})) - \nabla_x f(\mathbf{x},\mathbf{y}^*(\mathbf{x}))\| \\
&\qquad + \|[\widehat{\nabla}\mathbf{y}(\mathbf{x})]^T\nabla_y f(\mathbf{x},\widehat{\mathbf{y}}(\mathbf{x})) - [\nabla\mathbf{y}^*(\mathbf{x})]^T \nabla_y f(\mathbf{x},\mathbf{y}^*(\mathbf{x}))\| \\[4pt]
&\overset{(b)}{\leq} \|\nabla_x f(\mathbf{x},\widehat{\mathbf{y}}(\mathbf{x})) - \nabla_x f(\mathbf{x},\mathbf{y}^*(\mathbf{x}))\| \\
&\qquad + \|[\widehat{\nabla}\mathbf{y}(\mathbf{x})]^T\nabla_y f(\mathbf{x},\widehat{\mathbf{y}}(\mathbf{x})) - [\nabla\mathbf{y}^*(\mathbf{x})]^T \nabla_y f(\mathbf{x},\widehat{\mathbf{y}}(\mathbf{x}))\| \\
&\qquad + \| [\nabla\mathbf{y}^*(\mathbf{x})]^T \nabla_y f(\mathbf{x},\widehat{\mathbf{y}}(\mathbf{x})) - [\nabla\mathbf{y}^*(\mathbf{x})]^T \nabla_y f(\mathbf{x},\mathbf{y}^*(\mathbf{x}))\| \\[4pt]
&\overset{(c)}{\leq} L_f\|\widehat{\mathbf{y}}(\mathbf{x}) - \mathbf{y}^*(\mathbf{x}))\| + \|\widehat{\nabla}\mathbf{y}(\mathbf{x}) - \nabla\mathbf{y}^*(\mathbf{x}))\|\|\nabla_y f(\mathbf{x},\widehat{\mathbf{y}}(\mathbf{x}))\| \\
&\qquad + \|\nabla\mathbf{y}^*(\mathbf{x}))\|\|\nabla_y f(\mathbf{x},\widehat{\mathbf{y}}(\mathbf{x})) - \nabla_y f(\mathbf{x},\mathbf{y}^*(\mathbf{x}))\| \\[4pt]
&\overset{(d)}{\leq} L_f\|\widehat{\mathbf{y}}(\mathbf{x}) - \mathbf{y}^*(\mathbf{x}))\| + \|\widehat{\nabla}\mathbf{y}(\mathbf{x}) - \nabla\mathbf{y}^*(\mathbf{x}))\|\|\nabla_y f(\mathbf{x},\widehat{\mathbf{y}}(\mathbf{x}))\| \\
&\qquad + L_f\|\nabla\mathbf{y}^*(\mathbf{x}))\|\|\widehat{\mathbf{y}}(\mathbf{x}) - \mathbf{y}^*(\mathbf{x}))\| \\[4pt]
&\overset{(e)}{\leq} L_f\delta + L_{\mathbf{y}^*}\delta\overline{L}_f + L_f\overline{L}_{\mathbf{y}^*}\delta \\
&= \left(L_f + L_{\mathbf{y}^*}\overline{L}_f + L_f\overline{L}_{\mathbf{y}^*}\right)\delta := L_F\delta, \qquad\qquad (26)
\end{aligned}
$$

where $L_F = L_f + L_{\mathbf{y}^*}\overline{L}_f + L_f\overline{L}_{\mathbf{y}^*}$. Also, in inequality (a) above we apply the triangle inequality; in (b) we add and subtract the term $[\nabla\mathbf{y}^*(\mathbf{x})]^T \nabla_y f(\mathbf{x},\widehat{\mathbf{y}}(\mathbf{x}))$, and use triangle inequality; in (c) and (d) we use the Lipschitz gradient property of $f$ (Assumption 2(b)); in (e) we apply Assumptions 3(a) and 2(a), and Lemmas 7 and 9.

Now consider the expression $\|\nabla F(\mathbf{x})\|$. We have that

$$
\begin{aligned}
\|\nabla F(\mathbf{x})\| &= \left\|\nabla_x f(\mathbf{x},\mathbf{y}^*(\mathbf{x})) + [\nabla\mathbf{y}^*(\mathbf{x})]^T \nabla_y f(\mathbf{x},\mathbf{y}^*(\mathbf{x}))\right\| \\
&\leq |\nabla_x f(\mathbf{x},\mathbf{y}^*(\mathbf{x}))\| + \|\nabla\mathbf{y}^*(\mathbf{x})\| \|\nabla_y f(\mathbf{x},\mathbf{y}^*(\mathbf{x}))\| \\
&\leq \left(1 + \overline{L}_{\mathbf{y}^*}\right)\overline{L}_f := \overline{L}_F,
\end{aligned}
$$

where we applied Assumption 3(a) and Lemma 7. Similarly, we can see that

$$
\begin{aligned}
\left\|\widehat{\nabla}F(\mathbf{x})\right\| &= \left\|\nabla_x f(\mathbf{x},\widehat{\mathbf{y}}(\mathbf{x})) + \left[\widehat{\nabla}\mathbf{y}(\mathbf{x})\right]^T \nabla_y f(\mathbf{x},\widehat{\mathbf{y}}(\mathbf{x}))\right\| \\
&\leq \|\nabla_x f(\mathbf{x},\widehat{\mathbf{y}}(\mathbf{x}))\| + \left\|\widehat{\nabla}\mathbf{y}(\mathbf{x})\right\| \|\nabla_y f(\mathbf{x},\widehat{\mathbf{y}}(\mathbf{x}))\| \\
&\leq \left(1 + \overline{L}_{\mathbf{y}^*}\right)\overline{L}_f = \overline{L}_F.
\end{aligned}
$$

Therefore, the proof is completed. $\qquad\qquad\square$

### D.1.4 PROOF OF LEMMA 4

*Proof.* From the definition of the stochastic gradient in (10) we have

$$
\widehat{\nabla}F(\mathbf{x};\xi) = \nabla_x f(\mathbf{x},\widehat{\mathbf{y}}(\mathbf{x});\xi) + [\widehat{\nabla}\mathbf{y}^*(\mathbf{x})]^T\nabla_y f(\mathbf{x},\widehat{\mathbf{y}}(\mathbf{x});\xi).
$$

Taking expectation on both sides and utilizing Assumption 4, we get

$$
\begin{aligned}
\mathbb{E}_\xi[\widehat{\nabla}F(\mathbf{x};\xi)] &= \mathbb{E}_\xi\left[\nabla_x f(\mathbf{x},\widehat{\mathbf{y}}(\mathbf{x});\xi) + [\widehat{\nabla}\mathbf{y}^*(\mathbf{x})]^T\nabla_y f(\mathbf{x},\widehat{\mathbf{y}}(\mathbf{x});\xi)\right] \\
&= \mathbb{E}_\xi\left[\nabla_x f(\mathbf{x},\widehat{\mathbf{y}}(\mathbf{x});\xi)\right] + [\widehat{\nabla}\mathbf{y}^*(\mathbf{x})]^T\mathbb{E}_\xi\left[\nabla_y f(\mathbf{x},\widehat{\mathbf{y}}(\mathbf{x});\xi)\right] \\
&= \nabla_x f(\mathbf{x},\widehat{\mathbf{y}}(\mathbf{x});\xi) + [\widehat{\nabla}\mathbf{y}^*(\mathbf{x})]^T\nabla_y f(\mathbf{x},\widehat{\mathbf{y}}(\mathbf{x});\xi) \\
&= \widehat{\nabla}F(\mathbf{x}).
\end{aligned}
$$

Similarly, for the variance of the stochastic implicit gradient, we have

$$
\begin{aligned}
\mathbb{E}_\xi \|\widehat{\nabla} F(\mathbf{x};\xi) - \widehat{\nabla} F(\mathbf{x})\|^2 &= \mathbb{E}_\xi \big\| \nabla_x f(\mathbf{x},\widehat{\mathbf{y}}(\mathbf{x});\xi) + [\widehat{\nabla}\mathbf{y}^*(\mathbf{x})]^T \nabla_y f(\mathbf{x},\widehat{\mathbf{y}}(\mathbf{x});\xi) \\
&\qquad - \big[ \nabla_x f(\mathbf{x},\widehat{\mathbf{y}}(\mathbf{x})) + [\widehat{\nabla}\mathbf{y}^*(\mathbf{x})]^T \nabla_y f(\mathbf{x},\widehat{\mathbf{y}}(\mathbf{x})) \big] \big\|^2 \\
&\overset{(a)}{\le} 2\,\mathbb{E}_\xi \| \nabla_x f(\mathbf{x},\widehat{\mathbf{y}}(\mathbf{x});\xi) - \nabla_x f(\mathbf{x},\widehat{\mathbf{y}}(\mathbf{x})) \|^2 \\
&\qquad + 2 \, \| \widehat{\nabla}\mathbf{y}^*(\mathbf{x}) \|^2 \, \mathbb{E}_\xi \| \nabla_y f(\mathbf{x},\widehat{\mathbf{y}}(\mathbf{x});\xi) - \nabla_y f(\mathbf{x},\widehat{\mathbf{y}}(\mathbf{x})) \|^2 \\
&\overset{(b)}{\le} 2\sigma_f^2 + 2\overline{L}_{\mathbf{y}^*} \sigma_f^2 := \sigma_F^2,
\end{aligned}
$$

where $(a)$ follows from $\|\mathbf{x} + \mathbf{y}\|^2 \le 2\|\mathbf{x}\|^2 + 2\|\mathbf{y}\|^2$ and $(b)$ results from Assumption 4 and the application of Lemma 7.

Therefore, we have the proof. $\qquad\square$

## D.2 Proofs of Section 3

### D.2.1 Proof of Lemma 5

*Proof.* Let $\{\mathbf{x}^r\}_{r=0}^\infty$ with $\mathbf{x}^r \in \mathcal{X}$ be a given arbitrary countable sequence of points. Lemma 1 (adapted from (Lu et al., 2020, Proposition 1)) states that for any given point $\mathbf{x}^r$ in the above sequence, the SC condition holds w.p. 1 for the LL problem in (12), assuming that $\mathbf{q}$ is generated from a continuous measure and $\overline{A}(\mathbf{y}^*(\mathbf{x}^r))$ is full row rank. This further implies that the mapping $\mathbf{y}^*(\mathbf{x})$, and thereby, the implicit function, $F(\mathbf{x})$ is differentiable w.p. 1 for each given $\mathbf{x}^r$ in the above sequence (please see the discussion after Lemma 1), i.e., we have

$$
\mathbb{P}(F(\mathbf{x}) \text{ is differentiable at } \mathbf{x}^r) = 1 \ \text{ for each } r = \{0, 1, \dots, \infty\}. \tag{27}
$$

This further implies that we have

$$
\begin{aligned}
\mathbb{P}\Big( F(\mathbf{x}) \text{ is differentiable for all } \{\mathbf{x}^r\}_{r=0}^\infty \Big) &= \mathbb{P}\Big( \bigcap_{r=0}^\infty \{F(\mathbf{x}) \text{ is differentiable at } \mathbf{x}^r\} \Big) \\
&= 1 - \mathbb{P}\Big( \bigcup_{r=0}^\infty \{F(\mathbf{x}) \text{ is non-differentiable at } \mathbf{x}^r\} \Big) \\
&\ge 1 - \sum_{r=0}^\infty \mathbb{P}\left( \{F(\mathbf{x}) \text{ is non-differentiable at } \mathbf{x}^r\} \right) \\
&= 1.
\end{aligned}
$$

where the second equality follows from the fact that $\mathbb{P}(\omega \in A) = 1 - \mathbb{P}(\omega \in A^c)$ where $A^c$ denotes the complement of a measurable event $A$; the inequality uses the union bound; and the final equality utilizes (27) above. $\qquad\square$

### D.2.2 Proof of Proposition 2

*Proof.* From Assumption 1 we know that $h(\mathbf{x}, \mathbf{y})$ (and thus $g(\mathbf{x}, \mathbf{y})$) is strongly convex in $\mathbf{y}$ with modulus $\mu_g = \mu_h$. As a result we have that

$$
h(\mathbf{x}, \mathbf{y}^*(\mathbf{x})) \ge h(\mathbf{x}, \overline{\mathbf{y}}^*(\mathbf{x})) + \langle \nabla_y h(\mathbf{x}, \overline{\mathbf{y}}^*(\mathbf{x})), \mathbf{y}^*(\mathbf{x}) - \overline{\mathbf{y}}^*(\mathbf{x}) \rangle + \frac{\mu_g}{2} \|\mathbf{y}^*(\mathbf{x}) - \overline{\mathbf{y}}^*(\mathbf{x})\|^2 \tag{28}
$$

$$
g(\mathbf{x}, \overline{\mathbf{y}}^*(\mathbf{x})) \ge g(\mathbf{x}, \mathbf{y}^*(\mathbf{x})) + \langle \nabla_y g(\mathbf{x}, \mathbf{y}^*(\mathbf{x})), \overline{\mathbf{y}}^*(\mathbf{x}) - \mathbf{y}^*(\mathbf{x}) \rangle + \frac{\mu_g}{2} \|\overline{\mathbf{y}}^*(\mathbf{x}) - \mathbf{y}^*(\mathbf{x})\|^2. \tag{29}
$$

By definition $\overline{\mathbf{y}}^*(\mathbf{x})$ is the global minimum of the objective $h(\mathbf{x}, \mathbf{y})$, and so it holds that $\langle \nabla_y h(\mathbf{x}, \overline{\mathbf{y}}^*(\mathbf{x})), \mathbf{y}^*(\mathbf{x}) - \overline{\mathbf{y}}^*(\mathbf{x}) \rangle \ge 0$. Similarly, $\mathbf{y}^*(\mathbf{x})$ is the global minimum of the objective $g(\mathbf{x}, \mathbf{y})$, and so it holds that $\langle \nabla_y g(\mathbf{x}, \mathbf{y}^*(\mathbf{x})), \overline{\mathbf{y}}^*(\mathbf{x}) - \mathbf{y}^*(\mathbf{x}) \rangle \ge 0$. Then, using the above inequali-

ties and adding (28) and (29), we get

$$h(\mathbf{x}, \mathbf{y}^*(\mathbf{x})) + g(\mathbf{x}, \overline{\mathbf{y}}^*(\mathbf{x})) \geq h(\mathbf{x}, \overline{\mathbf{y}}^*(\mathbf{x})) + g(\mathbf{x}, \mathbf{y}^*(\mathbf{x})) + \mu_g \|\overline{\mathbf{y}}^*(\mathbf{x}) - \mathbf{y}^*(\mathbf{x})\|^2$$

$$\mu_g \|\overline{\mathbf{y}}^*(\mathbf{x}) - \mathbf{y}^*(\mathbf{x})\|^2 \leq [h(\mathbf{x}, \mathbf{y}^*(\mathbf{x})) - g(\mathbf{x}, \mathbf{y}^*(\mathbf{x}))] + [g(\mathbf{x}, \overline{\mathbf{y}}^*(\mathbf{x})) - h(\mathbf{x}, \overline{\mathbf{y}}^*(\mathbf{x}))]$$

$$\mu_g \|\overline{\mathbf{y}}^*(\mathbf{x}) - \mathbf{y}^*(\mathbf{x})\|^2 \leq -\mathbf{q}^T \mathbf{y}^*(\mathbf{x}) + \mathbf{q}^T \overline{\mathbf{y}}^*(\mathbf{x})$$

$$\|\overline{\mathbf{y}}^*(\mathbf{x}) - \mathbf{y}^*(\mathbf{x})\|^2 \leq \frac{\mathbf{q}^T \left(\overline{\mathbf{y}}^*(\mathbf{x}) - \mathbf{y}^*(\mathbf{x})\right)}{\mu_g}$$

$$\|\overline{\mathbf{y}}^*(\mathbf{x}) - \mathbf{y}^*(\mathbf{x})\|^2 \leq \frac{\|\mathbf{q}^T\| \|\overline{\mathbf{y}}^*(\mathbf{x}) - \mathbf{y}^*(\mathbf{x})\|}{\mu_g}$$

$$\|\overline{\mathbf{y}}^*(\mathbf{x}) - \mathbf{y}^*(\mathbf{x})\| \leq \frac{\|\mathbf{q}\|}{\mu_g}.$$

Using the above bound and the fact that $f$ is Lipschitz continuous (it follows from the bounded gradient assumption 3(a)) it is easy to see that

$$|F(\mathbf{x}) - G(\mathbf{x})| = |f(\mathbf{x}, \mathbf{y}^*(\mathbf{x})) - f(\mathbf{x}, \overline{\mathbf{y}}^*(\mathbf{x}))| \leq \overline{L}_f \|\mathbf{y}^*(\mathbf{x}) - \overline{\mathbf{y}}^*(\mathbf{x})\| \leq \overline{L}_f \frac{\|\mathbf{q}\|}{\mu_g}.$$

Therefore, the proof is complete. □

### D.2.3 PROOF OF THEOREM 1

**Lemma 10.** *Under Assumption 1, 2, 3, $\nabla F(\mathbf{x})$ is almost surely continuous at a neighborhood around $x$, for any given $\mathbf{x} \in \mathcal{X}$.*

*Proof.* To begin with, we already established in Lemma 2 that $F$ is almost surely differentiable at any given $\mathbf{x} \in \mathcal{X}$. Therefore, for any $\mathbf{x} \in \mathcal{X}$ there exists (almost surely) a neighborhood around it such that the matrix $\overline{A}$ corresponding to the active constraints at $\mathbf{y}^*(\mathbf{x})$ remains unchanged, where the gradient $\nabla \mathbf{y}^*(\mathbf{x})$ is defined in eq. (6), (7). Further, since $\overline{A}$ is locally (i.e., around any given $\mathbf{x}$) constant, and the formulas in (6), (7) can be seen as the results of a number of continuous operations over continuous functions, it is implied that $\nabla \mathbf{y}^*(\mathbf{x})$ is also a continuous function at a neighborhood around $\mathbf{x}$ almost surely. As a result, $\nabla F(\mathbf{x}) = \nabla_x f(\mathbf{x}, \mathbf{y}^*(\mathbf{x})) + [\nabla \mathbf{y}^*(\mathbf{x})]^T \nabla_y f(\mathbf{x}, \mathbf{y}^*(\mathbf{x}))$ is almost surely continuous locally around any given $\mathbf{x} \in \mathcal{X}$. □

*Proof of Theorem 1.* Here we follow a reasoning similar to the proof of (Bertsekas, 1998, Prop. 1.2.1). However, there are a number of differences that make this proof more challenging. First, in our setting we are optimizing an inexact version of the objective $\widehat{F}(\mathbf{x}) = f(\mathbf{x}, \widehat{\mathbf{y}}(\mathbf{x}))$, using an approximate version of the gradient $\widehat{\nabla} F(\mathbf{x})$. Since the approximate gradient we are using is not the gradient of the objective $\widehat{F}(\mathbf{x})$ (the gradient of this function might not even exist), we consider a modification of the standard Armijo rule where an additional error term is present. Secondly, in the proof below the (classical) mean value theorem (Bertsekas, 1998, Prop. 1.23) cannot be applied, since we cannot ensure that $F$ is (surely) differentiable at any given interval over $\mathbf{x}$. As we are going to show below, we use an alternative mean value theorem that does not require such assumption.

To begin with, we know that for the exact implicit objective $F(\mathbf{x})$ and gradient $\nabla F(\mathbf{x})$ (quantities to which we do not have access to) we can find at each iteration $r$ a step-size $a^r$ such that the following condition holds

$$F(\mathbf{x}^r) - F(\mathbf{x}^r + a^r \mathbf{d}^r) \geq -\sigma a^r [\nabla F(\mathbf{x}^r)]^T \mathbf{d}^r, \tag{30}$$

where $\mathbf{d}^r = \widetilde{\mathbf{x}}^r - \mathbf{x}^r$ with $\widetilde{\mathbf{x}}^r = \text{proj}_{\mathcal{X}}(\mathbf{x}^r - \nabla F(\mathbf{x}^r))$.

Next, the difference between the (approximate) objective values of two successive iterates (for simplicity we will use the notation $\mathbf{x}^{r+1} = \mathbf{x}^r + a^r \mathbf{d}^r$, $\widehat{\mathbf{x}}^{r+1} = \mathbf{x}^r + a^r \widehat{\mathbf{d}}^r$; $\widehat{\mathbf{d}}^r$ is defined in

Algorithm 1) is

$$
\begin{aligned}
\widehat{F}(\mathbf{x}^r) - \widehat{F}(\widehat{\mathbf{x}}^{r+1}) &= \widehat{F}(\mathbf{x}^r) - F(\mathbf{x}^r) + F(\mathbf{x}^r) - F(\mathbf{x}^{r+1}) + F(\mathbf{x}^{r+1}) - F(\widehat{\mathbf{x}}^{r+1}) + F(\widehat{\mathbf{x}}^{r+1}) - \widehat{F}(\widehat{\mathbf{x}}^{r+1}) \\
&= f(\mathbf{x}^r, \widehat{\mathbf{y}}(\mathbf{x}^r)) - f(\mathbf{x}^r, \mathbf{y}^*(\mathbf{x}^r)) + F(\mathbf{x}^r) - F(\mathbf{x}^{r+1}) + F(\mathbf{x}^{r+1}) - F(\widehat{\mathbf{x}}^{r+1}) \\
&\qquad + f(\widehat{\mathbf{x}}^{r+1}, \mathbf{y}^*(\widehat{\mathbf{x}}^{r+1})) - f(\widehat{\mathbf{x}}^{r+1}, \widehat{\mathbf{y}}(\widehat{\mathbf{x}}^{r+1})) \\
&\geq -L_f \|\mathbf{y}^*(\mathbf{x}^r) - \widehat{\mathbf{y}}(\mathbf{x}^r)\| + F(\mathbf{x}^r) - F(\mathbf{x}^{r+1}) - \overline{L}_F \|\mathbf{x}^{r+1} - \widehat{\mathbf{x}}^{r+1}\| \\
&\qquad - L_f \|\mathbf{y}^*(\mathbf{x}^{r+1}) - \widehat{\mathbf{y}}(\mathbf{x}^{r+1})\| \\
&\geq -L_f \delta^r - \sigma a^r \left[\nabla F(\mathbf{x}^r)\right]^T \mathbf{d}^r - \overline{L}_F a^r \|\mathbf{d}^r - \widehat{\mathbf{d}}^r\| - L_f \delta^{r+1} \\
&\geq -L_f \delta^r - \sigma a^r \left[\nabla F(\mathbf{x}^r)\right]^T \mathbf{d}^r - \overline{L}_F L_F a^r \delta^r - L_f \delta^{r+1} \\
&= -\sigma a^r \left[\nabla F(\mathbf{x}^r)\right]^T \mathbf{d}^r - \epsilon_1(\delta; r),
\end{aligned}
\tag{31}
$$

where we set $\epsilon_1(\delta; r) = L_f \delta^r + \overline{L}_F L_F a^r \delta^r + L_f \delta^{r+1}$. In the first inequality, we used the Lipschitz continuity of $f$ and $F$; in the second inequality Assumption 2(a) and condition (30) were applied; in the third inequality the non-expansive property of the projection operator was used.

Also, we have that

$$
\begin{aligned}
\nabla^T F(\mathbf{x}^r)\mathbf{d}^r &= \left(\nabla F(\mathbf{x}^r) - \widehat{\nabla} F(\mathbf{x}^r) + \widehat{\nabla} F(\mathbf{x}^r)\right)^T \left(\mathbf{d}^r + \widehat{\mathbf{d}}^r - \widehat{\mathbf{d}}^r\right) \\
&= \left(\nabla F(\mathbf{x}^r) - \widehat{\nabla} F(\mathbf{x}^r)\right)^T \left(\mathbf{d}^r - \widehat{\mathbf{d}}^r\right) + \left(\nabla F(\mathbf{x}^r) - \widehat{\nabla} F(\mathbf{x}^r)\right)^T \widehat{\mathbf{d}}^r \\
&\quad + \left[\widehat{\nabla} F(\mathbf{x}^r)\right]^T \left(\mathbf{d}^r - \widehat{\mathbf{d}}^r\right) + \left[\widehat{\nabla} F(\mathbf{x}^r)\right]^T \widehat{\mathbf{d}}^r \\
&\leq \|\nabla F(\mathbf{x}^r) - \widehat{\nabla} F(\mathbf{x}^r)\| \|\mathbf{d}^r - \widehat{\mathbf{d}}^r\| + \|\nabla F(\mathbf{x}^r) - \widehat{\nabla} F(\mathbf{x}^r)\| \|\widehat{\mathbf{d}}^r\| \\
&\quad + \|\widehat{\nabla} F(\mathbf{x}^r)\| \|\mathbf{d}^r - \widehat{\mathbf{d}}^r\| + \left[\widehat{\nabla} F(\mathbf{x}^r)\right]^T \widehat{\mathbf{d}}^r \\
&\leq L_F^2 \left(\delta^r\right)^2 + L_F \overline{L}_F \delta^r + L_F \overline{L}_F \delta^r + \left[\widehat{\nabla} F(\mathbf{x}^r)\right]^T \widehat{\mathbf{d}}^r \\
&= \left[\widehat{\nabla} F(\mathbf{x}^r)\right]^T \widehat{\mathbf{d}}^r + \epsilon_2(\delta; r),
\end{aligned}
\tag{32}
$$

where $\epsilon_2(\delta; r) = L_F^2 \left(\delta^r\right)^2 + 2L_F \overline{L}_F \delta^r$. Notice that the results in the second inequality follow from Lemma 3.

Then, combining (31) and (32) we get

$$
\begin{aligned}
\widehat{F}(\mathbf{x}^r) - \widehat{F}(\widehat{\mathbf{x}}^{r+1}) &\geq -\sigma a^r \left[\widehat{\nabla} F(\mathbf{x}^r)\right]^T \widehat{\mathbf{d}}^r - \epsilon_1(\delta; r) - \sigma a^r \epsilon_2(\delta; r) \\
&= -\sigma a^r \left[\widehat{\nabla} F(\mathbf{x}^r)\right]^T \widehat{\mathbf{d}}^r - \epsilon(\delta; r),
\end{aligned}
$$

where $\epsilon(\delta; r) = \epsilon_1(\delta; r) + \sigma a^r \epsilon_2(\delta; r)$; notice that $\lim_{\delta \to 0} \epsilon(\delta) = 0$. In conclusion, we can follow this (inexact) Armijo-type rule in our inexact problem; the existence of the (Armijo) step-size is guaranteed by its existence for the exact problem (30).

Now let us move to the main part of the proof, which follows the reasoning used in (Bertsekas, 1998, Prop. 1.2.1). Let $\{\mathbf{x}^r\} \in \mathcal{X}$ be the iterate sequence of our algorithm, and let $\bar{\mathbf{x}} \in \mathcal{X}$ be a limit point; the existence of such point is guaranteed by the closedness of the set $\mathcal{X}$. Moreover, it is established in Proposition 1 that $F(\mathbf{x})$ is continuous, and as a result it holds that $\lim_{r \to +\infty} F(\mathbf{x}^r) = F(\bar{\mathbf{x}})$. The latter results combined with the fact that all convergent sequences are also Cauchy sequences, implies that $\lim_{r \to +\infty} (F(\mathbf{x}^r) - F(\mathbf{x}^{r+1})) = 0$.

We want to show that $\bar{\mathbf{x}}$ is a stationary point of $F(\mathbf{x})$. We are going to show that by assuming that the opposite holds, i.e., $\bar{\mathbf{x}}$ is not a stationary point of $F(\mathbf{x})$, and arriving at a contradiction. From the Armijo rule of our problem we have that

$$
\widehat{F}(\mathbf{x}^r) - \widehat{F}(\widehat{\mathbf{x}}^{r+1}) \geq -\sigma a^r \left[\widehat{\nabla} F(\mathbf{x}^r)\right]^T \widehat{\mathbf{d}}^r - \epsilon(\delta; r).
\tag{33}
$$

Then, consider the following

$$
\begin{aligned}
\widehat{F}(\mathbf{x}^r) - \widehat{F}(\widehat{\mathbf{x}}^{r+1}) &= \widehat{F}(\mathbf{x}^r) - F(\mathbf{x}^r) + F(\mathbf{x}^r) - F(\mathbf{x}^{r+1}) + F(\mathbf{x}^{r+1}) - F(\widehat{\mathbf{x}}^{r+1}) + F(\widehat{\mathbf{x}}^{r+1}) - \widehat{F}(\widehat{\mathbf{x}}^{r+1}) \\
&= f(\mathbf{x}^r, \widehat{\mathbf{y}}(\mathbf{x}^r)) - f(\mathbf{x}^r, \mathbf{y}^*(\mathbf{x}^r)) + F(\mathbf{x}^r) - F(\mathbf{x}^{r+1}) + F(\mathbf{x}^{r+1}) - F(\widehat{\mathbf{x}}^{r+1}) \\
&\quad + f(\widehat{\mathbf{x}}^{r+1}, \mathbf{y}^*(\widehat{\mathbf{x}}^{r+1})) - f(\widehat{\mathbf{x}}^{r+1}, \widehat{\mathbf{y}}(\widehat{\mathbf{x}}^{r+1})) \\
&\leq L_f \|\mathbf{y}^*(\mathbf{x}^r) - \widehat{\mathbf{y}}(\mathbf{x}^r)\| + F(\mathbf{x}^r) - F(\mathbf{x}^{r+1}) + \overline{L}_F \|\mathbf{x}^{r+1} - \widehat{\mathbf{x}}^{r+1}\| \\
&\quad\quad\quad\quad\quad\quad\quad\quad\quad\quad\quad\quad + L_f \|\mathbf{y}^*(\mathbf{x}^{r+1}) - \widehat{\mathbf{y}}(\mathbf{x}^{r+1})\| \\
&\leq L_f \delta^r + F(\mathbf{x}^r) - F(\mathbf{x}^{r+1}) + \overline{L}_F a^r \|\mathbf{d}^r - \widehat{\mathbf{d}}^r\| + L_f \delta^{r+1} \\
&\leq F(\mathbf{x}^r) - F(\mathbf{x}^{r+1}) + L_f \delta^r + \overline{L}_F L_F a^r \delta^r + L_f \delta^{r+1} \\
&= F(\mathbf{x}^r) - F(\mathbf{x}^{r+1}) + \epsilon_1(\delta; r).
\end{aligned}
$$

In the first inequality, we used the Lipschitz continuity of $f$ and $F$; in the second inequality Assumption 2(a) and condition (30) were applied; in the third inequality the non-expansive property of the projection operator was used. Using the above derivation we can bound the left-hand side of inequality (33) as follows

$$
F(\mathbf{x}^r) - F(\mathbf{x}^{r+1}) + \epsilon_1(\delta; r) \geq \widehat{F}(\mathbf{x}^r) - \widehat{F}(\widehat{\mathbf{x}}^{r+1}) \geq -\sigma a^r \left[\widehat{\nabla} F(\mathbf{x}^r)\right]^T \widehat{\mathbf{d}}^r - \epsilon(\delta; r).
$$

It is easy to see that the left-hand side in the above inequality tends to $0$. Therefore, $\lim_{r \to +\infty} \left(-\sigma a^r \left[\widehat{\nabla} F(\mathbf{x}^r)\right]^T \widehat{\mathbf{d}}^r - \epsilon(\delta; r)\right) \leq 0$. In addition, we know that $\lim_{r \to +\infty} \epsilon(\delta; r) = 0$ and $-\sigma a^r \left[\widehat{\nabla} F(\mathbf{x})\right]^T \widehat{\mathbf{d}}^r \geq 0, \forall \mathbf{x} \in \mathcal{X}$. From the above statements we can conclude that

$$
\lim_{r \to +\infty} \sigma a^r \left[\widehat{\nabla} F(\mathbf{x}^r)\right]^T \widehat{\mathbf{d}}^r = 0. \tag{34}
$$

Moreover, from the gradient-related assumption we know that for a non-stationary point $\bar{\mathbf{x}}$ we have that

$$
\limsup_{r \to \infty, r \in \mathcal{R}} \left[\widehat{\nabla} F(\mathbf{x}^r)\right]^T \widehat{\mathbf{d}}^r < 0, \tag{35}
$$

where $\{\mathbf{x}^r\}_{\mathcal{R}}$ is subsequence with $\lim_{r \to \infty, r \in \mathcal{R}} \mathbf{x}^r = \bar{\mathbf{x}}$. Then, the conditions (34), (35) imply that

$$
\lim_{r \to \infty, r \in \mathcal{R}} a^r = 0.
$$

In the subsequence $\mathcal{R}$ we can find an index $\bar{r} \geq 0$ such that

$$
\widehat{F}(\mathbf{x}^r) - \widehat{F}\left(\mathbf{x}^r + \left(\frac{a^r}{\beta}\right) \widehat{\mathbf{d}}^r\right) < -\sigma \left(\frac{a^r}{\beta}\right) \widehat{\nabla}^T F(\mathbf{x}^r) \widehat{\mathbf{d}}^r - \epsilon(\delta; r), \forall r \in \mathcal{R}, r \geq \bar{r}. \tag{36}
$$

Similarly with the proof of (Bertsekas, 1998, Prop. 1.2.1) let us introduce the following sequences:

$$
\widehat{\mathbf{p}}^r = \frac{\widehat{\mathbf{d}}^r}{\|\widehat{\mathbf{d}}^r\|}, \bar{a}^r = \frac{a^r \|\widehat{\mathbf{d}}^r\|}{\beta}
$$

The first sequence $\{\widehat{\mathbf{p}}^r\}$ is bounded and so it admits a limit point $\bar{\mathbf{p}}$ with $\|\bar{\mathbf{p}}\| = 1$, that is $\lim_{r \to +\infty, r \in \overline{\mathcal{R}}} \mathbf{p}^r = \bar{\mathbf{p}}$, where $\overline{\mathcal{R}}$ denotes the indices of a subsequence of $\mathcal{R}$. In addition, taking into account the facts that $\lim_{r \to +\infty, r \in \mathcal{R}} a^r = 0$ and the fact that the sequence $\{\|\mathbf{d}^r\|\}_{\mathcal{R}}$ is bounded we can easily see that $\lim_{r \to +\infty, r \in \mathcal{R}} \bar{a}^r = 0$.

Dividing both sides of (36) by $\bar{a}^r$ and using the definitions of $\widehat{\mathbf{p}}^r$ and $\bar{a}_r$ from above we get

$$
\frac{\widehat{F}(\mathbf{x}^r) - \widehat{F}\left(\mathbf{x}^r + \bar{a}^r \widehat{\mathbf{p}}^r\right)}{\bar{a}^r} < -\sigma \left[\widehat{\nabla} F(\mathbf{x}^r)\right]^T \widehat{\mathbf{p}}^r - \epsilon(\delta; r), \forall r \in \overline{\mathcal{R}}, r > \bar{r}. \tag{37}
$$

Then, we have that (for convenience we adopt the notation $\widehat{\mathbf{x}}^{r+1} = \mathbf{x}^r + \bar{a}^r\widehat{\mathbf{p}}^r$)

$$
\begin{aligned}
\widehat{F}(\mathbf{x}^r) - \widehat{F}(\widehat{\mathbf{x}}^{r+1}) &= \widehat{F}(\mathbf{x}^r) - F(\mathbf{x}^r) + F(\mathbf{x}^r) - F(\widehat{\mathbf{x}}^{r+1}) + F(\widehat{\mathbf{x}}^{r+1}) - \widehat{F}(\widehat{\mathbf{x}}^{r+1}) \\
&= f(\mathbf{x}^r, \widehat{\mathbf{y}}(\mathbf{x}^r)) - f(\mathbf{x}^r, \mathbf{y}^*(\mathbf{x}^r)) + F(\mathbf{x}^r) - F(\widehat{\mathbf{x}}^{r+1}) \\
&\qquad\qquad + f(\widehat{\mathbf{x}}^{r+1}, \mathbf{y}^*(\widehat{\mathbf{x}}^{r+1})) - f(\widehat{\mathbf{x}}^{r+1}, \widehat{\mathbf{y}}(\widehat{\mathbf{x}}^{r+1})) \\
&\geq -L_f\|\mathbf{y}^*(\mathbf{x}^r) - \widehat{\mathbf{y}}(\mathbf{x}^r)\| - L_f\|\mathbf{y}^*(\widehat{\mathbf{x}}^{r+1}) - \widehat{\mathbf{y}}(\widehat{\mathbf{x}}^{r+1})\| + F(\mathbf{x}^r) - F(\widehat{\mathbf{x}}^{r+1}) \\
&\geq F(\mathbf{x}^r) - F(\widehat{\mathbf{x}}^{r+1}) - L_f\delta^r - L_f\delta^{r+1},
\end{aligned} \tag{38}
$$

where the first inequality above follows the Lipschitz continuity of $f$, and the second inequality is an application of Assumption 2(a). Incorporating inequality (38) into (37) results to

$$
\frac{F(\mathbf{x}^r) - F(\mathbf{x}^r + \bar{a}^r\widehat{\mathbf{p}}^r)}{\bar{a}_r} - L_f\frac{\delta^r + \delta^{r+1}}{\bar{a}_r} < -\sigma\left[\widehat{\nabla}F(\mathbf{x}^r)\right]^T\widehat{\mathbf{p}}^r - \epsilon(\delta; r), \forall r \in \overline{\mathcal{R}}, r > \bar{r}. \tag{39}
$$

Lebourg's mean value theorem (Lebourg, 1979, Theorem 1.7) implies that

$$
\frac{F(\mathbf{x}^r) - F(\mathbf{x}^r + \bar{a}^r\widehat{\mathbf{p}}^r)}{\bar{a}_r} = \mathbf{u}^{\mathbf{T}}\widehat{\mathbf{p}}^{\mathbf{r}}
$$

with $\mathbf{u} \in \vartheta\mathbf{F}(\mathbf{x}^{\mathbf{r}} + \widetilde{\mathbf{a}}^{\mathbf{r}}\widehat{\mathbf{p}}^{\mathbf{r}})$ and $\widetilde{a}^r \in [0, \bar{a}^r]$, where $\vartheta F(\cdot)$ is the Clarke subdifferntial of $F$. We know that $F$ is almost surely continuously differentiable (Lemma 10) at any $\mathbf{x}^r + \widetilde{a}^r\widehat{\mathbf{p}}^r \in \mathcal{X}$, and so the Clarke subdifferential at $\mathbf{x}^r + \bar{a}^r\widehat{\mathbf{p}}^r$ becomes w.p. 1 equal to $\nabla F(\mathbf{x}^r + \widetilde{a}^r\widehat{\mathbf{p}}^r)$. Note that the we cannot use here the (classical) mean value theorem (Bertsekas, 1998, Prop. 1.23), as in the proof of (Bertsekas, 1998, Prop. 1.2.1), because it requires that the function $F(\mathbf{x})$ is (surely) differentiable on the interval $[\mathbf{x}^r, \mathbf{x}^r + \bar{a}^r\widehat{\mathbf{p}}^r]$.

Then, we can rewrite the expression in (39) as follows

$$
-L_f\beta\frac{\delta^r + \delta^{r+1}}{a_r\|\widehat{\mathbf{d}}^r\|} - [\nabla F(\mathbf{x}^r + \widetilde{a}^r\widehat{\mathbf{p}}^r)]^T\widehat{\mathbf{p}}^r < -\sigma\left[\widehat{\nabla}F(\mathbf{x}^r)\right]^T\widehat{\mathbf{p}}^r - \epsilon(\delta; r), \forall r \in \overline{\mathcal{R}}, r > \bar{r},
$$

where $\widetilde{a}^r \in [0, \bar{a}^r]$.

Using the assumption that $0 \leq \frac{\delta^r}{a^r} \sim \mathcal{O}(c^r)$, where $c^r$ is some sequence with $\lim_{r\to\infty, r\in\overline{\mathcal{R}}} c^r = 0$, and the fact that $\lim_{r\to\infty, r\in\overline{\mathcal{R}}} \|\widehat{\mathbf{d}}^r\| \neq 0$ (because of the assumption that the sequence $\mathbf{x}^r$ converges to a non-stationary point), we compute the limit in the above expression and get

$$
\begin{aligned}
-[\nabla F(\bar{\mathbf{x}})]^T\bar{\mathbf{p}} &< -\sigma[\nabla F(\bar{\mathbf{x}})]^T\bar{\mathbf{p}} \\
0 &< (1 - \sigma)[\nabla F(\bar{\mathbf{x}})]^T\bar{\mathbf{p}} \\
0 &< [\nabla F(\bar{\mathbf{x}})]^T\bar{\mathbf{p}}.
\end{aligned} \tag{40}
$$

However, note that $\left[\widehat{\nabla}F(\mathbf{x}^r)\right]^T\widehat{\mathbf{p}}^r = \frac{\widehat{\nabla}^T F(\mathbf{x}^r)\widehat{\mathbf{d}}^r}{\|\widehat{\mathbf{d}}^r\|}$ and therefore if we take limits in both sides we obtain

$$
[\nabla F(\bar{\mathbf{x}})]^T\bar{\mathbf{p}}^r \leq \frac{\limsup_{r\to\infty, r\in\overline{\mathcal{R}}} \widehat{\nabla}^T F(\mathbf{x}^r)\widehat{\mathbf{d}}^r}{\limsup_{r\to\infty, r\in\overline{\mathcal{R}}} \|\widehat{\mathbf{d}}^r\|} < 0, \tag{41}
$$

due to the gradient-related assumption. We notice that expressions (40) and (41) lead to a contradiction. Therefore, $\bar{\mathbf{x}}$ is a stationary point of $F(\mathbf{x})$.

The proof is now complete. $\qquad\square$

### D.2.4 WEAKLY-CONVEX OBJECTIVE: PROOF OF THEOREM 2

*Proof.* Define $\hat{\mathbf{x}}^r = \arg\min_{\mathbf{z}\in\mathbb{R}^{d_u}} \left\{ H(\mathbf{z}) + \frac{\hat{\rho}}{2}\|\mathbf{x}^r - \mathbf{z}\|^2 \right\}$. Using the definition of Moreau envelope, we have

$$
\begin{aligned}
\mathbb{E}[H_{1/\hat{\rho}}(\mathbf{x}^{r+1})] &\leq \mathbb{E}\Big[F(\hat{\mathbf{x}}^r) + \frac{\hat{\rho}}{2}\|\mathbf{x}^{r+1} - \hat{\mathbf{x}}^r\|^2\Big]\\
&\overset{(a)}{=} \mathbb{E}\Big[F(\hat{\mathbf{x}}^r) + \frac{\hat{\rho}}{2}\|\mathrm{proj}_{\mathcal{X}}(\mathbf{x}^r - \beta\widehat{\nabla}F(\mathbf{x}^r;\xi^r)) - \mathrm{proj}_{\mathcal{X}}(\hat{\mathbf{x}}^r)\|^2\Big]\\
&\overset{(b)}{\leq} \mathbb{E}\Big[F(\hat{\mathbf{x}}^r) + \frac{\hat{\rho}}{2}\|\mathbf{x}^r - \beta\widehat{\nabla}F(\mathbf{x}^r;\xi^r) - \hat{\mathbf{x}}^r\|^2\Big]\\
&= F(\hat{\mathbf{x}}^r) + \frac{\hat{\rho}}{2}\mathbb{E}\Big[\|\mathbf{x}^r - \hat{\mathbf{x}}^r\|^2 - 2\langle\mathbf{x}^r - \hat{\mathbf{x}}^r, \beta\widehat{\nabla}F(\mathbf{x}^r;\xi^r)\rangle + \beta^2\|\widehat{\nabla}F(\mathbf{x}^r;\xi^r)\|^2\Big]\\
&\overset{(c)}{\leq} F(\hat{\mathbf{x}}^r) + \frac{\hat{\rho}}{2}\mathbb{E}\Big[\|\mathbf{x}^r - \hat{\mathbf{x}}^r\|^2 - 2\langle\mathbf{x}^r - \hat{\mathbf{x}}^r, \beta\widehat{\nabla}F(\mathbf{x}^r;\xi^r)\rangle\\
&\qquad\qquad\qquad\qquad + 2\beta^2\|\widehat{\nabla}F(\mathbf{x}^r;\xi^r) - \widehat{\nabla}F(\mathbf{x}^r)\|^2 + 2\beta^2\|\widehat{\nabla}F(\mathbf{x}^r)\|^2\Big]\\
&\overset{(d)}{\leq} F(\hat{\mathbf{x}}^r) + \frac{\hat{\rho}}{2}\Big[\|\mathbf{x}^r - \hat{\mathbf{x}}^r\|^2 - 2\beta\langle\mathbf{x}^r - \hat{\mathbf{x}}^r, \widehat{\nabla}F(\mathbf{x}^r)\rangle + 2\beta^2\big(\sigma_F^2 + \overline{L}_F^2\big)\Big]\\
&\leq F(\hat{\mathbf{x}}^r) + \frac{\hat{\rho}}{2}\Big[\|\mathbf{x}^r - \hat{\mathbf{x}}^r\|^2 + 2\beta\underbrace{\langle\hat{\mathbf{x}}^r - \mathbf{x}^r, \nabla F(\mathbf{x}^r)\rangle}_{\text{Term I}}\\
&\qquad\qquad\qquad + 2\beta\underbrace{\langle\hat{\mathbf{x}}^r - \mathbf{x}^r, \widehat{\nabla}F(\mathbf{x}^r) - \nabla F(\mathbf{x}^r)\rangle}_{\text{Term II}} + 2\beta^2\big(\sigma_F^2 + \overline{L}_F^2\big)\Big],
\end{aligned}
$$
$$(42)$$

where $(a)$ follows from the fact that $\mathbf{x}^{r+1} \in \mathcal{X}$ and $\hat{\mathbf{x}}^r \in \mathcal{X}$; $(b)$ results from the non-expansiveness of the projection operator; $(c)$ uses $\|\mathbf{a} - \mathbf{b}\|^2 = 2\|\mathbf{a}\|^2 + \|\mathbf{b}\|^2$; and $(d)$ results from the application of Lemmas 3 and 4.

Next, considering Term I and Term II separately in (42) above. For Term I, we get using the weak convexity of $F(\cdot)$

$$
\text{Term I} = \langle\hat{\mathbf{x}}^r - \mathbf{x}^r, \nabla F(\mathbf{x}^r)\rangle \leq F(\hat{\mathbf{x}}^r) - F(\mathbf{x}^r) + \frac{\rho}{2}\|\mathbf{x}^r - \hat{\mathbf{x}}^r\|^2
$$

We bound Term II using the Young's inequality as

$$
\begin{aligned}
\text{Term II} &= \langle\hat{\mathbf{x}}^r - \mathbf{x}^r, \widehat{\nabla}F(\mathbf{x}^r) - \nabla F(\mathbf{x}^r)\rangle\\
&\leq \frac{\rho}{2}\|\mathbf{x}^r - \hat{\mathbf{x}}^r\|^2 + \frac{1}{2\rho}\|\widehat{\nabla}F(\mathbf{x}^r) - \nabla F(\mathbf{x}^r)\|^2\\
&\overset{(e)}{\leq} \frac{\rho}{2}\|\mathbf{x}^r - \hat{\mathbf{x}}^r\|^2 + \frac{1}{2\rho}L_F^2\delta^2
\end{aligned}
$$

where $(e)$ follows from Lemma 3. Next, substituting the bounds of Term I and Term II in (42) and using the definition of $\hat{\mathbf{x}}^r$, we get

$$
\begin{aligned}
\mathbb{E}[H_{1/\hat{\rho}}(\mathbf{x}^{r+1})] &\leq F(\hat{\mathbf{x}}^r) + \frac{\hat{\rho}}{2}\|\mathbf{x}^r - \hat{\mathbf{x}}^r\|^2 + \hat{\rho}\beta\Big[F(\hat{\mathbf{x}}^r) - F(\mathbf{x}^r) + \rho\|\mathbf{x}^r - \hat{\mathbf{x}}^r\|^2\Big]\\
&\qquad\qquad\qquad + \frac{\hat{\rho}\beta}{2\rho}L_F^2\delta^2 + \beta^2\hat{\rho}\big(\sigma_F^2 + \overline{L}_F^2\big)\\
&\leq H_{1/\hat{\rho}}(\mathbf{x}^r) + \hat{\rho}\beta\underbrace{\Big[F(\hat{\mathbf{x}}^r) - F(\mathbf{x}^r) + \rho\|\mathbf{x}^r - \hat{\mathbf{x}}^r\|^2\Big]}_{\text{Term III}} + \frac{\hat{\rho}\beta}{2\rho}L_F^2\delta^2 + \beta^2\hat{\rho}\big(\sigma_F^2 + \overline{L}_F^2\big).
\end{aligned}
$$
$$(43)$$

Next, we bound Term III in (43) above.

$$\text{Term III} = F(\hat{\mathbf{x}}^r) - F(\mathbf{x}^r) + \rho\|\mathbf{x}^r - \hat{\mathbf{x}}^r\|^2$$

$$= F(\hat{\mathbf{x}}^r) + \frac{\hat{\rho}}{2}\|\mathbf{x}^r - \hat{\mathbf{x}}^r\|^2 - F(\mathbf{x}^r) + \frac{2\rho - \hat{\rho}}{2}\|\mathbf{x}^r - \hat{\mathbf{x}}^r\|^2$$

$$\leq \frac{3\rho - 2\hat{\rho}}{2}\|\mathbf{x}^r - \hat{\mathbf{x}}^r\|^2 \leq \frac{3\rho - 2\hat{\rho}}{2\hat{\rho}^2}\|\nabla H_{1/\hat{\rho}}(\mathbf{x}^r)\|^2,$$

where the last equality follows from (13) and the first inequality follows from the fact that $F(\mathbf{x}) + \frac{\hat{\rho}}{2}\|\mathbf{x}^r - \mathbf{x}\|^2$ is $(\hat{\rho} - \rho)$-strongly convex. This implies the following

$$F(\hat{\mathbf{x}}^r) + \frac{\hat{\rho}}{2}\|\mathbf{x}^r - \hat{\mathbf{x}}^r\|^2 - F(\mathbf{x}^r) \leq -\langle \nabla F(\hat{\mathbf{x}}^r) + \hat{\rho}(\mathbf{x}^r - \hat{\mathbf{x}}^r), x^r - \hat{x}^r \rangle - \frac{\hat{\rho} - \rho}{2}\|\mathbf{x}^r - \hat{\mathbf{x}}^r\|^2$$

$$\leq \frac{\rho - \hat{\rho}}{2}\|\mathbf{x}^r - \hat{\mathbf{x}}^r\|^2,$$

where the second inequality results from the definition of $\hat{\mathbf{x}}^r$.

Finally, substituting Term III in (43) and rearranging the terms we get:

$$\beta\left[\frac{2\hat{\rho} - 3\rho}{2\hat{\rho}}\right]\|\nabla H_{1/\hat{\rho}}(\mathbf{x}^r)\|^2 \leq \mathbb{E}\big[H_{1/\hat{\rho}}(\mathbf{x}^r) - H_{1/\hat{\rho}}(\mathbf{x}^{r+1})\big] + \beta^2\hat{\rho}(\sigma_F^2 + \overline{L}_F^2) + \frac{\hat{\rho}\beta}{2\rho}L_F^2\delta^2.$$

Summing over all $r \in \{0, 1, \ldots, T-1\}$ and dividing by $T$, we get

$$\frac{1}{T}\sum_{r=0}^{T-1}\|\nabla H_{1/\hat{\rho}}(\mathbf{x}^r)\|^2 \leq \left[\frac{2\hat{\rho}}{2\hat{\rho} - 3\rho}\right]\left[\frac{H_{1/\hat{\rho}}(\mathbf{x}^0) - H^*}{\beta T} + \beta\hat{\rho}(\sigma_F^2 + \overline{L}_F^2) + \frac{\hat{\rho}}{2\rho}L_F^2\delta^2\right].$$

Therefore, we have the result. □

### D.2.5 STRONGLY-CONVEX OBJECTIVE: PROOF OF THEOREM 3

*Proof.* Using the update rule of the Algorithm 2, we have

$$\mathbb{E}\|\mathbf{x}^{r+1} - \mathbf{x}^*\|^2 = \mathbb{E}\|\text{proj}_X(\mathbf{x}^r - \beta^r\widehat{\nabla}F(\mathbf{x}^r;\xi^r)) - \mathbf{x}^*\|^2$$

$$= \mathbb{E}\|\text{proj}_X(\mathbf{x}^r - \beta^r\widehat{\nabla}F(\mathbf{x}^r;\xi^r)) - \text{proj}_X(\mathbf{x}^*)\|^2$$

$$\overset{(a)}{\leq} \mathbb{E}\|\mathbf{x}^r - \beta^r\widehat{\nabla}F(\mathbf{x}^r;\xi^r) - \mathbf{x}^*\|^2$$

$$= \mathbb{E}\Big[\|\mathbf{x}^r - \mathbf{x}^*\|^2 + (\beta^r)^2\|\widehat{\nabla}F(\mathbf{x}^r;\xi^r)\|^2 - 2\beta^r\langle\mathbf{x}^r - \mathbf{x}^*, \widehat{\nabla}F(\mathbf{x}^r;\xi^r)\rangle\Big]$$

$$\overset{(b)}{\leq} \mathbb{E}\Big[\|\mathbf{x}^r - \mathbf{x}^*\|^2 + 2(\beta^r)^2\|\widehat{\nabla}F(\mathbf{x}^r;\xi^r) - \widehat{\nabla}F(\mathbf{x}^r)\|^2$$

$$+ 2(\beta^r)^2\|\widehat{\nabla}F(\mathbf{x}^r)\|^2 - 2\beta^r\langle\mathbf{x}^r - \mathbf{x}^*, \widehat{\nabla}F(\mathbf{x}^r)\rangle\Big]$$

$$\overset{(c)}{\leq} \mathbb{E}\Big[\|\mathbf{x}^r - \mathbf{x}^*\|^2 + 2(\beta^r)^2(\sigma_F^2 + B_F^2) - 2\beta^r\langle\mathbf{x}^r - \mathbf{x}^*, \widehat{\nabla}F(\mathbf{x}^r)\rangle\Big]$$

$$\leq \mathbb{E}\Big[\|\mathbf{x}^r - \mathbf{x}^*\|^2 + 2(\beta^r)^2(\sigma_F^2 + B_F^2) - 2\beta^r\langle\mathbf{x}^r - \mathbf{x}^*, \nabla F(\mathbf{x}^r)\rangle$$

$$- 2\beta^r\langle\mathbf{x}^r - \mathbf{x}^*, \widehat{\nabla}F(\mathbf{x}^r) - \nabla F(\mathbf{x}^r)\rangle\Big]$$

$$\overset{(d)}{\leq} \mathbb{E}\Big[\|\mathbf{x}^r - \mathbf{x}^*\|^2 + 2(\beta^r)^2(\sigma_F^2 + B_F^2) - 2\beta^r[F(\mathbf{x}^r) - F^*] - \mu_F\beta^r\|\mathbf{x}^r - \mathbf{x}^*\|^2$$

$$+ 2\beta^r\|\mathbf{x}^r - \mathbf{x}^*\|\|\widehat{\nabla}F(\mathbf{x}^r) - \nabla F(\mathbf{x}^r)\|\Big]$$

$$\overset{(e)}{\leq} \mathbb{E}\Big[(1 - \mu_F\beta^r)\|\mathbf{x}^r - \mathbf{x}^*\|^2 + 2(\beta^r)^2(\sigma_F^2 + B_F^2) - 2\beta^r[F(\mathbf{x}^r) - F^*] + 2\beta^r D_{\mathcal{X}}L_F\delta\Big]$$

where in $(a)$ the non-expansive property of the projection operator is applied, in $(b)$ we add and subtract the term $\widehat{\nabla}F(\mathbf{x}^r)$, use the well-known inequality $\|\mathbf{a} + \mathbf{b}\|^2 \leq 2\|\mathbf{a}\|^2 + 2\|\mathbf{b}\|^2$, and in

the last term we utilize Lemma 4; $(c)$ results from the application of Lemmas 3 and 4; $(d)$ uses strong-convexity (Assumption 6) of $F(\cdot)$ and Cauchy-Schwartz inequality; finally, $(e)$ results from the application of Lemma 3 and Assumption 1(b).

Rearranging the terms, we get

$$2\beta^r \mathbb{E}[F(\mathbf{x}^r) - F^*] \le \mathbb{E}\Big[(1 - \mu_F \beta^r)\|\mathbf{x}^r - \mathbf{x}^*\|^2 - \|\mathbf{x}^{r+1} - \mathbf{x}^*\|^2 + 2(\beta^r)^2(\sigma_F^2 + B_F^2) + 2\beta^r D_{\mathcal{X}} L_F \delta\Big]$$

$$\mathbb{E}[F(\mathbf{x}^r) - F^*] \le \mathbb{E}\Big[\Big(\frac{1}{2\beta^r} - \frac{\mu_F}{2}\Big)\|\mathbf{x}^r - \mathbf{x}^*\|^2 - \frac{1}{2\beta^r}\|\mathbf{x}^{r+1} - \mathbf{x}^*\|^2 + \beta^r(\sigma_F^2 + B_F^2) + D_{\mathcal{X}} L_F \delta\Big].$$

Summing over $r \in \{0, \ldots, T-1\}$, multiplying by $1/T$ and using the fact that $\beta^r = \frac{1}{\mu_F(r+1)}$, we get

$$\frac{1}{T}\sum_{r=0}^{T-1} \mathbb{E}[F(\mathbf{x}^r) - F^*] \le \frac{\mu_F}{2T}\sum_{r=0}^{T-1} \mathbb{E}\Big[r\|\mathbf{x}^r - \mathbf{x}^*\|^2 - (r+1)\|\mathbf{x}^{r+1} - \mathbf{x}^*\|^2\Big]$$

$$+ \frac{1}{T}\sum_{r=0}^{T-1} \beta^r(\sigma_F^2 + B_F^2) + D_{\mathcal{X}} L_F \delta.$$

Telescoping the sum we get the following

$$\frac{1}{T}\sum_{r=0}^{T-1} \mathbb{E}[F(\mathbf{x}^r) - F^*] \le -\frac{\mu_F}{2}\|\mathbf{x}^T - \mathbf{x}^*\|^2 + \frac{(\sigma_F^2 + B_F^2)}{\mu_F}\frac{\log(T)}{T} + D_{\mathcal{X}} L_F \delta$$

$$\le \frac{(\sigma_F^2 + B_F^2)}{\mu_F}\frac{\log(T)}{T} + D_{\mathcal{X}} L_F \delta.$$

Therefore, we have the proof. $\qquad\square$

### D.2.6 CONVEX OBJECTIVE: PROOF OF THEOREM 4

*Proof.* From the update rule of Algorithm 2, we have for $\beta^r = \beta$ for all $r \in \{0, 1, \ldots, T-1\}$

$$\mathbb{E}[\|\mathbf{x}^{r+1} - \mathbf{x}^*\|^2] = \mathbb{E}[\|\text{proj}_{\mathcal{X}}(\mathbf{x}^r - \beta\widehat{\nabla}F(\mathbf{x}^r; \xi^r)) - \mathbf{x}^*\|^2]$$

$$\overset{(a)}{=} \mathbb{E}[\|\text{proj}_{\mathcal{X}}(\mathbf{x}^r - \beta\widehat{\nabla}F(\mathbf{x}^r); \xi^r) - \text{proj}_{\mathcal{X}}(\mathbf{x}^*)\|^2]$$

$$\overset{(b)}{\le} \mathbb{E}[\|\mathbf{x}^r - \beta\widehat{\nabla}F(\mathbf{x}^r; \xi^r) - \mathbf{x}^*\|^2]$$

$$\overset{(c)}{=} \mathbb{E}[\|\mathbf{x}^r - \mathbf{x}^*\|^2 + \beta^2\|\widehat{\nabla}F(\mathbf{x}^r; \xi^r)\|^2 - 2\beta\langle\mathbf{x}^r - \mathbf{x}^*, \widehat{\nabla}F(\mathbf{x}^r; \xi^r)\rangle]$$

$$\overset{(d)}{\le} \mathbb{E}[\|\mathbf{x}^r - \mathbf{x}^*\|^2 + 2\beta^2\|\widehat{\nabla}F(\mathbf{x}^r; \xi^r) - \widehat{\nabla}F(\mathbf{x}^r)\|^2 + 2\beta^2\|\widehat{\nabla}F(\mathbf{x}^r)\|^2$$

$$- 2\beta\langle\mathbf{x}^r - \mathbf{x}^*, \nabla F(\mathbf{x}^r)\rangle - 2\beta\langle\mathbf{x}^r - \mathbf{x}^*, \widehat{\nabla}F(\mathbf{x}^r) - \nabla F(\mathbf{x}^r)\rangle]$$

$$\overset{(e)}{\le} \mathbb{E}[\|\mathbf{x}^r - \mathbf{x}^*\|^2 + 2\beta^2\sigma_F^2 + 2\beta^2\overline{L}_F^2 - 2\beta(F(\mathbf{x}^r) - F^*)$$

$$+ 2\beta\|\mathbf{x}^r - \mathbf{x}^*\|\,\|\widehat{\nabla}F(\mathbf{x}^r) - \nabla F(\mathbf{x}^r)\|]$$

$$\overset{(f)}{\le} \mathbb{E}[\|\mathbf{x}^r - \mathbf{x}^*\|^2 + 2\beta^2(\sigma_F^2 + \overline{L}_F^2) - 2\beta(F(\mathbf{x}^r) - F^*) + 2\beta D_{\mathcal{X}} L_F \delta]$$

where $(a)$ follows from the fact that $\mathbf{x}^* \in \mathcal{X}$; $(b)$ results from the non-expansiveness of the projection operator; $(c)$ uses $\|\mathbf{a} - \mathbf{b}\|^2 = \|\mathbf{a}\|^2 + \|\mathbf{b}\|^2 - 2\langle\mathbf{a}, \mathbf{b}\rangle$; $(d)$ utilizes the fact that $\|\mathbf{a} - \mathbf{b}\|^2 = 2\|\mathbf{a}\|^2 + 2\|\mathbf{b}\|^2$ and unbiased gradient from Lemma 4; $(e)$ results from Lemmas 3 and 4, the convexity assumption of the implicit function (Assumption 6 with $\mu_F = 0$) and the Cauchy-Schwartz inequality; and $(f)$ results from Assumption 1(b) and Lemma 3.

Summing over $r \in \{0, 1 \ldots, T-1\}$, multiplying by $1/T$ and rearranging the terms, we get

$$\frac{1}{T}\sum_{r=0}^{T-1} \mathbb{E}[F(\mathbf{x}^r) - F^*] \le \frac{\|\mathbf{x}^1 - \mathbf{x}^*\|^2}{2\beta T} + \beta(\sigma_F^2 + \overline{L}_F^2) + D_{\mathcal{X}} L_F \delta.$$

Using Jensen's inequality and denoting $\underline{\mathbf{x}} = \frac{1}{T} \sum_{r=0}^{T-1} \mathbf{x}^r$, we get

$$\mathbb{E}[F(\underline{\mathbf{x}}) - F^*] \leq \frac{\|\mathbf{x}^1 - \mathbf{x}^*\|^2}{2\beta T} + \beta(\sigma_F^2 + \overline{L}_F^2) + D_\mathcal{X} L_F \delta.$$

Therefore, we have the proof. $\qquad\square$

