# OpenReview forum: "Linearly Constrained Bilevel Optimization: A Smoothed Implicit Gradient Approach"
_ICLR.cc/2023/Conference — Submitted to ICLR 2023_

### Official Review · Reviewer_eqw3 · 2022-10-23

**Confidence:** 2
**Clarity, Quality, Novelty And Reproducibility:** See above.
**Correctness:** 4
**Technical Novelty And Significance:** 4
**Empirical Novelty And Significance:** 3
**Recommendation:** 6

**Strength And Weaknesses:**

On the smoothing technique:
This paper proposes a very novel way of smoothing the constrained bilevel optimization problem by slightly perturbing the objective function of the inner problem. The perturbation only has a small influence on the main problem (Proposition 2), but it makes the problem smooth and differentiable. I think the idea is novel and interesting, and may also be useful in other related optimization problems.


On the convergence guarantees:
1)	The non-convex non-smooth problem is in general NP-hard, and it is a bit supervising that the algorithm can still converge to the stationary point (Although asymptotically). But I have a hard time understanding Theorem 1, especially the definition of $hat{d}^r$. Can the authors help me understand this condition (e.g., intuitively, what is $hat{d}^r$? and why the algorithm can still converge when the problem is non-convex non-smooth)?
2)	On the other hand, when the functions are non-smooth by convex/strongly convex, I am not sure about the significance of the proposed algorithm: It seems to me that, now since we have a closed form of gradient, we could just to GD (or other more advanced algorithms) and still have similar rates. What are the advantages of the proposed methods?

On the writing:
This paper is in general very easy to read.


Other comments:
This work reminds me of Nesterov’s excessive gap technique [1]/ Nesterov’s smoothing technique [2] for bilevel (min-max) constrained optimization problem. For the inner function, they added a smoothed regularizer to smooth the function instead of using perturbation. Moreover, some recent work shows that in some cases perturbation and regularization are strongly related [3]. I wonder if the authors could add some comments on this point?



[1] https://epubs.siam.org/doi/pdf/10.1137/S1052623403422285
[2] Smooth minimization of non-smooth functions.
http://luthuli.cs.uiuc.edu/~daf/courses/Optimization/MRFpapers/nesterov05.pdf
[3] Perturbation Techniques in Online
Learning and Optimization https://dept.stat.lsa.umich.edu/~tewaria/research/abernethy16perturbation.pdf

There are two definitions of y^*(x) (1b and 5b), one is the exact inner solution and the other one is the approximation. I wonder if the authors could use different notions for the two to make it more clear?


**Summary Of The Paper:**

This paper studies a special class of bilevel optimization, where the inner problem has linear constraints. For this problem, one major difficulty is that outer (main) problem might be non-differentiable. The authors elegantly address this challenge by constructing a smoothed approximation of the original problem via adding noise to the inner problem. The show their smoothed version is a good approximation of the original problem, and has differentiable gradients. They then developed implicit gradient methods on the smoothed problem, and prove nice convergence rates for the original problem.

**Summary Of The Review:**

I find the smoothed technique for constrained bilevel optimization is novel and interesting, and I vote for acceptance.

---

> ### Author Response · Authors · 2022-11-18
> **Response to Reviewer eqw3 (3)**
>
> >This work reminds me of Nesterov’s excessive gap technique [1] Nesterov’s smoothing technique [2] for bilevel (min-max) constrained optimization problem. For the inner function, they added a smoothed regularizer to smooth the function instead of using perturbation. Moreover, some recent work shows that in some cases perturbation and regularization are strongly related [3]. I wonder if the authors could add some comments on this point?
>
> > [1] https://epubs.siam.org/doi/pdf/10.1137/S1052623403422285
> >
> >[2] Smooth minimization of non-smooth functions. http://luthuli.cs.uiuc.edu/~daf/courses/Optimization/MRFpapers/nesterov05.pdf
> >
> >[3] Perturbation Techniques in Online Learning and Optimization https://dept.stat.lsa.umich.edu/~tewaria/research/abernethy16perturbation.pdf
>
> **Response:** At a high-level the smoothing techniques of [1],[2] and our work's are similar, in the sense that they are introduced to make certain objectives differentiable. However, the setting and mechanism with which differentiability is ensured are different. We provide more details below.
>
> In [1],[2] the authors add a quadratic strongly convex regularization term (i.e., the smoothing term) to make a certain inner max function differentiable and Lipschitz smooth. On the other hand, in our work we add a random linear perturbation of the form $q^{T}y$ (i.e., the smoothing term) on the lower-level problem to ensure that the strict complementarity (SC) property holds (albeit with probability 1). The SC property combined with Assumption 1 imply the differentiablity of the implicit function $F(x)$, at any given $x$, and offer a way to compute a closed form expression for the gradient $\nabla F(x)$
>
> Comparing our work with [3], our understanding is that the main utility of adding a regularizer (FTRL algorithm) or a perturbation (FTPL algorithm) to the problems at hand, is to make the relevant algorithms more stable, and not to make some objective differentiable. There is a discussion from a smoothing perspective about FTRL and FTPL and how they are related, however it does not seem very relevant to our work. The FTRL smoothing is similar to the one in [1],[2], while FTPL can be seen as an instance of stochastic smoothing where an expectation over the random perturbation is computed (and this is not the case in our work).
>
> > **Comment:** There are two definitions of $y^\ast(x)$ (1b and 5b), one is the exact inner solution and the other one is the approximation. I wonder if the authors could use different notions for the two to make it more clear?
>
> **Response:** Thanks for your comment. We changed the notations in the revised paper. Specifically, we are using the notation $\overline{y}^{\ast}(x)$ for the solution of the lower-level task (1b) of the original bilevel problem, and the notation $y^\ast(x)$ for the solution of the perturbed lower-level task (5).

---

> ### Author Response · Authors · 2022-11-18
> **Response to  Reviewer eqw3 (2)**
>
> > *" But I have a hard time understanding Theorem 1, especially the definition of $\hat{d}^r$. Can the authors help me understand this condition (e.g., intuitively, what is ? and why the algorithm can still converge when the problem is non-convex non-smooth)?"*
>
> Allow us to explain the proposed [D]SIGD algorithm in more detail. The [D]SIGD algorithm follows the (idea of) the gradient projection method with Armijo stepsize rule of [sec. 2.2,2.3, Bertsekas1998]. The key idea of this method is the computation of a feasible descent direction $d^{r}$ and of a stepsize $\alpha^{r}$, such that if we take one step along that direction, i.e., $x^{r+1} = x^{r} + \alpha^{r} d^{r}$, the new iterate remains inside the constraint set $\mathcal{X}$. One such feasible direction [sec. 2.3.1, Bertsekas1998] is formulated as follows: $d^{r} = \tilde{x}^{r}-x^{r}$ with  $\tilde{x}^{r}= proj_{X} (x^{r} - s \nabla F(x^{r}))$. To make this more clear, notice that $\tilde{x}^{r} = proj_{X} (x^{r} - s \nabla F(x^{r}))$ takes one gradient descent step from the current iterate, and projects the new iterate back to the constraint set $\mathcal{X}$. Then, $d^{r} = \tilde{x}^{r}-x^{r}$ is a feasible direction from the current iterate $x^{r}$. Finally, the gradient projection method with Armijo rule, enjoys similar asymptotic convergence guarantees (see [prop  2.2.1, Bertsekas1998]) with the (unconstrained) method mentioned above, without assuming that the objective is Lipschitz smooth.
>
> Note that in the proposed method [D]SIGD we do not follow the gradient projection method exactly as given in [sec. 2.2,2.3, Bertsekas1998], but rather we only follow the general idea. More precisely, differently than the setting presented above, we assume that we do not have access to the exact implicit gradient $\nabla F(x)$, only to the approximate one $\hat{\nabla} F(x)$. As a result, we employ the approximate implicit gradient $\hat{\nabla} F(x)$ in the  feasible direction $\widehat{d}^{r}$ computation (line 5 in Algorithm 1), and in addition we modify the original Armijo rule condition to account for that approximation (line 6 in Algorithm 1).
>
> [Bertsekas1998]: Dimitri P Bertsekas. Nonlinear programming, 2nd ed. Athena Scientific Belmont, MA, 1998
>
> > 2. On the other hand, when the functions are non-smooth by convex/strongly convex, I am not sure about the significance of the proposed algorithm: It seems to me that, now since we have a closed form of gradient, we could just to GD (or other more advanced algorithms) and still have similar rates. What are the advantages of the proposed methods?
>
> **Response:** Thanks for the comment. In the case where the implicit objective $F(x)$ is convex/strongly-convex (or weakly convex) we do actually propose a GD-type algorithm, as suggested by the reviewer. More precisely, [S]SIGD is a stochastic projected **gradient descent** method. Given the fact that we have a closed-form expression for the implicit gradient, the significance of [S]SIGD lies in its simplicity. After obtaining an (approximate) estimate of the implicit gradient at each iteration, [S]SIGD performs a single (projected) gradient step.

---

> ### Author Response · Authors · 2022-11-18
> **Response to Reviewer eqw3**
>
> > On the convergence guarantees:
> >
> > 1. The non-convex non-smooth problem is in general NP-hard, and it is a bit surprising that the algorithm can still converge to the stationary point (Although asymptotically). But I have a hard time understanding Theorem 1, especially the definition of $\hat{d}^r$. Can the authors help me understand this condition (e.g., intuitively, what is ? and why the algorithm can still converge when the problem is non-convex non-smooth)?
>
> **Response:** We thank you for your comments.
>
> > *"The non-convex non-smooth problem is in general NP-hard, and it is a bit surprising that the algorithm can still converge to the stationary point (Although asymptotically)"*
>
> To begin with, the bilevel problem we study is indeed non-smooth (i.e., non-differentiable). However, as we explain in sec. 2, we introduce the perturbation-based "smoothing" technique, where at any fixed $x \in \mathcal{X}$, we modify the lower-level objective $h(x,y)$ by adding the random ($q$ is a random vector) linear perturbation term $q^{T}y$. This modification and Assumption 1 suffice to ensure (with probability $1$) the differentiability of the implicit function $F(x)$ at the given point $x\in\mathcal{X}$, and offer a way to compute the gradient formula (see Lemma 2). Our proposed algorithms leverage this perturbation-based "smoothing" technique, and as a result they are effectively solving smooth problems.
>
> Further, even though the original problem is eventually "smoothed", the implicit objective we are optimizing, is still non-Lipschitz-smooth, i.e., it does not have Lipschitz continuous gradients. This non-convex non-Lipschitz-smooth minimization problem is indeed a hard problem. However, it is still possible to show convergence to a stationary solution, albeit asymptotically (without convergence rate) and by employing a line-search-type method. This is not something we show here for the first time, but it is an established result in literature (e.g., see [sec. 1.2.2, Bertsekas1998]). In our work, we extend this line-search method and its results to our bilevel setting, and show (asymptotic) convergence to stationary solutions. We provide more details below.
>
> Consider the gradient method described in [sec. 1.2, Bertsekas1998] for an unconstrained minimization problem, where at each iteration the following step is executed: $x^{r+1} = x^{r} + \alpha^{r} d^{r}$; $\alpha^{r}$ is the stepsize and $d^{r}$ is a descent direction. If the stepsize $\alpha^{r}$ is selected using a line-search approach, such as applying the Armijo rule, and a "gradient-related" condition [eq. (1.13), Bertsekas1998] holds, it can be shown that all the limit points of this method are stationary points [prop. 1.2.1, Bertsekas1998]. Note that this convergence result holds **without** imposing the Lipschitz smoothness assumption on the objective. In our work, we extend these types of methods and proofs to our specific setting. As a result, establishing (asymptotic) convergence, even in the absence of Lipschitz smoothness, is not something surprising.

---

### Official Review · Reviewer_YhfE · 2022-10-24

**Confidence:** 4
**Correctness:** 3
**Technical Novelty And Significance:** 3
**Empirical Novelty And Significance:** 3
**Recommendation:** 8

**Clarity, Quality, Novelty And Reproducibility:**

The paper is fairly well written, idea is interesting, and the tackled problem is important.

**Strength And Weaknesses:**

Weaknesses:

**Literature review** Non-smooth optimization
- The provided results are very interesting and seems to extend older and more restrictive results [1] [3] [9]. More recent results on implicit differentiation for non-smooth problems also include [5] and [6].
- In particular, assumptions 1c and 1e seem to imply assumption 7 of [3] or assumption 2 of [4]. For Lasso-type problems, [1] and [3] restricts the linear system to solve only to the non-zeros coefficients (ie the non-active constraints in the dual). It seems that you observe the same kind of computational speedups for $\nabla {\bar \lambda}^*(x) $, but not for $\nabla_{y^*}(x)$. Could you comment on this?
- Related to the previous point, could you comment on the algorithmic complexity of the proposed methods? (formula in Lemma 2 for instance). It feels like one could inverse a linear system of size the dimension of the active constraints (using the tangent space of the projection operator as in [4])
- In the same vein, if the constraint in the inner bilevel optimization problem is rewritten as an indicator over a set, would it be possible to apply the framework of [4] to your problem?

**Literature review** Convex programming differentiation:
- It seems that other works previously obtained implicit differentiation for quadratic programming [7] and more general constrained optimization problems [8]. How does the proposed implicit differentiation formula compares to this previous approaches?


**Proposition 1**: Proposition 1 seems misleading to me, it requires Assumption 1 to be true for all $x$, which is very strong. It seems to me that Assumption 1 is true for almost every $x$, and thus the $x \mapsto G(x)$ is continuous by part, not on all $\khi$. Could authors comment on this.

**Assumption 3** is very strong, and is usually assumed only locally, see assumption 4 of [3], or Assumption 3 of [4].

**Experiments**
- I found it underwhelming that authors do not propose experiments on problem with non-separable matrix constraints. The proposed constrained problems are separable: on these examples one could use [3] and [4] to compute the hypergradient. **In other words: the problems considered in the experiments could be solved with preexisting techniques**.
- I think there are plenty of other interesting non-separable problems, I was wondering if authors considered experiments on more complex problems, and if authors planned to released open-source code, potentially compatible with pytorch?
- I am not sure what is the takeaway message from the *Adversarial Learning* part, could authors comment on it?


Typos:
p27 in appendix "Proof of theorem1" >> "Proof of Theorem 1"



[1] Mairal, J., Bach, F. and Ponce, J., 2011. Task-driven dictionary learning. IEEE transactions on pattern analysis and machine intelligence, 34(4)

[2] Deledalle, C.A., Vaiter, S., Fadili, J. and Peyré, G., 2014. Stein Unbiased GrAdient estimator of the Risk (SUGAR) for multiple parameter selection. SIAM Journal on Imaging Sciences, 7(4), pp.2448-2487.

[3] Bertrand, Q., Klopfenstein, Q., Massias, M., Blondel, M., Vaiter, S., Gramfort, A. and Salmon, J. Implicit differentiation for fast hyperparameter selection in non-smooth convex learning. JMLR 2022.

[4] Mehmood, S. and Ochs, P., 2022. Fixed-Point Automatic Differentiation of Forward--Backward Splitting Algorithms for Partly Smooth Functions. arXiv preprint arXiv:2208.03107.

[5] Bolte, J., Le, T., Pauwels, E. and Silveti-Falls, T., 2021. Nonsmooth implicit differentiation for machine-learning and optimization. Advances in neural information processing systems, 34, pp.13537-13549.

[6] Bolte, J., Pauwels, E. and Vaiter, S., 2022. Automatic differentiation of nonsmooth iterative algorithms. arXiv preprint arXiv:2206.00457.

[7] Amos, B. and Kolter, J.Z., 2017, July. Optnet: Differentiable optimization as a layer in neural networks. In International Conference on Machine Learning (pp. 136-145). PMLR.

[8] Agrawal, A., Amos, B., Barratt, S., Boyd, S., Diamond, S. and Kolter, J.Z., 2019. Differentiable convex optimization layers. Advances in neural information processing systems, 32.

[9] Bertrand, Q., Klopfenstein, Q., Blondel, M., Vaiter, S., Gramfort, A. and Salmon, J., 2020, November. Implicit differentiation of Lasso-type models for hyperparameter optimization. In International Conference on Machine Learning (pp. 810-821). PMLR


**Summary Of The Paper:**

Authors extend implicit differentiation to constraint bilevel optimization problem. Assuming usual qualification conditions, they provide a implicit differentiation formula. Experiments are proposed on toy  bilevel problem, and real adversarial learning.


**Summary Of The Review:**

I think that differentiating efficiently constrained optimization problem is extremely important.
My current score is due to some important references missed in the literature review, as well as problems which can be solved with preexisting techniques in the experiments. **If authors solved these two issues (which can take some time I agree) I will raise my score**.

---

> ### Author Response · Authors · 2022-11-18
> **Response to Reviewer YhfE (4)**
>
> > **Experiments:**
> >
> >  I found it underwhelming that authors do not propose experiments on problem with non-separable matrix constraints. The proposed constrained problems are separable: on these examples one could use [3] and [4] to compute the hypergradient. In other words: the problems considered in the experiments could be solved with preexisting techniques.
>
> **Response:** Thank you for bringing this issue to our attention. In the revised version of this work we added (in Appendix C.1; the new results are highlighted with blue color) numerical experiments with non-separable matrix constraints, e.g. with constraints such as $-5 \leq y_{1}+y_{2} \leq 5$. Our results show that the two proposed algorithms ([D]SIGD and [S]SIGD), and the baseline PDBO converge to a stationary solution. In addition, it appears that [D]SIGD and [S]SIGD converge faster compared to PDBO. Finally, as mentioned by the reviewer such examples cannot be handled within the framework of [3].
>
> > I think there are plenty of other interesting non-separable problems, I was wondering if authors considered experiments on more complex problems, and if authors planned to released open-source code, potentially compatible with pytorch?
>
> **Response:** We did not consider experiments on more complex problems, since we deemed that an extensive experimental evaluation was outside the current scope of our work. The aim of our work is to study a wide range of aspects of the linearly constrained bilevel problem, rather than focus on a specific one. These include the problem properties (e.g.,non-differentiability and how to deal with it), implicit gradient (formulation, approximations and their errors), algorithms and convergence analysis (deterministic and stochastic algorithms, in the absence of Lipschitz smoothness), and finally experiments that support the theoretical findings and show the potential of the proposed approach. However, in the future we would be interested in delving deeper into the individual aspects, including evaluating the performance of the proposed algorithm in more complex settings, and potentially releasing the related code.
>
> > I am not sure what is the takeaway message from the Adversarial Learning part, could authors comment on it?
>
> **Response:** We thank the reviewer for raising this concern. The goal of presenting Adversarial Learning experiments is two-fold. First, the goal is to demonstrate that the adversarial learning problem is a special linearly constrained bilevel problem, and can be solved using algorithms designed for solving constrained bilevel problems. Secondly, and more importantly, via numerical experiments we establish that the smoothing-based algorithmic framework SIGD, proposed for solving linearly constrained bilevel problems, provides a competitive algorithm for solving adversarial learning problems. Specifically, our experiments demonstrate that SIGD outperforms the popular adversarial learning baseline algorithm, AT, and closely matches the performance of the stronger baseline TRADES.

---

> ### Author Response · Authors · 2022-11-18
> **Response to Reviewer YhfE (3)**
>
> > Proposition 1: Proposition 1 seems misleading to me, it requires Assumption 1 to be true for all $x$, which is very strong. It seems to me that Assumption 1 is true for almost every $x$, and thus the $x \to G(x)$ is continuous by part, not on all $\mathcal{X}$. Could authors comment on this.
>
> **Response:** Thanks for your comment. Below we comment on each one of the assumptions in Assumption 1, and on why the requirement for these to hold for every $x \in \mathcal{X}$ is not too strong.
>
> * Ass. 1b,d describe the properties of the constraints sets and thus they are independent of $x$. Nonetheless, note that Ass. 1b is crucial for the validity of Proposition 1.
> * Ass. 1a ensures the differentiability of the upper $f(x,y)$ and lower-level $g(x,y)$ objectives of the bilevel problem, for every $x \in \mathcal{X}, y \in \mathbb{R}^{d_l}$. In the context of our work (gradient-based methods for bilevel optimization), this is a very common and reasonable assumption (e.g. see  [(Ghadimi & Wang, 2018; Hong et al., 2020; Chen et al., 2021a)]).
> * Ass. 1c assumes that the lower-level objective $g(x,y)$ is strongly convex in $y$ for every $x \in \mathcal{X}$. This is a typical assumption in bilevel optimization literature, e.g. see the works [(Ghadimi & Wang, 2018; Hong et al., 2020; Chen et al., 2021a)] about implicit gradient methods.
> * Ass. 1e is introduced to ensure that the matrix of active constraints $\overline{A}$ at the optimal solution of the lower-level problem $y^{\ast}(x)$ has full row rank, for every $x \in \mathcal{X}$. This is actually, the well known Linear Independence Constraint Qualification (LICQ) regularity condition which is common in the constrained optimization literature. Most importantly note that Ass. 1e is not involved in the proof of Proposition 1.
>
> To summarize the above discussion, we believe that it is reasonable to assume that Assumption 1 holds for all $x \in \mathcal{X}$. Then the continuity of $G(x)$ for all $x \in \mathcal{X}$ follows naturally.
>
> > Assumption 3 is very strong, and is usually assumed only locally, see assumption 4 of [3], or Assumption 3 of [4].
>
> **Response:** First, we comment on each one of the assumptions in Assumption 3 (note that Ass. 3 became Ass. 2 in the revised paper), and explain the reasons why they are reasonable.
>
> * Ass. 3a,b are introduced to ensure that the approximate solutions $\widehat{y}^{\ast}(x)$ (that are needed in the formulation of the approximate implicit gradient of eq. (9)) of the lower-level problem satisfy certain properties. More precisely, Ass. 3a bounds the error (i.e., $\|\widehat{y}^{\ast}(x)- y^{\ast}(x)\|$) of the approximate solution, and Ass. 3b, ensures that the approximate solution is feasible. Since the lower-level is a strongly-convex linearly constrained one, we can easily find an approximate solution that satisfies both assumptions using known methods (e.g. projected gradient descent or by using some convex optimization solver).
>
> * Ass. 3c states the active set of the approximate solution $\widehat{y}^{\ast}(x)$ is the same as the active set of the exact one $y^{\ast}(x)$. We know that this condition can be ensured. Specifically, from  Calamai & Moré (1987, Theorem 4.1) we know that if $\widehat{y}^{k}(x) \in \mathcal{Y}$ is an arbitrary sequence that converges to a non-degenerate (i.e., Assumption 1e and strict complementrity holds) stationary solution $y^{\ast}(x)$, then there exists an integer $k_{0}$ such that $\overline{A}(y^{\ast}(x))=\overline{A}(\widehat{y}^{k}(x))$, $\forall k>k_{0}$. In addition, for certain special constraint sets we can provide an upper bound for $k_{0}; please see remark 1 at pg. 5 for more details.
>
> Next, we compare the Assumption 3 with Assumption 4 of [3] and notice that the two assumptions are no different in terms of whether they hold locally or globally. Assumption 4 in [3] states the non-degeneracy property holds "for any solution $\hat{\beta}$ of Problem (1)", where $\hat{\beta}$ is the solution of the lower-level problem (1). This is no different than our statements in Assumption 3, where we assume that the properties hold at $y^{\ast}(x)$ for every $x \in \mathcal{X}$ or equivalently for every solution of the lower-level problem $y^{\ast}$.

---

> ### Author Response · Authors · 2022-11-18
> **Response to Reviewer YhfE (2)**
>
> > - Related to the previous point, could you comment on the algorithmic complexity of the proposed methods? (formula in Lemma 2 for instance). It feels like one could inverse a linear system of size the dimension of the active constraints (using the tangent space of the projection operator as in [4])
>
> **Response:** The computation of the implicit gradient formula in (6),(7) in Lemma 2 at a given $x$ includes the following computations (assuming access to the Hessians of the lower-level objective $g(x,y)$): the solution of the problem in the lower-level, the matrix $\overline{A}$ of active constraints, matrix multiplications, and matrix inversions. Let us denote with $d=d_{u}=d_{l}$ the problem dimension (where $d_{u}, d_{l}$ are the dimensions of the variables of the upper and lower-level problems, respectively) and with $k$ the number of constraints.
>
> The lower-level problem is a strongly-convex one with linear inequality constraints. Therefore, it can be solved using some convex optimization solver, e.g. using interior point methods to obtain a solution with a complexity $\mathcal{O}(poly(d,k) \log(d/\epsilon))$, where $poly(\cdot)$ is some polynomial and $\epsilon$ is the solution accuracy [Nesterov,1994]. Moreover, we can practically ignore the complexity of the computing the active constraints set, since we only have to check which of the $k$ constraints are active at a given solution $y^{\ast}(x)$). The matrix multiplications and inversions incur a cost of $\mathcal{O}(d^{3}+d^{2}\overline{k}+d\overline{k}^{2})$ and $\mathcal{O}(d^{3} + \overline{k}^{3})$, respectively, where $\overline{k}$ is the number of active constraints. In particular, we note that the complexity depends on the number of active constraints $\overline{k}$.
>
> [Nesterov,1994]: Interior-point polynomial algorithms in convex programming, Nesterov, Yurii and Nemirovskii, Arkadii, 1994
>
> > - In the same vein, if the constraint in the inner bilevel optimization problem is rewritten as an indicator over a set, would it be possible to apply the framework of [4] to your problem?
>
> **Response:** We thank the reviewer for the question. We note that as pointed out by the reviewer the linearly constrained LL problem can be equivalently reformulated as an unconstrained problem with an indicator function over the set of linear inequality constraints as:
> $$\min_{y \in \mathcal{Y}} G(x,y) = g(x, y) + \mathbb{I}_\mathcal{Z} ~~~~~~~~~~~~~~~~ (P)$$
> where $\mathbb{I}_\mathcal{Z} = 0$ if $y \in \mathcal{Z}$ and $\mathbb{I}_\mathcal{Z} = \infty$ otherwise; and $\mathcal{Z} = \{ y \in \mathcal{Y} : Ay \leq b \}$. Note that this is a non-smooth LL problem. The work in [4] introduces sufficient conditions (Assumptions 1, 2, and 3 in [4]) required for the mapping $y^\ast(x)$ to be locally differentiable in an open neighborhood of any $x \in \mathcal{X}$, please see Theorem 12 in [4]. Note that in our problem there might be points $x \in \mathcal{X}$ where the mapping  $y^\ast(x)$ can be non-differentiable (please see the Example on page 4 of the revised manuscript). As a consequence, the derivative expression derived in [4] (Theorem 12) will not hold at such points. In contrast, our approach ensures almost sure differentialibity at such points (through perturbation).  In summary, we believe that the approach introduced in [4] may not be directly applied to our problem; however, it is worth studying if a subset of our problem (e.g., some special linear constraints) will satisfy the sufficient conditions developed in [4], therefore making the entire problem differentiable.
>
> > Literature review Convex programming differentiation:
>
> > It seems that other works previously obtained implicit differentiation for quadratic programming [7] and more general constrained optimization problems [8]. How does the proposed implicit differentiation formula compares to this previous approaches?
>
> **Response:** Overall, both papers [7],[8] and our work follow the implicit differentiation approach for computing gradients/Jacobians of solutions of optimization problems. However, the setting in which this computation takes place (e.g. neural network layers in [7,8] or bilevel optimization problems in our work), the approach followed (e.g., numerical computation of Jacobians in [7,8] or analytical solutions in our work), and the type of derived results (e.g. contrary to [7,8], in our work we propose algorithms and perform a rigorous theoretical analysis), are different. As a result, it is not clear how a comparison between the implicit differentiation formulas can be performed. Nonetheless, we discuss these references (along with other related references mentioned by reviewer 1APN) in the revised version of this work.

---

> ### Author Response · Authors · 2022-11-18
> **Response to Reviewer YhfE**
>
> > Literature review Non-smooth optimization
>
> > - The provided results are very interesting and seems to extend older and more restrictive results [1] [3] [9]. More recent results on implicit differentiation for non-smooth problems also include [5] and [6].
>
> **Response:** Thank you for bringing to our attention these references. We addded some of the missing references in the revised version of our work (in Appendix A).
>
> > - In particular, assumptions 1c and 1e seem to imply assumption 7 of [3] or assumption 2 of [4].
>
> **Response:**  Thanks for the comment. Yes, it seems that Assumptions 1c an 1e imply Assumption 7 of [3]. However, work [3] is "simpler" compared to our work in other aspects. Therefore we cannot claim that either work studies a setting that is a special case of the other work's setting.
>
> For instance, the authors in [3] considers a simple bilevel (see problems (1),(2) in [3]) problem where the implicit function is of the form $F(x)=f(y^{\ast}(x))$ and not $F(x)=f(x, y^{\ast}(x))$ as in our case. In addition, we notice that in Assumption H2 in Theorem 9 the authors make a strong assumption, they assume that the mapping $y^{\ast}(x)$ is continuously differentiable. In contrast, our approach does not assume any differentiability, but tries to show rigorously that under certain conditions the mapping $y^{\ast}(x)$ is differentiable at any given $x$ (with probability 1).
>
> > For Lasso-type problems, [1] and [3] restricts the linear system to solve only to the non-zeros coefficients (i.e., the non-active constraints in the dual). It seems that you observe the same kind of computational speedups for $\nabla \bar{\lambda}^\ast(x)$ but not for $\nabla y^\ast(x)$. Could you comment on this?
>
> **Response:** Thanks for the comment. Yes, in both works the implicit gradient formula involves computations that depend at certain properties (e.g. active constraints) of the point at which the gradient is being evaluated. This can potentially result to a speedup of the computation process.
>
> Specifically, in our case the submatrix $\overline{A}$ of the constraints matrix $A$ is involved in the implicit gradient computation. $\overline{A}$ is constructed by stacking the rows of $A$ that correspond to the active constraints at some given point. Then, we can reduce the computation of the formula
> \begin{align}
> &\nabla \overline{\lambda}^{\ast}(x)  =  -   \big[ \overline{A}\big[\nabla_{yy}^{2}g(x,y^{\ast}(x)) \big]^{-1}\overline{A}^{T} \big]^{-1}
>      \big[\overline{A}\big[\nabla_{yy}^2 g(x,y^{\ast}(x))\big]^{-1} \nabla_{xy}^2 g(x,y^{\ast}(x))\big]
> \end{align}
> to the solution of the following (set of) linear systems
> \begin{align}
> & \big[ \overline{A}\big[\nabla_{yy}^{2}g(x,y^{\ast}(x)) \big]^{-1}\overline{A}^{T} \big] \nabla \overline{\lambda}^{\ast}(x) = -\big[\overline{A}\big[\nabla_{yy}^2 g(x,y^{\ast}(x))\big]^{-1} \nabla_{xy}^2 g(x,y^{\ast}(x))\big]
> \end{align}
> Note that $\nabla \bar{\lambda}^{\ast}(x)$ is gradient matrix of dimensions $\bar{k} \times d_{u}$, where $\bar{k}$ is the number of active constraints. Therefore the above expression involves the solution of $d_{u}$ systems, where the dimensions of each system depend on the number of active constraints $\bar{k}$. As a result, it is possible to observe some kind of computational speedup in the calculation of the implicit gradient at points with a small number of active constraints. Moreover, as noted by the reviewer, the formula of $\nabla y^{\ast}(x)$ does not depend on $\overline{A}$ (at least not directly). Note however that the implicit gradient $\nabla y^{\ast}(x)$ is essentially a single formula (we can plug $\nabla \lambda^{\ast}(x)$ into $\nabla y^{\ast}(x)$); it is currently broken in two equations for assisting the presentation. Therefore, overall we can say that some kind of computational speedup in the calculation of the implicit gradient $\nabla y^{\ast}(x)$ can be observed at points with a small number of active constraints.

---

> ### Comment · Reviewer_YhfE · 2022-12-06
> **Response to author response**
>
> The author response answered my questions and I thus raised my score

---

### Official Review · Reviewer_gXua · 2022-10-27

**Confidence:** 3
**Clarity, Quality, Novelty And Reproducibility:** See above.
**Correctness:** 2
**Technical Novelty And Significance:** 2
**Empirical Novelty And Significance:** 4
**Recommendation:** 5

**Strength And Weaknesses:**

Strengths
======================
  - The problem of finding provably convergent methods for bilevel optimization problems with constraints on the LL problem is very well-motivated.


Comments to the authors
========================

  Below I explain an issue that I have with the current presentation of the paper. I feel that I don't understand something very important and this makes it difficult for me to judge the overall contribution of the paper. Since ICLR gives us this opportunity I would like to first discuss about the following issue and if this is resolved I commit to updating fast my review to address the contributions of the paper.

  My issue is that I don't see how the addition of the term $q^T y$ makes the function $y^*(x)$ smooth. In particular, I don't even see how this works for the example in page 4. Let me explain with the following simpler example.

$\min_{x \in [0, 1]} x + y*(x) s.t. y^*(x) = \arg\min ((y - x)^2 | 1/2 <= y <= 1) (1)$

  This is almost the same example as in page 4 expect that I slightly changed the objective and the boundaries of y, which I don't understand why they are so complicated in the example in the paper. Now the solution $y^*(x)$ of the LL problem in $(1)$ is similar to the example of the paper. $y^*(x) = 1/2$ for $x <= 1/2$ and $y^*(x) = x$ for $x > 1/2$. So, indeed $y^*(x)$ is not differentiable at $x = 1/2$. Also, unless I am missing something, (1) satisfies Assumptions 1 and 2. Now let's consider the perturbed problem
  $y^*(x) = \arg\min {(y - x)^2 + q y| 1/2 <= y <= 1}$
  where $q$ is sampled from some distribution. In this case, the solution $y^*(x)$ becomes $y^*(x) = 1/2$ for $x <= (1 + q)/2$ and $y^*(x) = x$ for $x > (1 + q)/2$. So, $y^*(x)$ is still non-differentiable with some constant probability, assuming that the distribution of q has constant probability in the interval $[0, 1]$. The only thing that changed is the point where $y^*(x)$ is non-differentiable. I get exactly the same behavior when solving the example of page 4 in Mathematica it is just that the closed form solution is much more involved.

  The only way that I can see to make $y^*(x)$ smooth is if we define
  $y_q^*(x) = \arg\min
9(y - x)^2 + q y| 1/2 <= y <= 1)$
  and then $y^*(x) = \mathbb{E}_q[y_q^*(x)]$, then maybe there is a way to show that $y^*(x)$ is smooth, at least this is indeed happening in the example of page 4 and the above example. But:
  1. the authors never mention that they take the expected value with respect to q, and
  2. looking in Algorithms 1 and 2, the authors seems to first sample q from its distribution and then proceed with a fixed value of q for the rest of the algorithm as if $y^*(x)$ is smooth. If they indeed want to take the expected value with respect to q then they need to get a different sample q every time they compute the gradient of $y^*(x)$. Also, they need to argue that this additional stochasticity does not affect their algorithm.

  I am very confused with the above. I apologize in advance if I am missing something trivial here and I am looking forward to your response.

(I apologize for the format issue in my review, I just realized and fixed it)

**Summary Of The Paper:**

In this paper, the authors consider the problem of bilevel optimization when the lower level optimization (LL) problem is strongly convex and has to satisfy some linear constraints. The main issue with the constraints in the lower level optimization problem is that the solution of the LL problem might not be differentiable and hence applying standard gradient based method to the upper level optimization is not possible. The authors claim that:
  1. They provide a randomized smoothing procedure that makes the LL problem differentiable. Then they show that there is a way to compute the derivatives of the solution of the LL problem approximately and that this is enough for solving the bilevel problem.
  2. The authors then provide experiments that illustrate the advantage of their method compared to state-of-the art.

**Summary Of The Review:**

  Above I explained an issue that I have with the current presentation of the paper. I feel that I don't understand something very important and this makes it difficult for me to judge the overall contribution of the paper. Since ICLR gives us this opportunity I would like to first discuss about the following issue and if this is resolved I commit to updating fast my review to address the contributions of the paper.

---

> ### Author Response · Authors · 2022-11-18
> **Response to Reviewer gXua (2)**
>
> > "*2. looking in Algorithms 1 and 2, the authors seems to first sample $q$ from its distribution and then proceed with a fixed value of $q$ for the rest of the algorithm as if $y^\ast(x)$ is smooth. If they indeed want to take the expected value with respect to $q$ then they need to get a different sample $q$ every time they compute the gradient of $y^\ast(x)$. Also, they need to argue that this additional stochasticity does not affect their algorithm.*"
>
> Further, as argued above the perturbation-based "smoothing" technique applies only on a fixed $x$ with the sampling of a new $q$ each time. However, in the context of algorithm design, where we make use of the "smoothing" technique, there are two choices depending on how we sample $q$. More precisely, we can either sample a new $q$ each time we encounter a new iterate $x^{r}$ or we can sample a single $q$ in the beginning and keep it fixed during the algorithm execution. In the former case, the implicit function $F(x)$ we are optimizing changes at each iteration. This combined with the fact that the gradient of the resulting function is only differentiable, but may not have Lipshcitz gradient,  makes the analysis challenging. Therefore, in this work we adopt the latter approach (i.e., sample a single $q$), and postpone the study of the alternative approach for the future; see Sec. 3.1 for more details.
>
> > "*I am very confused with the above. I apologize in advance if I am missing something trivial here and I am looking forward to your response.*"
>
> We apologize for the confusion. To make the presentation clear, we have revised the structure of the paper (the changes are highlighted with blue color). Now the logic of the technical part of the paper is given as below:
>
> 1. For a given point $x\in X$, we show that it is possible to perturb the problem a bit so to obtain the gradient (Sec. 2).
> 2. We design algorithms which utilizes the above perturbation idea; in the algorithm, we simply generate a single perturbation at the beginning, and use this perturbation vector to compute the approximate gradients for the rest of the iterates; we explain in Sec. 3.1 why this is a reasonable choice in the context of algorithm design. Specifically, Lemma 5 shows that, for a sequence of countable points, the function $F(\cdot)$ (defined by using a fixed but randomly generated vector $q$ in the lower level problem) is differentiable almost surely  (this statement was also given in the previous version of the algorithm). Using this argument, we assume that the gradient $\nabla F(\cdot)$ exists over all the iterates generated by the algorithm. Admittedly, this argument is slightly non-rigorous, in the sense that the Lemma first picks the sequence of points, then samples $q$, while in the algorithm, $q$ is first sampled then the sequence of points is generated; However, this kind of argument has been often used in machine learning literature for proving convergence of SGD based algorithms in the interpolation regime; for example please see Theorem 7 in [Liu et. al, 2021] where it is first assumed that the sequence of iterates lie within a ball of the optimal solution, and then the algorithm is executed.
>
> [Liu et. al, 2021] Loss landscapes and optimization in over-parameterized non-linear systems and neural networks, arxiv 2021.
>
> 3. We show that the proposed algorithm is in fact solving a "surrogate" problem, and this surrogate problem is very close to the original problem. Then we perform analysis on the surrogate problem (Sec. 3.2).
>
> Note that we have only revised the narrative of the paper in order to clarify the point on the perturbation; theoretical analysis remains the same as in the previous version.
>
> Further,  we also removed the part of the example showing the "smoothing" of $y^{\ast}(x)$, since we no longer think that it is helpful, it might in fact create confusion.
>
> Finally, we note that although we choose to analyze the version where a single perturbation is used (mainly for the reason of simplicity), we believe that what the reviewer suggested is reasonable, in the sense that we can also analyze the stochastic version of the problem where the perturbation is a function of the feasible solution $x\in X$. We can either take expectation of the solution $y^*(x)$, as suggested by the reviewer, of take the expectation of the outer-objective function, $F(x)$. Due to space limitation, we will study these two cases in the journal version of this paper.

---

> > ### Comment · Area_Chair_h5ZA · 2022-12-12
> > **Need clarification**
> >
> > Could you comment regarding the specific example raised by the reviewer? Especially the following sentence
> >
> > > So, $y^*(x)$  is still non-differentiable with some constant probability, assuming that the distribution of q has constant probability in the interval $[0, 1]$
> >
> > I agree with the reviewer: adding perturbation only once just shifts the kink, but doesn't smooth it out.

---

> > > ### Author Response · Authors · 2022-12-13
> > > **Response**
> > >
> > > **Response:** We thank the area chair and the reviewer for raising this concern. The statement of the reviewer is correct that the perturbation might shift the non-differentiability, however, we would like to point out that our claim that, at any given point of interest $x \in \mathcal{X}$, $y^\ast(x)$ is differentiable with probability 1, does not contradict with the reviewer's statement. Here we provide the reasoning why $y^{\ast}(x)$ is differentiable with probability 1.
> > >
> > > First, let's say that $q \sim \mathcal{Q}$, where $\mathcal{Q}$ is some continuous distribution over $[0,1]$. Then, there is no fixed (deterministic) point $\bar{x}=(1+q)/2$  such that $y^{\ast}(\bar{x})$ is non-differentiable. Instead, the appropriate way to see this is the following. We select some point $\bar{x} \in [0,1]$ and we sample a random $q \sim \mathcal{Q}$. Then, if $q=2\bar{x}-1$ the point $\bar{x}$ is non-differentiable, otherwise, i.e., $q \in [0,1] \setminus \{2\bar{x}-1\}$, $y^{\ast}(x)$ is differentiable at $\bar{x}$. The set $\{2\bar{x}-1\}$ is a singleton, and so it is a subset of the sample space $[0,1]$ with probability measure 0. Therefore, given $\bar{x}$, the probability of $y^{\ast}(x)$ being differentiable at $\bar{x}$ is $1$. Finally, we say that the mapping $y^{\ast}(x)$ is "smoothed" in the sense that we can apply the same reasoning for any selected point $\bar{x}$, sampling each time a new $q \sim \mathcal{Q}$.

---

> > > > ### Comment · Reviewer_gXua · 2022-12-14
> > > > **Thank you but I still have an issue...**
> > > >
> > > > Thank you for clarifying this and fixing the example, now it makes more sense!
> > > >
> > > > I still have an issue with sampling q once thought. You added Lemma 5 to address this issue but I cannot see how Lemma 5 can be applied to the analysis of Algorithms 1 and 2. In particular, Lemma 5 is true when we first have a fixed set of countable points and then we pick a q at random. But the set of points that Algorithm 1 or 2 generate depend on q because q is chosen in the beginning of the algorithm. So even the point after the first iteration of Algorithm 1 or 2 will depend on q and hence I cannot see how you can apply Lemma 5 anymore. If you cannot apply Lemma 5 then I cannot see how you can use the fact that the next point that the algorithm will encounter will be differentiable and, if I understand correctly, this is used in the proofs of Theorem 1 and 2. Am I missing something?

---

> ### Author Response · Authors · 2022-11-18
> **Response to Reviewer gXua**
>
> **Response:** We would like to thank the reviewer for this question, because it allows us to clarify a central while subtle aspect of our work. It also gives us the opportunity to streamline the presentation to (hopefully) make the point clearer.
>
> First, let us clarify that, in the context of our work, "smoothing" means a modification of lower-level objective $h(x,y)$ in a way that ensures that a certain set of conditions is satisfied (with probability $1$). These conditions suffice to imply the differentiability of the resulting implicit function $F(x)$ at a **fixed** $x \in \mathcal{X}$ and, importantly, allows us to compute a closed-form gradient expression at the given point $x\in\mathcal{X}$.
>
> More precisely, we introduce the perturbation-based "smoothing" technique, where at any fixed $x \in \mathcal{X}$, we modify the lower-level objective $h(x,y)$ by adding the linear perturbation term $q^{T}y$; where $q$ is a random vector sampled form some continuous distribution $\mathcal{Q}$. This modification suffices to ensure that the strict complementarity (SC) property holds at the lower-level problem, for the given $x \in \mathcal{X}$, with probability $1$ (see Lemma 1). Then, the SC property and Assumption 1 ensure differentiability of the implicit function $F(x)$ at the given point $x\in\mathcal{X}$, and offer a way to compute the gradient formula (see Lemma 2). Note that, in the absence of the perturbation term, we would have had to introduce the SC property as an assumption. We do not make such assumption, and instead we modify the problem such that the SC property follows naturally (albeit with probability 1).
>
> To make things more clear we would like to stress the following subtle points:
>
> 1. The reviewer is correct in pointing out that, for two arbitrary points $x,\hat{x}\in X$, it requires two different perturbations, one for each point, to make both $F(x)$ and $F(\hat{x})$ differentiable. If we only generate one perturbation vector for $x$, then it is possible that there exists other solutions $\hat{x}\ne x$, for which $F(\hat{x})$ is not differentiable (as the reviewer has correctly pointed on in the comments above). However, this does not contradict with our derivations in Sec. 1-2, since over there we have only focused on the differentiability property for a single point $x\in X$. This might appear confusing since in the Algorithms section (Sec. 3) we claim that we only sample $q$ once. However, in Sec. 3.1 in revised manuscript we explain why in the context of algorithm design this is a reasonable choice.
>
> 2. When we say that, for a fixed $x$, the SC property holds with probability $1$, we mean that the probability is computed over the sample space $\mathcal{Q}$. Notice that it is still possible to have a set of measure $0$ on the sample space $\mathcal{Q}$, such that the property does not hold.

---

### Official Review · Reviewer_1APN · 2022-10-30

**Confidence:** 4
**Correctness:** 4
**Technical Novelty And Significance:** 3
**Empirical Novelty And Significance:** Not applicable
**Recommendation:** 8

**Clarity, Quality, Novelty And Reproducibility:**

The papers is clearly written. The idea proposed is simple yet powerful; however, it is not properly situated within the implicit layers/differentiable optimization literature. Sufficient details are provided for reproducibility.

**Strength And Weaknesses:**

Strengths:
* The approach is simple yet powerful. It leverages implicit differentiation and gradient descent to address an important class of bi-level optimization problems, and does so in a very principled manner - establishing theoretically for which cases differentiation-based techniques are likely to work, introducing perturbation-based smoothing to expand the set of problems for which such methods might (approximately) work, and then utilizing techniques such as adaptive step sizing to ensure convergence in a wide variety of cases.
* The experiments, though not overwhelmingly convincing, do reasonably demonstrate the efficacy of the proposed method.
* The paper is accessible and clearly written.

Weaknesses:
* The authors do not seem to be aware of the large body of work in implicit differentiation of optimization problems within the deep learning literature - see, e.g., the list below. The authors should be sure to discuss and situate their work within this literature, as it is very closely related. In particular, [8] below also uses implicit differentiation to address a class of bilevel optimization problems via gradient descent.
* Minor: In Equation (1a), it would be cleaner if the notation $f$ were not overloaded.
* Minor: The paper is missing a Conclusion section, which is standard for such papers.

Overarching note: On the one hand, the existing literature on implicit layers diminishes the contribution of this work from an empirical perspective. On the other hand, this paper perhaps inadvertently fills several gaps in the implicit layers literature (which often lacks principled analyses), thus contributing to the strengths of the present work. For instance, these works often do not contain analysis of the quality of the gradient under an approximate solution to the optimization solution, but the present work does.

Example literature on implicit differentiation of optimization problems:
* [1] Brandon Amos and J Zico Kolter. “OptNet: Differentiable optimization as a layer in neural networks.” International Conference on Machine Learning. 2017.
* [2] Josip Djolonga and Andreas Krause. “Differentiable Learning of Submodular Models.” Advances in Neural Information Processing Systems (NeurIPS). 2017.
* [3] Priya L. Donti, Brandon Amos, and J. Zico Kolter. “Task-based End-to-End Model Learning in Stochastic Optimization.” Advances in Neural Information Processing Systems. 2017.
* [4] Sebastian Tschiatschek, Aytunc Sahin, and Andreas Krause. “Differentiable submodular maximization.” International Joint Conference on Artificial Intelligence. 2018.
* [5] Akshay Agrawal, Brandon Amos, Shane Barratt, Stephen Boyd, Steven Diamond, and J Zico Kolter. “Differentiable convex optimization layers.” Advances in Neural Information Processing Systems (NeurIPS). 2019.
* [6] Stephen Gould, Richard Hartley, and Dylan Campbell. “Deep declarative networks.” IEEE Transactions on Pattern Analysis and Machine Intelligence 44.8 (2021), 3988–4004.
* [7] http://implicit-layers-tutorial.org/
* [8] Priya L. Donti, Aayushya Agarwal, Neeraj Vijay Bedmutha, Larry Pileggi, and J. Zico Kolter. “Adversarially Robust Learning for Security-Constrained Optimal Power Flow.” Advances in Neural Information Processing Systems. 2021.

**Summary Of The Paper:**

This paper provides gradient descent-based for solving bi-level optimization problems where the lower-level problem has a strongly convex objective and linear inequality constraints, using implicit gradients. The authors derive conditions under which the implicit objective is differentiable, and propose a perturbation-based smoothing technique to address non-differentiable cases. The authors also propose adaptive step sizing techniques for the gradient descent procedure to ensure convergence. They empirically demonstrate the efficacy of their method on two test cases: (1) linearly constrained quadratic bilevel problems, and (2) adversarial training.

**Summary Of The Review:**

This paper provides a simple, powerful, and principled approach for addressing a class of bi-level optimization problems using gradient descent, specifically leveraging implicit gradients (combined with perturbation-based smoothing and adaptive step sizing). The authors could do a better job situating their work within the literature on implicit layers/differentiable optimization - in my impression, the existence of that literature serves to diminish some of the empirical contributions of the present work, but enhances the significance of some of the theoretical contributions.

---

> ### Author Response · Authors · 2022-11-18
> **Response to Reviewer 1APN**
>
> > **Comment:** The authors do not seem to be aware of the large body of work in implicit differentiation of optimization problems within the deep learning literature - see, e.g., the list below. The authors should be sure to discuss and situate their work within this literature, as it is very closely related. In particular, [8] below also uses implicit differentiation to address a class of bilevel optimization problems via gradient descent.
>
> **Response:** We thank the reviewer for bringing these works (let us refer to those works as "DL works" for simplicity) to our attention. DL works are indeed related to our work, in the sense that at the core of both of them lies the computation of the gradient/Jacobian of the solution of an optimization problem. However the setting (e.g. neural networks or bilevel optimization) in which this computation takes place, and the type of the derived results (e.g., numerical computation or algorithms and convergence analysis), are different. Therefore, our paper complements the results from the DL line of works.
>
> More precisely, some key differences (these might not describe with complete accuracy all of the works, but we think that they capture the general idea) between our work and the DL ones  are the following:
> * In our work we consider a constrained bilevel optimization problem and we are interested in analyzing this problem from an optimization perspective. On the other hand, in the DL works the optimization problems that are studied describe the input-output relationships of neural networks layers and the main focus lies in deriving Jacobians for the backward pass.
> * In our work we study a special bilevel problem (the constraints are linear) and derive a closed form expression for the implicit gradient. On the contrary, in the DL works the underlying problems have more general constraints and the Jacobian is usually computed using numerical methods (e.g., solving iteratively a system of KKT equations), rather than analytically. For instance, this is the case in [8].
> * In our work the focus is on studying the properties of the bilevel problem (e.g. differentiability, approximation errors), developing (deterministic and stochastic) algorithms, and performing a convergence analysis. On the other hand, DL works focus mainly on the Jacobian computation and its implementation. For instance, [8] lacks theoretical convergence guarantees.
>
> In the revised version of our paper, we added a brief comment about the DL line of works on the main text, as well as a more detailed discussion of these references in the appendix A (due to space limitations on the main text).
>
> > **Comment:** Minor: In Equation (1a), it would be cleaner if the notation $f$ were not overloaded.
>
> **Response:** Thanks for your comment. In the revised version we changed the notation for the stochastic objective (inside the expectation) from $f$ to $\tilde{f}$.
>
>
> > **Comment:** Minor: The paper is missing a Conclusion section, which is standard for such papers.
>
> **Response:** In our original (and revised) submission we omitted the conclusion due to lack of space.

---

> > ### Comment · Reviewer_1APN · 2022-11-28
> > **Response**
> >
> > Thanks to the authors for adding the discussion of the DL works. I have raised my score accordingly.
> >
> > That said, I do still think the following sentence at the end of the abstract is an overclaim: "To our knowledge, this is the first time that (implicit) gradient based methods have been developed and analyzed for the considered class of bilevel problems." The authors should tone down this claim, and consider addressing in the submission itself (rather than only in this reviewer discussion) how the current work improves on prior works (e.g. but not limited to [8]) - for instance, by giving guarantees whereas the prior work does not.

---

> > > ### Author Response · Authors · 2022-11-30
> > > **Response to Reviewer 1APN**
> > >
> > > Thank you for your comments and the positive evaluation of our work.
> > >
> > > We thank the reviewer for the suggestions. To address the reviewer’s concern we will update the abstract in the revised version of the manuscript, and we will also modify the discussions in the paper to reflect properly the novelty of our work. Regarding the comparison of our work with the prior ones, we would like to mention that we had already added a discussion in the revised paper. Specifically, there is a brief comment in page 2, with a more detailed comparison provided in Appendix A (Page 14). The changes are highlighted with blue colored font.

---

### Decision · Program_Chairs · 2023-01-20

**Decision:**

Reject

**Justification For Why Not Higher Score:**

Incomplete proof for Theorem 1 and 2

**Justification For Why Not Lower Score:**

N/A

**Metareview: Summary, Strengths And Weaknesses:**

In this paper, the authors consider the problem of bilevel optimization when the inner optimization problem is strongly convex and has to satisfy some linear constraints.

Strengths:
- The approach is simple
- The experiments demonstrate the efficacy of the proposed method

Weaknesses:
- As pointed out by reviewer gXua, and agreed by the authors themselves, Lemma 5 is currently not applicable in the proofs of Theorem 1 and 2: the proof is incomplete

The authors say that they will add an assumption that all the intermediate points of the algorithm are differentiable for their proof to be complete. As pointed out by reviewer gXua (1) such an assumption defies the whole purpose of the paper which was to make the function  smooth, and (2) it should be possible to be proven, so it is worth to wait for a newer version that includes this proof as well (rather than using an assumption).

Therefore, while the paper is promising, we believe it is not ready for publication in its current state and recommend rejection.